# ENTANGLED SCHRÖDINGER BRIDGE MATCHING

## ABSTRACT

Simulating trajectories of multi-particle systems on complex energy landscapes is a central task in molecular dynamics (MD) and drug discovery, but remains challenging at scale due to computationally expensive and long simulations. Previous approaches leverage techniques such as flow or Schrödinger bridge matching to implicitly learn joint trajectories through data snapshots. However, many systems, including biomolecular systems and heterogeneous cell populations, undergo *dynamic* interactions that evolve over their trajectory and cannot be captured through static snapshots. To close this gap, we introduce **Entangled Schrödinger Bridge Matching (EntangledSBM)**, a framework that learns the first- and second-order stochastic dynamics of interacting, multi-particle systems where the direction and magnitude of each particle's path depend dynamically on the paths of the other particles. We define the Entangled Schrödinger Bridge (EntangledSB) problem as solving a coupled system of bias forces that *entangle* particle velocities. We show that our framework accurately simulates heterogeneous cell populations under perturbations and rare transitions in high-dimensional biomolecular systems.

## 1 INTRODUCTION

Schrödinger bridge matching (SBM) has enabled significant progress in problems ranging from molecular dynamics (MD) simulation (Holdijk et al., 2023) to cell state modeling (Tong et al., 2024) by effectively learning dynamic trajectories between initial and target distributions. Diverse SBM frameworks have been developed for learning trajectories with minimized state-cost (Liu et al., 2023) to branching trajectories that map from a single initial distribution (Tang et al., 2025a). Most of these frameworks assume that the particle acts independently or undergoes interactions that can be implicitly captured through training on static snapshots of the system. Existing approaches for modeling interacting particle systems rely on the mean-field assumption, where all particles are exchangeable and interact only through averaged effects (Liu et al., 2022; Zhang et al., 2025d). While this assumption may hold for homogeneous particle dynamics, it fails to describe the heterogeneous dynamics observed in more complex domains, such as conformationally dynamic proteins (Vincoff et al., 2025; Chen et al., 2023), heterogeneous cell populations (Smela et al., 2023b), or interacting token sequences (Geshkovski et al., 2023). In these settings, the motion of each particle depends not only on its own position but also on the evolving configuration of surrounding particles.

To accurately model such dependencies, the joint evolution of an $n$-particle system must be described by a coupled probability distribution that transports the initial distribution $\pi_0(\boldsymbol{X}_0)$ to the target distribution $\pi_{\mathcal{B}}(\boldsymbol{X}_T)$ in the phase space of both positions and velocities $\boldsymbol{X}_t = (\boldsymbol{R}_t, \boldsymbol{V}_t) \in \mathcal{X}$. However, modeling these interactions requires learning to simulate second-order dynamics, where interactions between velocity fields evolve over time, which remains largely unexplored. To address this gap, we introduce **Entangled Schrödinger Bridge Matching (EntangledSBM)**, a novel framework that learns interacting second-order dynamics of $n$-particle systems, capturing dependencies on both the static position and dynamic velocities of the system at each time step.

**Contributions**   Our contributions can be summarized as follows: **(1)** We formulate the Entangled Schrödinger Bridge (EntangledSB) problem, which considers the optimal path between distributions following second-order Langevin dynamics with an entangled bias force (Sec 3). **(2)** We introduce **EntangledSBM**, a novel parameterization of the bias force that can be conditioned, *at inference-time*, on a target distribution or terminal state, enabling generalizable sampling of diverse target distributions (Sec 4). **(3)** We evaluate EntangledSBM on mapping cell cluster dynamics under drug

perturbations (Sec 5.1) and transition path sampling of high-dimensional molecular systems (Sec 5.2).

**Related Works** We provide a comprehensive discussion on related works in App A.

## 2 PRELIMINARIES

**Langevin Dynamics** A time-evolving $n$-particle molecular system can be represented as $\boldsymbol{X}_t = (\boldsymbol{R}_t, \boldsymbol{V}_t)$, where $\boldsymbol{R}_t = \{\boldsymbol{r}_t^i \in \mathbb{R}^d\}_{i=1}^n$ denotes the set of coordinates $\boldsymbol{V}_t = \{\boldsymbol{v}_t^i \in \mathbb{R}^d\}_{i=1}^n$ denotes the set of velocities of each particle $i$. The evolution of the positions and velocities of the system given a potential energy function $U : \mathcal{X} \to \mathbb{R}$ can be modelled with *Langevin dynamics*, which effectively captures the motion of particles under **conservative forces** between particles and **stochastic collisions** with the surrounding environment (Bussi & Parrinello, 2007; Yang et al., 2006) using a pair of stochastic differential equations (SDEs) defined as

$$d\boldsymbol{r}_t^i = \boldsymbol{v}_t^i dt, \quad d\boldsymbol{v}_t^i = \frac{-\nabla_{\boldsymbol{r}_t^i} U(\boldsymbol{R}_t)}{m_i} dt - \gamma \boldsymbol{v}_t^i dt + \sqrt{\frac{2\gamma k_B \tau}{m_i}} d\boldsymbol{W}_t^i \tag{1}$$

where $m_i$ is the mass of each particle, $\gamma$ is the friction coefficient, $k_B$ is the Boltzmann constant, $\tau$ is the temperature, and $d\boldsymbol{W}_t$ is standard Brownian motion. In molecular dynamics (MD) simulations of biomolecules, the particles undergo *underdamped* Langevin dynamics with small $\gamma$, where *inertia* is not negligible.

Many biological systems, including cell clusters for cell-state trajectory simulation, can be modeled with *overdamped* Langevin dynamics, where inertia is negligible but the system still undergoes external forces that define its motion. This can be represented with the first-order SDE given by

$$d\boldsymbol{r}_t^i = \frac{-\nabla_{\boldsymbol{r}_t^i} U(\boldsymbol{R}_t)}{\gamma} dt + \sqrt{\frac{2k_B \tau}{\gamma}} d\boldsymbol{W}_t^i \tag{2}$$

**Schrödinger Bridge Matching** Tasks such as simulating cell state trajectories and transition path sampling aim to simulate the Langevin dynamics from an initial distribution to a desired target state or distribution. Given an initial distribution $\pi_{\mathcal{A}}$ and a target distribution $\pi_{\mathcal{B}}$, we define the distribution of paths $\boldsymbol{X}_{0:T} := (\boldsymbol{X}_t)_{t \in [0,T]}$ satisfying the endpoint constraints $\boldsymbol{R}_0 \sim \pi_{\mathcal{A}}$ and $\boldsymbol{R}_T \sim \pi_{\mathcal{B}}$ as the *optimal bridge distribution* $\mathbb{P}^\star(\boldsymbol{X}_{0:T})$ defined as

$$\mathbb{P}^\star(\boldsymbol{X}_{0:T}) = \frac{1}{Z} \mathbb{P}^0(\boldsymbol{X}_{0:T}) \pi_{\mathcal{B}}(\boldsymbol{R}_T), \quad Z = \mathbb{E}_{\boldsymbol{X}_{0:T} \sim \mathbb{P}^0} [\pi_{\mathcal{B}}(\boldsymbol{R}_T)] \tag{3}$$

where $\mathbb{P}^0(\boldsymbol{X}_{0:T})$ is the base path distribution generated from the SDEs (1) or (2) and $Z$ is the normalizing constant. Schrödinger Bridge Matching (SBM) aims to parameterize a **control** or **bias force** $\boldsymbol{b}_\theta$ that tilts the path distribution $\mathbb{P}^{b_\theta}(\boldsymbol{X}_{0:T})$ minimizes the KL-divergence from the bridge path distribution $\mathbb{P}^\star$ given by

$$\boldsymbol{b}^\star = \arg\min_{\boldsymbol{b}_\theta} D_{\text{KL}} \left( \mathbb{P}^{b_\theta} \| \mathbb{P}^\star \right) \quad \text{s.t.} \quad \begin{cases} d\boldsymbol{r}_t^i = \boldsymbol{v}_t^i dt \\ d\boldsymbol{v}_t^i = \frac{-\nabla_{\boldsymbol{r}_t^i} U(\boldsymbol{R}_t) + \boldsymbol{b}_\theta(\boldsymbol{R}_t)}{m_i} dt - \gamma \boldsymbol{v}_t^i dt + \sqrt{\frac{2\gamma k_B \tau}{m_i}} d\boldsymbol{W}_t^i \end{cases} \tag{4}$$

In the case of transition path sampling (TPS) where there is a single initial state $\boldsymbol{R}_{\mathcal{A}}$ and target state $\boldsymbol{R}_{\mathcal{B}}$, we define the target distribution as the relaxed indicator function $\pi_{\mathcal{B}}(\boldsymbol{R}_T) = \mathbf{1}_{\mathcal{B}}(\boldsymbol{R}_T)$ centered around $\boldsymbol{R}_{\mathcal{B}}$.

## 3 LEARNING INTERACTING MULTI-PARTICLE DYNAMICS

The **key challenge** with simulating the dynamics of multi-particle systems lies in the question: *how can we simulate dynamic trajectories from static snapshot data?* The emergence of flow and Schrödinger bridge matching frameworks have effectively approached this problem by defining a **parameterized velocity field** that learns feasible trajectories from data snapshots. However, current strategies remain limited in their ability to model *interacting* multi-particle systems where

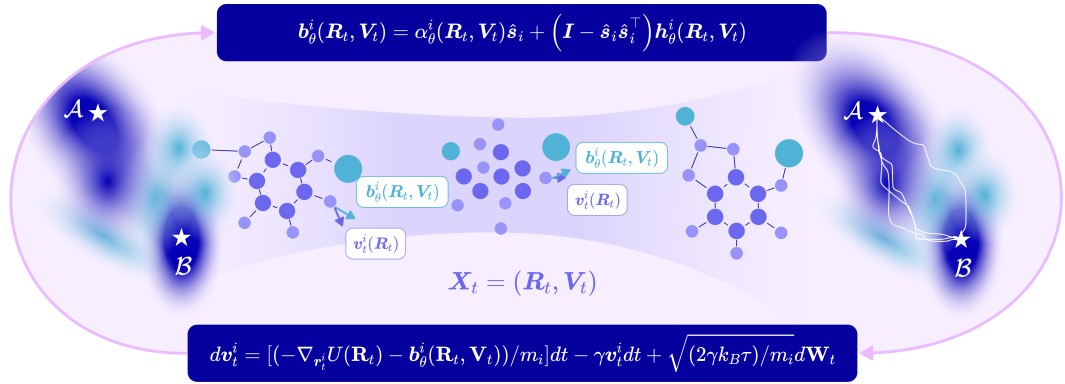

Figure 1: **Entangled Schrödinger Bridge Matching.** We consider the problem of simulating interacting multi-particle systems, where each particle's velocity depends dynamically on the velocities of the other particles in the system, and introduce **EntangledSBM**, a framework that parameterizes an entangled bias force capturing the dynamic interactions between particles.

the velocities carry inherent dependencies that cannot be captured via static snapshots of the system. Prior attempts to capture interactions between particles rely on the *mean-field assumption*, where each particle acts as an average of its surrounding particles, which **does not hold for heterogeneous biomolecular systems**.

To address this challenge, we propose a framework that introduces an additional degree of freedom through a **bias force** that implicitly captures the dependencies between *both* the static positions and dynamic velocities of each particle in the system to control the joint trajectories. We leverage a Transformer architecture and treat each particle as an individual token with features that attend to the features of all other tokens. Crucially, our approach requires no handcrafted features, making it scalable to high-complexity systems.

In Sec 3.1, we formalize this problem as the **Entangled Schrödinger Bridge (EntangledSB)** problem, which aims to determine the optimal set of trajectories to a target state while capturing the dynamic interactions between particles. We then provide a tractable approach to solve the EntangledSB problem with stochastic optimal control (SOC) theory in Sec 3.2. We illustrate the high-level framework of learning **entangled** bias forces in Alg 1 and detail our specific implementation and parameterization in Sec 4.

## 3.1 ENTANGLED SCHRÖDINGER BRIDGE PROBLEM

Here, we formalize the **Entangled Schrödinger Bridge (EntangledSB) problem**, which aims to find a set of optimal bias forces for each particle in the system that depends *dynamically* on the positions and velocities across the entire multi-particle system to guide the Schrödinger bridge trajectory to a target distribution. Specifically, we consider an $n$-particle system $\boldsymbol{X}_t = (\boldsymbol{R}_t, \boldsymbol{V}_t)$ with $\boldsymbol{R}_t = \{\boldsymbol{r}_t^i \in \mathbb{R}^d\}_{i=1}^n$ and $\boldsymbol{V}_t = \{\boldsymbol{v}_t^i \in \mathbb{R}^d\}_{i=1}^n$, where each particle evolves over the time horizon $t \in [0, T]$ via the **bias-controlled** SDE given by

$$d\boldsymbol{r}_t^i = \frac{1}{\gamma}\left(-\nabla_{\boldsymbol{r}_t^i}U(\boldsymbol{R}_t) + \boldsymbol{b}^i(\boldsymbol{R}_t, \boldsymbol{V}_t)\right)dt + \sqrt{\frac{2k_B\tau}{\gamma}}d\boldsymbol{W}_t^i \quad (5)$$

The velocities follow the potential energy landscape defined over the joint coordinates of the system $U(\boldsymbol{R}_t)$ with an *entangled* bias force $\boldsymbol{b}^i(\boldsymbol{R}_t, \boldsymbol{V}_t)$ that depends dynamically on the position and velocity of *all* particles in the system. We can further extend this framework to **second-order** Langevin dynamics via the pair of SDEs

$$d\boldsymbol{r}_t^i = \boldsymbol{v}_t^i dt, \quad d\boldsymbol{v}_t^i = \frac{-\nabla_{\boldsymbol{r}_t^i}U(\boldsymbol{R}_t) + \boldsymbol{b}^i(\boldsymbol{R}_t, \boldsymbol{V}_t)}{m_i}dt - \gamma\boldsymbol{v}_t^i dt + \sqrt{\frac{2\gamma k_B\tau}{m_i}}d\boldsymbol{W}_t^i \quad (6)$$

Formally, we seek the optimal $\boldsymbol{b}^\star(\boldsymbol{R}_t, \boldsymbol{V}_t) := \{\boldsymbol{b}^i(\boldsymbol{R}_t, \boldsymbol{V}_t)\}_{i=1}^n$ that solves the EntangledSB problem defined below.

---

**Algorithm 1** Framework for **Entangled Schrödinger Bridge Matching**

---

1: **Input:** Parameterized velocity network $\boldsymbol{b}_\theta(\boldsymbol{R}_t, \boldsymbol{V}_t) : \mathbb{R}^{n \times d} \times \mathbb{R}^{n \times d} \to \mathbb{R}^{n \times d}$, initial and target joint distributions of the system $\pi_0, \pi_T$, energy function $U(\boldsymbol{R}_t) : \mathbb{R}^{n \times d} \to \mathbb{R}$, objective function in path space $\mathcal{L}(\mathbb{P}^\star, \mathbb{P}^{b_\theta})$ where $\mathbb{P}^\star = \arg\min_{\mathbb{P}^{b_\theta}} \mathcal{L}(\mathbb{P}^\star, \mathbb{P}^{b_\theta})$
2: **while** not converged **do**
3:     Sample initial positions $\boldsymbol{R}_0 \sim \pi_0$
4:     **for** $t = 0$ to $T$ **do**
5:         Evaluate unconditional velocities $\boldsymbol{V}_t(\boldsymbol{R}_t)$
6:         Compute entangled bias force $\boldsymbol{b}_\theta(\boldsymbol{R}_t, \boldsymbol{V}_t)$
7:         Simulate step over a discrete time step $\Delta t$ to get updated coordinates $\boldsymbol{R}_t$
8:     **end for**
9:     Compute reward on terminal states $r(\boldsymbol{X}_T)$
10:    Evaluate path objective $\mathcal{L}(\mathbb{P}^\star, \mathbb{P}^{b_\theta})$ on simulated paths
11:    Update $\theta$ using the gradient $\nabla_\theta \mathcal{L}(\mathbb{P}^\star, \mathbb{P}^{b_\theta})$
12: **end while**

---

**Definition 3.1** (Entangled Schrödinger Bridge Problem). *Given an initial distribution $\pi_\mathcal{A}$, a target distribution $\pi_\mathcal{B}$, the EntangledSB problem aims to determine the set of optimal bias forces $\boldsymbol{b}^\star := \{\boldsymbol{b}^i(\boldsymbol{R}_t, \boldsymbol{V}_t)\}_{i=1}^n$ for each particle $i$ in the system that solves the optimization problem*

$$\boldsymbol{b}^\star(\boldsymbol{R}_t, \boldsymbol{V}_t) = \arg\min_{\boldsymbol{b}_\theta} D_{KL}\left(\mathbb{P}^{b_\theta} \| \mathbb{P}^\star\right) \quad s.t. \quad \begin{cases} \mathbb{P}^\star = \frac{1}{Z}\mathbb{P}^0 \pi_\mathcal{B}(\boldsymbol{X}_T) \\ \mathbb{P}_0^\star = \pi_\mathcal{A}(\boldsymbol{X}_0) \end{cases} \tag{7}$$

*where $\mathbb{P}^0$ is the base path measure with joint dynamics that follow the SDEs in (6).*

## 3.2 Solving EntangledSB with Stochastic Optimal Control

To solve the EntangledSB problem, we leverage stochastic optimal control (SOC) theory where we aim to find the set of optimal control drifts $\boldsymbol{b}^\star(\boldsymbol{R}_t, \boldsymbol{V}_t)$ given a target distribution $\pi_\mathcal{B}$.

**Proposition 3.1** (Solving EntangledSB with Stochastic Optimal Control). *We can solve the EntangledSB problem with the stochastic optimal control (SOC) objective given by*

$$\boldsymbol{b}^\star = \arg\min_{\boldsymbol{b}_\theta} \mathbb{E}_{\boldsymbol{X}_{0:T} \sim \mathbb{P}^{b_\theta}} \left[ \int_0^T \frac{1}{2}\|\boldsymbol{b}_\theta(\boldsymbol{R}_t, \boldsymbol{V}_t)\|^2 dt - r(\boldsymbol{X}_T) \right] \quad s.t. \quad (6) \tag{8}$$

*where $r(\boldsymbol{X}_T) := \log \pi_\mathcal{B}(\boldsymbol{R}_T)$ is the terminal reward that measures the log-probability under the target distribution.*

The proof is provided in App C.1. We highlight that the SOC problem is optimized over *unconstrained* trajectories that need not map to explicit samples from $\pi_\mathcal{B}$ but are iteratively refined to generate trajectories with a terminal distribution that matches $\pi_\mathcal{B}$. In Sec 4.2, we introduce an importance-weighted cross-entropy objective that efficiently solves the SOC problem with theoretical guarantees.

## 4 Entangled Schrödinger Bridge Matching

In this section, we introduce **Entangled Schrödinger Bridge Matching** (EntangledSBM), a novel framework for learning to simulate trajectories of $n$-particle systems that with a **bias force that dynamically depends on the positions and velocities of the other particles** in the system and can generalize to *unseen* target distributions without further training. Our unique parameterization ensures a non-increasing distance toward the target distribution without sacrificing expressivity with hard constraints (Sec 4.1). We introduce a **weighted cross-entropy** objective that enables efficient *off-policy learning* with a replay buffer of simulated trajectories (Sec 4.2).

## 4.1 PARAMETERIZING THE ENTANGLED BIAS FORCE

**Bias Force Parameterization** To ensure that the bias force does not increase the distance from the target distribution $\pi_{\mathcal{B}}$, we ensure that the projection of the predicted bias force onto the direction of the gradient of the target distribution is positive for all particles $i$ in the system

$$\langle \boldsymbol{b}_\theta^i(\boldsymbol{R}_t, \boldsymbol{V}_t), \nabla_{\boldsymbol{r}_t^i} \log \pi_{\mathcal{B}} \rangle \geq 0 \tag{9}$$

Let $\boldsymbol{s}_i = \nabla_{\boldsymbol{r}_t^i} \log \pi_{\mathcal{B}} \in \mathbb{R}^d$ denote the direction towards the target state. To ensure that the bias force is within the cone surrounding $\boldsymbol{s}_i$, we use the following parameterization

$$\boldsymbol{b}_\theta^i(\boldsymbol{R}_t, \boldsymbol{V}_t) = \underbrace{\alpha_\theta^i(\boldsymbol{R}_t, \boldsymbol{V}_t)\hat{\boldsymbol{s}}_i}_{\text{parallel component}} + \underbrace{\left(\boldsymbol{I} - \hat{\boldsymbol{s}}_i\hat{\boldsymbol{s}}_i^\top\right)\boldsymbol{h}_\theta^i(\boldsymbol{R}_t, \boldsymbol{V}_t)}_{\text{orthogonal component}}, \quad \hat{\boldsymbol{s}}_i = \frac{\boldsymbol{s}_i}{\|\boldsymbol{s}_i\|} \tag{10}$$

where $\alpha_\theta^i(\boldsymbol{R}_t, \boldsymbol{V}_t) := \texttt{softplus}(\alpha_\theta^i(\boldsymbol{R}_t, \boldsymbol{V}_t)) \geq 0$ is a scaling factor for the unit vector $\hat{\boldsymbol{s}}_i$ and $\boldsymbol{h}_\theta^i(\boldsymbol{R}_t, \boldsymbol{V}_t) \in \mathbb{R}^d$ is the per-atom correction vector that is projected onto the plane orthogonal to $\hat{\boldsymbol{s}}_i$, both parameterized with neural networks $\theta$. Since the orthogonal component does not affect the dot product with $\hat{\boldsymbol{s}}_i$, the non-negativity constraint $\langle \boldsymbol{b}_\theta^i(\boldsymbol{R}_t, \boldsymbol{V}_t), \nabla_{\boldsymbol{r}_t^i} \log \pi_{\mathcal{B}} \rangle \geq 0$ in (9) is guaranteed by the first term. Intuitively, the orthogonal component enables the bias force to have greater flexibility in moving sideways (e.g., to avoid infeasible regions or add rotational/collective effects) while ensuring that the distance from some target state remains non-increasing.

> **Proposition 4.1** (Non-Increasing Distance from Target Distribution). *For small enough $\Delta t$, the distance from some target state $\boldsymbol{R}_{\mathcal{B}} \in \pi_{\mathcal{B}}$ state after an update with the bias force $\boldsymbol{b}_\theta^i(\boldsymbol{R}_t, \boldsymbol{V}_t)$ defined in (10) is non-increasing, such that*
>
> $$\exists \boldsymbol{R}_{\mathcal{B}} \in \pi_{\mathcal{B}} \ \ s.t. \ \ \|\boldsymbol{R}_{t+\Delta t} - \boldsymbol{R}_{\mathcal{B}}\| \leq \|\boldsymbol{R}_t - \boldsymbol{R}_{\mathcal{B}}\| \tag{11}$$
>
> *where $\boldsymbol{R}_{t+\Delta t} = \boldsymbol{R}_t + (\boldsymbol{b}_\theta^i(\boldsymbol{R}_t, \boldsymbol{V}_t)/m_i)\Delta t$.*

The proof is provided in App C.2. In contrast to previous works that constrain the bias force to point strictly in the direction of a fixed target position, our approach allows greater flexibility in the direction of the orthogonal component.

**Model Architecture** To integrate dependencies on the positions and velocities across $n$ particles, we leverage a Transformer-based architecture where each particle has input features $\texttt{Cat}(\boldsymbol{r}_t^i, \boldsymbol{v}_t^i) \in \mathbb{R}^{2d}$ and the full system features are $\texttt{Cat}(\boldsymbol{R}_t, \boldsymbol{V}_t) \in \mathbb{R}^{n \times 2d}$. We further input the direction $\boldsymbol{R}_{\mathcal{B}} - \boldsymbol{R}_t$ and distance $\|\boldsymbol{R}_{\mathcal{B}} - \boldsymbol{R}_t\|$ from the target distribution, which enables generalization to unseen target distributions at inference by learning the dependence of the bias force on the target direction. The Transformer encoder enables efficient and expressive propagation of feature information across all pairs of particles in the system to generate context-aware embeddings for bias force parameterization. For MD systems, we ensure invariance of the coordinate frame using the Kabsch algorithm (Kabsch, 1976), which aligns the position $\boldsymbol{R}_t$ with the target position $\boldsymbol{R}_{\mathcal{B}}$ before input into the model. Further details are provided in App D and E.

## 4.2 OFF-POLICY LEARNING WITH WEIGHTED CROSS-ENTROPY

**Log-Variance Objective** To train the bias force to match the optimal $\boldsymbol{b}^\star$, we can adapt the *log-variance* (LV) divergence (Seong et al., 2025; Nüsken & Richter, 2021) defined as

$$\mathcal{L}_{\text{LV}}(\mathbb{P}^\star, \mathbb{P}^{b_\theta}) = \text{Var}_{\mathbb{P}^v}\left[\log \frac{\mathrm{d}\mathbb{P}^\star}{\mathrm{d}\mathbb{P}^{b_\theta}}\right] = \mathbb{E}_{\mathbb{P}^v}\left[\left(\log \frac{\mathrm{d}\mathbb{P}^\star}{\mathrm{d}\mathbb{P}^{b_\theta}} - \mathbb{E}_{\mathbb{P}^v}\left[\log \frac{\mathrm{d}\mathbb{P}^\star}{\mathrm{d}\mathbb{P}^{b_\theta}}\right]\right)^2\right] \tag{12}$$

where $\boldsymbol{v}$ is an arbitrary control that enables *off-policy learning* from trajectories that need not be generated by the current bias force $\boldsymbol{b}_\theta$. While the optimal solution to the LV objective is exactly the optimal bias force $\boldsymbol{b}^\star$, the non-convex nature of the LV objective with respect to the path measure $\mathbb{P}^{b_\theta}$ inhibits convergence. We provide more details in App B.

**Cross-Entropy Objective** To achieve a more theoretically-favorable optimization problem, we propose a cross-entropy objective that is *convex* with respect to the biased path measure $\mathbb{P}^{b_\theta}$ given by

$$\mathcal{L}_{\text{CE}}(\mathbb{P}^\star, \mathbb{P}^{b_\theta}) = \mathbb{E}_{\mathbb{P}^\star}\left[-\log \mathbb{P}^{b_\theta}\right] := D_{\text{KL}}(\mathbb{P}^\star \| \mathbb{P}^{b_\theta}) = \mathbb{E}_{\mathbb{P}^\star}\left[\log \frac{d\mathbb{P}^\star}{d\mathbb{P}^{b_\theta}}\right] = \mathbb{E}_{\mathbb{P}^v}\left[\frac{d\mathbb{P}^\star}{d\mathbb{P}^v}\log\frac{d\mathbb{P}^\star}{d\mathbb{P}^{b_\theta}}\right] \quad (13)$$

where $\mathbb{P}^\star$ is the target bridge measure and $\mathbb{P}^v$ is the path measure generated with an arbitrary control $v$. To avoid taking the expectation with respect to the unknown path measure $\mathbb{P}^\star$, we define an importance weight $w^\star(\boldsymbol{X}_{0:T}) := \frac{d\mathbb{P}^\star}{d\mathbb{P}^v}(\boldsymbol{X}_{0:T})$ independent of $b_\theta$ that enables optimization using trajectories generated from an arbitrary control $v$. We note that similar CE objectives have been adopted in earlier work for different applications (Kappen & Ruiz, 2016; Zhu et al., 2025; Tang et al., 2025b), which we discuss in App A.

> **Proposition 4.2** (Convexity and Uniqueness of Cross-Entropy Objective). *The cross-entropy objective $\mathcal{L}_{CE}$ is convex in $\mathbb{P}^{b_\theta}$ and there exists a unique minimizer $b^\star$ that is the solution to the EntangledSOC problem in Proposition 3.1.*

The proof is given in App C.3. To amortize the cost of simulation and reinforce high-reward trajectories, we define the arbitrary control as the biased path measure from the previous iteration $v := \bar{b} = \text{stopgrad}(b_\theta)$, which allows us to reuse the trajectories over multiple training steps by maintaining a replay buffer $\mathcal{R}$ that contains the trajectories $\boldsymbol{X}_{0:T}$ and their importance weights.

> **Proposition 4.3** (Equivalence of Variational and Path Integral Objectives). *The cross-entropy objective can be expressed in path-integral form as*
>
> $$\mathcal{L}_{CE}(\theta) = \mathbb{E}_{\mathbb{P}^v}\left[w^\star(\boldsymbol{X}_{0:T})\mathcal{F}_{\boldsymbol{b}_\theta, \boldsymbol{v}}(\boldsymbol{X}_{0:T})\right] \quad (14)$$
>
> $$\begin{cases} w^\star(\boldsymbol{X}_{0:T}) = \frac{d\mathbb{P}^\star}{d\mathbb{P}^v}(\boldsymbol{X}_{0:T}) = \frac{e^{r(\boldsymbol{X}_T)}}{Z}\frac{d\mathbb{P}^0}{d\mathbb{P}^v}(\boldsymbol{X}_{0:T}) \\ \mathcal{F}_{\boldsymbol{b}_\theta, \boldsymbol{v}}(\boldsymbol{X}_{0:T}) = \frac{1}{2}\int_0^T \|\boldsymbol{b}_\theta(\boldsymbol{R}_t, \boldsymbol{V}_t)\|^2 - \int_0^T (\boldsymbol{b}_\theta^\top \boldsymbol{v})(\boldsymbol{R}_t, \boldsymbol{V}_t)dt - \int_0^T \boldsymbol{b}_\theta(\boldsymbol{R}_t, \boldsymbol{V}_t)^\top d\boldsymbol{W}_t \end{cases} \quad (15)$$
>
> *where we define the reference measure $\boldsymbol{v} = \bar{\boldsymbol{b}} := \text{stopgrad}(\boldsymbol{b}_\theta)$ is the off-policy control drift from the previous iteration.*

The proof is given in App C.4. Since the normalizing constant $Z$ is intractable in practice, we compute the importance weight as $w^\star(\boldsymbol{X}_{0:T}) := \text{softmax}_{\mathcal{B}}(r(\boldsymbol{X}_T) + \log p^0(\boldsymbol{X}_{0:T}) - \log p^{\bar{b}}(\boldsymbol{X}_{0:T}))$, which is a batch estimate of $\frac{d\mathbb{P}^\star}{d\mathbb{P}^{\bar{b}}}$. When a batch contains only a single trajectory (i.e., when the system is large), we additionally store the value of $\log r(\boldsymbol{X}_T)$ sample from the importance weighted distribution over the replay buffer as $\boldsymbol{X}_{0:T} \sim \text{Cat}(\text{softmax}_{\mathcal{R}}(w^\star(\boldsymbol{X}_{0:T})))$.

**Discrete Time Objective** Since we want to train on off-policy trajectories from previous iterations to dynamically update the learned bias force given the velocities of the remaining particles, we require storing the simulated trajectories in a discretized form $\boldsymbol{X}_{0:K} := (\boldsymbol{X}_k)_{k \in \{0,...,K\}}$ with step size $\Delta t$. To compute the term $\mathcal{F}_{\boldsymbol{b}_\theta, \bar{\boldsymbol{b}}}$ in the CE loss, we discretize it as

$$\hat{\mathcal{F}}_{\boldsymbol{b}_\theta, \bar{\boldsymbol{b}}}(\boldsymbol{X}_{0:K}) = \frac{1}{2}\sum_{k=0}^{K-1}\|\boldsymbol{b}_\theta(\boldsymbol{X}_k)\|^2\Delta t - \sum_{k=0}^{K-1}(\boldsymbol{b}_\theta \cdot \bar{\boldsymbol{b}})(\boldsymbol{X}_k)\Delta t - \sum_{k=0}^{K-1}\boldsymbol{b}_\theta(\boldsymbol{X}_k)\cdot\Delta\boldsymbol{W}_k \quad (16)$$

where $\Delta\boldsymbol{W}_k = \boldsymbol{\Sigma}^{-1}[\boldsymbol{R}_{k+1} - \boldsymbol{R}_k - (\boldsymbol{f}(\boldsymbol{X}_k) + \boldsymbol{\Sigma}\bar{\boldsymbol{b}}(\boldsymbol{R}_k, \boldsymbol{V}_k))\Delta t]$. Now, we can establish a simplified version of the cross-entropy objective in Prop 4.3.

> **Proposition 4.4** (Discretized Cross-Entropy). *Given the discretized $\hat{\mathcal{F}}_{\boldsymbol{b}_\theta, \bar{\boldsymbol{b}}}(\boldsymbol{X}_{0:K})$ in (16), we can derive a simplified loss function as*
>
> $$\hat{\mathcal{L}}_{CE}(\theta) = \mathbb{E}_{\boldsymbol{X}_{0:K} \sim \mathbb{P}^{\bar{b}}}\left[\underbrace{\frac{d\mathbb{P}^\star}{d\mathbb{P}^{\bar{b}}}(\boldsymbol{X}_{0:K})}_{w^\star(\boldsymbol{X}_{0:K})}\log\left(\frac{p^0(\boldsymbol{X}_{0:K})\exp(r(\boldsymbol{X}_K))}{p^{b_\theta}(\boldsymbol{X}_{0:K})}\right)\right] \quad (17)$$
>
> *where $w^\star(\boldsymbol{X}_{0:K})$ is the importance weight of the discrete time trajectory $\boldsymbol{X}_{0:K}$.*

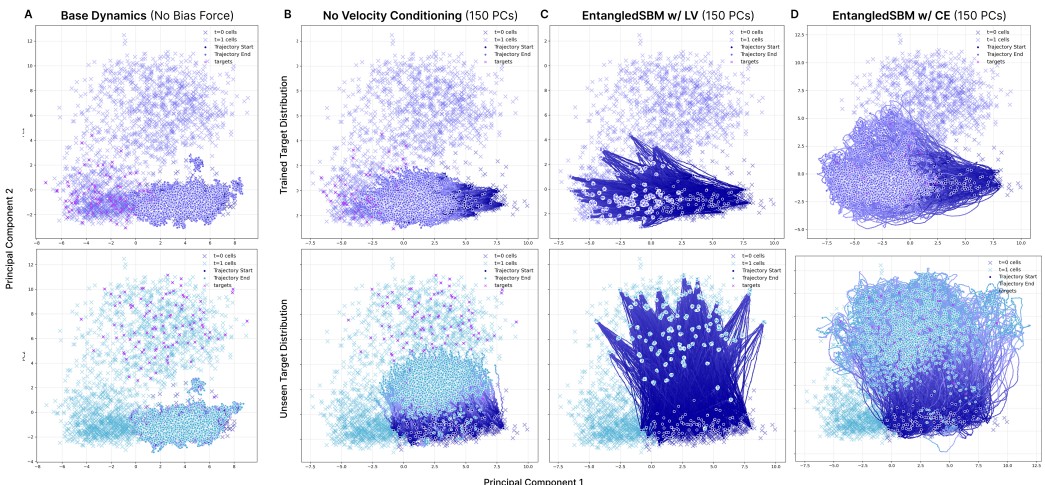

Figure 2: **Simulated cell cluster dynamics with EntangledSBM under Clonidine perturbation with 150 PCs.** Nearest neighbour cell clusters with $n = 16$ cells are simulated over 100 time steps to perturbed cells seen during training (**Top**) and an unseen perturbed population (**Bottom**). The gradient indicates the evolution of timesteps from the initial time $t = 0$ (navy) to the final time $t = T$ (purple or turquoise). **(A)** Trajectories under base dynamics with no bias force. **(B)** Trajectories simulated with base and bias forces with **(B)** no velocity conditioning, **(C)** log-variance (LV) objective, and **(D)** cross-entropy (CE) objective.

The proof is provided in App C.5. This allows us to train by sampling trajectories, tracking their log-probabilities under the biased and base path measure, computing the terminal reward $r(\boldsymbol{X}_K)$, storing the values in the replay buffer $\mathcal{R}$, and reusing the trajectories over $N_{\text{epochs}}$ training iterations.

## 5 EXPERIMENTS

We evaluate **EntangledSBM** on several trajectory simulation tasks involving $n$-particle systems with interacting dynamics, including simulating cell cluster dynamics under drug perturbation (Sec 5.1) and transition path sampling (TPS) for molecular dynamics (MD) simulation of proteins (Sec 5.2).

### 5.1 SIMULATING INTERACTING CELL DYNAMICS UNDER PERTURBATION

Populations of cells undergo dynamic signaling interactions that cause shifts in cell state. Under a drug perturbation, these interactions determine the perturbed state of the cell population, which is of significant importance in drug discovery and screening. In this experiment, we use **EntangledSBM** to parameterize an *entangled* bias force that learns **dynamic interactions between cells**, guiding the trajectories of a batch of cells to the perturbed state. We demonstrate that EntangledSBM can not only accurately reconstruct the perturbed cell states within the training distribution following trajectories on the data manifold, but can also generate paths to *unseen* target distributions that diverge from the training distribution, demonstrating its potential to **scale to diverse perturbations and cell types with sparse data**.

**Setup and Baselines**  To model cell state trajectories with high resolution, we evaluate two perturbations (Clonidine and Trametinib at $5\mu L$) from the Tahoe-100M dataset (Zhang et al., 2025a), each containing data for a total of 60K genes. We select the top 2000 highly variable genes (HVGs) and perform principal component analysis (PCA), to maximally capture the variance in the data via the top principal components (38% in the top-50 PCs). To evaluate the ability of EntangledSBM to simulate trajectories to *unseen* target cell clusters that diverge in its distribution from the training target distribution, we further cluster the perturbed cell data to construct multiple disjoint perturbed cell populations. For Clonidine, we generated two clusters, and for Trametinib, we generated three clusters (Figure 5). For each experiment, we trained on a single cluster, and the remaining clusters were left for evaluation.

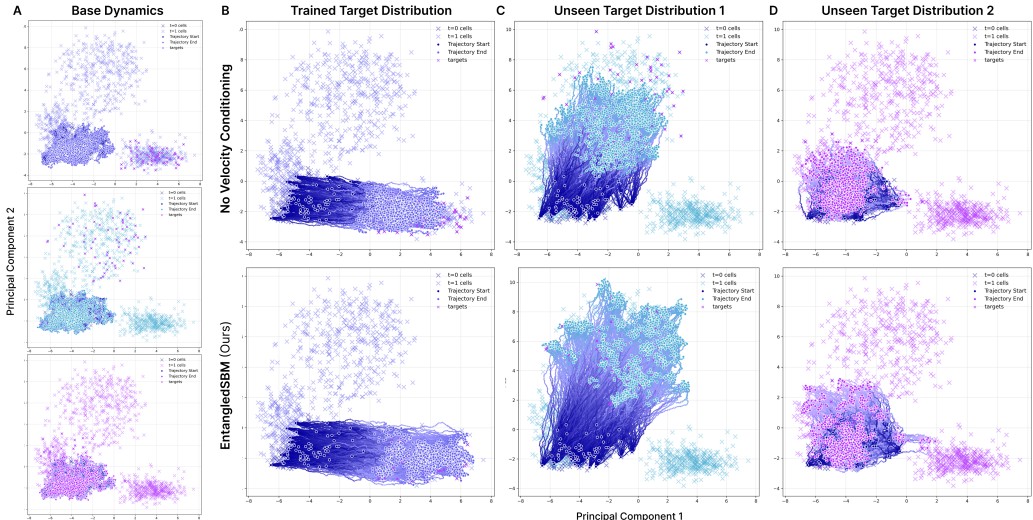

Figure 3: **Comparison of EntangledSBM with and without velocity conditioning for cell cluster simulation under Trametinib perturbation.** 50 PCs are simulated with the learned bias force trained with the CE objective without velocity conditioning (**Top**) and with velocity conditioning (**Bottom**) to (**B**) the perturbed population used for training and (**C**, **D**) the two unseen target populations.

We trained the bias force given batches of cell clusters with $n = 16$ cells, where the initial cluster is sampled from the unperturbed cell data $\pi_{\mathcal{A}}$, and the target cluster is sampled from one of the perturbed cell populations $\pi_{\mathcal{B}}$. Each batch of cells $\boldsymbol{X}_t = (\boldsymbol{R}_t, \boldsymbol{V}_t)$ has positions $\boldsymbol{R}_t \in \mathbb{R}^{n \times d}$ and velocities $\boldsymbol{V}_t \in \mathbb{R}^{n \times d}$, where $d$ is the dimension of the principal components (PCs) that we simulate. We define the energy landscape such that regions of high data density have low energy and the base dynamics $\mathbb{P}^0$ follow the gradient toward high data density as detailed in App D.1. We evaluate the reconstruction accuracy of the perturbed distribution at $t = T$ using the maximum mean discrepancy (**RBF-MMD**) for all $d$ PCs and Wasserstein distances ($\mathcal{W}_1$ and $\mathcal{W}_2$) of the top two PCs between ground truth and reconstructed clusters after 100 simulation steps with details in D.2. To demonstrate the significance of entangled velocity conditioning on performance, we trained our model with only the positions as input $\boldsymbol{b}_\theta(\boldsymbol{R}_t)$. We additionally compare our cross-entropy objective $\mathcal{L}_{\text{CE}}$ from Sec 4.2 with the log-variance divergence $\mathcal{L}_{\text{LV}}$ in App F. Additional experiment details are provided in App D.

**Clonidine Perturbation Results**   We demonstrate that EntangledSBM accurately guides the base dynamics, which largely remain in the initial data-dense state, to the target perturbed state across increasing PC dimensions $d = \{50, 100, 150\}$ and reconstructs the target distribution (Fig. 2; Table 1). Notably, we show that our parameterization of the bias force enables generalization to **unseen perturbed populations** that *diverge* from the training distribution by learning dependencies on the target state (Fig. 2). Comparing the reconstruction metrics of the target perturbed cell states with and without velocity conditioning, we confirm our hypothesis that introducing dependence on the dynamic velocities across a cell cluster enables more accurate reconstruction of the target distribution compared to when the bias term is trained with only positional dependence (Table 1). Furthermore, we observe that the LV objective $\mathcal{L}_{\text{LV}}$ generates nearly straight abrupt trajectories to the target state while the CE objective $\mathcal{L}_{\text{CE}}$ generates smooth paths along the data manifold (Table 7; Fig. 7), demonstrating its superiority as an **unconstrained objective** for learning controls for optimizing path distributions as further discussed in App F.

**Trametinib Perturbation Results**   We further evaluate our method to predict trajectories of cell clusters under perturbation with Trametinib, which induces three distinct perturbed cell distributions (Figure 5B). Despite training on only one of the perturbed distributions, we demonstrate that EntangledSBM trained with $\mathcal{L}_{\text{CE}}$ is capable of accurately reconstructing the remaining cell distributions *without* additional training, demonstrating significantly improved performance compared to the bias force trained without velocity conditioning (Table 1; Fig. 3). Furthermore, we observe the same

Table 1: **Results for simulating cell cluster dynamics under Clonidine and Trametinib perturbation with EntangledSBM.** Best values for each PC dimension are **bolded**. We report RBF-MMD for all PCs and $\mathcal{W}_1$ and $\mathcal{W}_2$ distances of the top 2 PCs between ground truth and reconstructed clusters after 100 simulation steps and cluster size set to $n = 16$. Mean and standard deviation of metrics from 5 independent simulations are reported. We simulate PCs $d = \{50, 100, 150\}$ for Clonidine and $d = 50$ for Trametinib to cells sampled from the training target distribution and unseen target distributions, and compare against the learned bias force with no velocity conditioning.

| | Clonidine Perturbation | | | | | |
| --- | --- | --- | --- | --- | --- | --- |
| | Trained Target Distribution | | | Unseen Target Distribution | | |
| **Model** | RBF-MMD ($\downarrow$) | $\mathcal{W}_1$ ($\downarrow$) | $\mathcal{W}_2$ ($\downarrow$) | RBF-MMD ($\downarrow$) | $\mathcal{W}_1$ ($\downarrow$) | $\mathcal{W}_2$ ($\downarrow$) |
| **Base Dynamics** (50 PCs) | $0.677_{\pm 0.001}$ | $5.947_{\pm 0.005}$ | $6.015_{\pm 0.005}$ | $0.784_{\pm 0.001}$ | $8.217_{\pm 0.005}$ | $8.384_{\pm 0.005}$ |
| **EntangledSBM w/o Velocity Conditioning** | | | | | | |
| 50 PCs | $0.440_{\pm 0.000}$ | $1.741_{\pm 0.003}$ | $1.857_{\pm 0.004}$ | $0.478_{\pm 0.000}$ | $2.907_{\pm 0.006}$ | $3.022_{\pm 0.006}$ |
| 100 PCs | $0.494_{\pm 0.000}$ | $2.315_{\pm 0.004}$ | $2.423_{\pm 0.004}$ | $0.539_{\pm 0.000}$ | $4.110_{\pm 0.004}$ | $4.249_{\pm 0.003}$ |
| 150 PCs | $0.510_{\pm 0.000}$ | $2.497_{\pm 0.006}$ | $2.620_{\pm 0.006}$ | $0.560_{\pm 0.000}$ | $4.573_{\pm 0.006}$ | $4.716_{\pm 0.007}$ |
| **EntangledSBM w/ CE** | | | | | | |
| 50 PCs | $\mathbf{0.401}_{\pm 0.000}$ | $\mathbf{0.342}_{\pm 0.002}$ | $\mathbf{0.400}_{\pm 0.001}$ | $\mathbf{0.419}_{\pm 0.000}$ | $\mathbf{0.538}_{\pm 0.013}$ | $\mathbf{0.705}_{\pm 0.030}$ |
| 100 PCs | $\mathbf{0.455}_{\pm 0.000}$ | $\mathbf{0.953}_{\pm 0.025}$ | $\mathbf{1.015}_{\pm 0.025}$ | $\mathbf{0.500}_{\pm 0.001}$ | $\mathbf{0.899}_{\pm 0.006}$ | $\mathbf{1.055}_{\pm 0.008}$ |
| 150 PCs | $\mathbf{0.478}_{\pm 0.000}$ | $\mathbf{0.753}_{\pm 0.008}$ | $\mathbf{0.826}_{\pm 0.007}$ | $\mathbf{0.506}_{\pm 0.000}$ | $\mathbf{0.700}_{\pm 0.009}$ | $\mathbf{0.811}_{\pm 0.011}$ |

| | Trametinib Perturbation | | | | | | | | |
| --- | --- | --- | --- | --- | --- | --- | --- | --- | --- |
| | Trained Target Distribution | | | Unseen Target Distribution 1 | | | Unseen Target Distribution 2 | | |
| **Method** | RBF-MMD ($\downarrow$) | $\mathcal{W}_1$ ($\downarrow$) | $\mathcal{W}_2$ ($\downarrow$) | RBF-MMD ($\downarrow$) | $\mathcal{W}_1$ ($\downarrow$) | $\mathcal{W}_2$ ($\downarrow$) | RBF-MMD ($\downarrow$) | $\mathcal{W}_1$ ($\downarrow$) | $\mathcal{W}_2$ ($\downarrow$) |
| **Base Dynamics** | $0.938_{\pm 0.001}$ | $7.637_{\pm 0.005}$ | $7.653_{\pm 0.006}$ | $0.900_{\pm 0.000}$ | $7.766_{\pm 0.009}$ | $7.877_{\pm 0.009}$ | $0.754_{\pm 0.001}$ | $1.201_{\pm 0.009}$ | $1.455_{\pm 0.012}$ |
| **EntangledSBM w/o Velocity Conditioning** | $0.449_{\pm 0.000}$ | $1.506_{\pm 0.005}$ | $1.544_{\pm 0.005}$ | $0.476_{\pm 0.000}$ | $2.116_{\pm 0.005}$ | $2.197_{\pm 0.005}$ | $0.480_{\pm 0.000}$ | $0.505_{\pm 0.004}$ | $0.627_{\pm 0.005}$ |
| **EntangledSBM w/ CE** | $\mathbf{0.428}_{\pm 0.000}$ | $\mathbf{0.392}_{\pm 0.005}$ | $\mathbf{0.434}_{\pm 0.006}$ | $\mathbf{0.409}_{\pm 0.000}$ | $\mathbf{0.453}_{\pm 0.008}$ | $\mathbf{0.561}_{\pm 0.009}$ | $\mathbf{0.451}_{\pm 0.000}$ | $\mathbf{0.394}_{\pm 0.003}$ | $\mathbf{0.469}_{\pm 0.004}$ |

phenomena observed for Clonidine when training with the LV objective, which results in trajectories that fail to capture the intermediate cell dynamics (Table 8; Fig. 8).

## 5.2 Transition Path Sampling of Protein Folding Dynamics

Simulating rare transitions that occur over long timescales and between metastable states on molecular dynamics (MD) landscapes remains a significant challenge due to the high feature dimensionality of biomolecules. In this experiment, we aim to simulate feasible transition paths across high-energy barriers at an *all-atom* resolution, a task that is challenging for both traditional MD and ML-based methods. We demonstrate that EntangledSBM generates feasible transition paths with higher target accuracy against a range of baselines.

**Setup and Baselines** We follow the setup in Seong et al. (2025) and simulate the position and velocity using OpenMM (Eastman et al., 2017) with an annealed temperature schedule. To evaluate the performance of EntangledSBM, we consider the RMSD of the Kabsch-aligned coordinates averaged across 64 paths (**RMSD**; Å) (Kabsch, 1976), percentage of simulated trajectories that hit the target state (**THP**; %), and the highest energy transition state along the biased trajectories averaged across the trajectories that hit the target (**ETS**; kJ/mol). For baselines, we compare against unbiased MD (**UMD**) with temperatures of 3600K for alanine dipeptide and 1200K for the fast-folding proteins, steered MD (**SMD**; Schlitter et al. (1994); Izrailev et al. (1999)) with temperatures of 10K and 20K, path integral path sampling (**PIPS**; Holdijk et al. (2023)), transition path sampling with diffusion path samplers (**TPS-DPS**; Seong et al. (2025)) with the highest performing *scaled* parameterization. Additional experiment details are provided in App E.

**Alanine Dipeptide** First, we consider Alanine Dipeptide with two alanine residues and 22 atoms. We aim to simulate trajectories to the target state $\pi_{\mathcal{B}} = \{\mathbf{R} \mid \|\xi(\mathbf{R}) - \xi(\mathbf{R}_{\mathcal{B}})\| < 0.75\}$ defined by the backbone dihedral angles $\xi(\mathbf{R}) = (\phi, \psi)$. We show that EntangledSBM generates feasible trajectories through both saddle points, representing the two reaction channels (Figure 4A), achieving superior target hit potential (THP) and lower root mean squared error (RMSD) from the target state than all baselines (Table 2).

**Fast-Folding Proteins** We evaluate EntangledSBM on the more challenging task of modeling the all-atom transition paths of four fast-folding proteins, including **Chignolin** (10 amino acids; 138

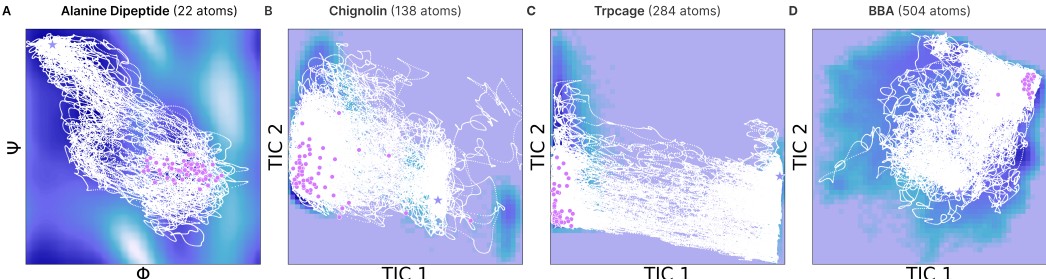

Figure 4: **Transition paths generated with EntangledSBM.** The energy landscape is colored from dark navy (low potential energy) to light purple (high potential energy) and plotted with the backbone dihedral angles $(\phi, \psi)$ for Alanine Dipeptide and the top two TICA components for the fast-folding proteins. The starting state $\boldsymbol{R}_0$ and target states $\boldsymbol{R}_\mathcal{B}$ are indicated with purple stars, and the final states of the simulated trajectories $\boldsymbol{R}_T$ are indicated with pink circles.

Table 2: **Transition path sampling benchmarks with EntangledSBM.** Best values are **bolded**. All metrics are averaged over 64 paths. Unless specified in brackets, paths are generated at 300K for Alanine Dipeptide and Chignolin and 400K for the others. Hyperparameters and evaluation metrics are detailed in App E. † denotes values taken from Seong et al. (2025).

| Protein | Alanine Dipeptide | | | Chignolin | | |
|---|---|---|---|---|---|---|
| Method | RMSD (↓) | THP (↑) | ETS (↓) | Method | RMSD (↓) | THP (↑) | ETS (↓) |
| UMD † | $1.19_{\pm0.32}$ | 6.25 | — | UMD † | $7.23_{\pm0.93}$ | 1.56 | 388.17 |
| SMD (10K) † | $0.86_{\pm0.21}$ | 7.81 | $812.47_{\pm148.80}$ | SMD (10K) † | $1.26_{\pm0.31}$ | 6.25 | $-527.95_{\pm93.58}$ |
| SMD (20K) † | $0.56_{\pm0.27}$ | 54.69 | $78.40_{\pm12.76}$ | SMD (20K) † | $\mathbf{0.85}_{\pm0.24}$ | 34.38 | $179.52_{\pm138.87}$ |
| PIPS (Force) † | $0.66_{\pm0.15}$ | 43.75 | $28.17_{\pm10.86}$ | PIPS (Force) † | $4.66_{\pm0.17}$ | 0.00 | — |
| TPS-DPS (Scalar) † | $0.25_{\pm0.20}$ | 76.00 | $\mathbf{22.79}_{\pm13.57}$ | TPS-DPS (Scalar) † | $1.17_{\pm0.66}$ | 59.38 | $-780.18_{\pm216.93}$ |
| **EntangledSBM** (Ours) | $\mathbf{0.18}_{\pm0.07}$ | **92.19** | $47.91_{\pm22.76}$ | **EntangledSBM** (Ours) | $0.92_{\pm0.13}$ | **64.06** | $2825.61_{\pm318.94}$ |

| Protein | Trp-cage | | | BBA | | |
|---|---|---|---|---|---|---|
| Method | RMSD (↓) | THP (↑) | ETS (↓) | Method | RMSD (↓) | THP (↑) | ETS (↓) |
| UMD † | $8.27_{\pm1.13}$ | 0.00 | — | UMD † | $10.81_{\pm1.05}$ | 0.00 | - |
| SMD (10K) † | $1.68_{\pm0.23}$ | 3.12 | $-312.54_{\pm20.67}$ | SMD (10K) † | $2.89_{\pm0.32}$ | 0.00 | - |
| SMD (20K) † | $1.20_{\pm0.20}$ | 42.19 | $-226.40_{\pm85.59}$ | SMD (20K) † | $1.66_{\pm0.30}$ | 26.56 | $-3104.95_{\pm97.57}$ |
| PIPS (Force) † | $7.47_{\pm0.19}$ | 0.00 | - | PIPS (Force) † | $9.84_{\pm0.18}$ | 0.00 | - |
| TPS-DPS (Scalar) † | $\mathbf{0.76}_{\pm0.12}$ | 81.25 | $-317.61_{\pm140.89}$ | TPS-DPS (Scalar) † | $1.21_{\pm0.09}$ | 84.38 | $-3801.68_{\pm139.38}$ |
| **EntangledSBM** (Ours) | $1.04_{\pm0.22}$ | **82.81** | $765.74_{\pm155.28}$ | **EntangledSBM** (Ours) | $\mathbf{0.84}_{\pm0.08}$ | **96.88** | $1453.80_{\pm367.84}$ |

atoms), **Trp-cage** (284 atoms; 20 amino acids), and **BBA** (504 atoms; 28 amino acids). We define a hit as a trajectory where the final state reaches the metastable distribution $\pi_\mathcal{B} = \{\boldsymbol{R} \mid \|\xi(\boldsymbol{R}) - \xi(\boldsymbol{R}_\mathcal{B})\| < 0.75\}$, where $\xi(\boldsymbol{R})$ are the top two time-lagged independent component analysis (TICA; Pérez-Hernández et al. (2013)) components. As shown in Fig. 4, EntangledSBM generates *diverse* transition paths across high-energy barriers that successfully reach the target state. We achieve a superior target hit percentage than all baselines across all proteins and a lower or comparable RMSD (Table 2). While we observe a higher average energy of transition state (ETS) compared to baselines, this can be attributed to the larger proportion and greater diversity of successful target-hitting paths. Given that the base dynamics $\mathbb{P}^0$ rarely move beyond the initial metastable state, we show that EntangledSBM effectively learns the target bridge dynamics $\mathbb{P}^\star$ despite high energy barriers.

## 6 CONCLUSION

In this work, we present **Entangled Schrödinger Bridge Matching (EntangledSBM)**, a principled framework for learning the second-order dynamics of interacting multi-particle systems through entangled bias forces and an unconstrained cross-entropy objective. EntangledSBM captures dependencies between particle positions and velocities, enabling the modeling of complex dynamics across biological scales. For perturbation modeling, EntangledSBM reconstructs perturbed cell states while generalizing to divergent target states not seen during training, and for molecular dynamics (MD), it generates physically plausible transition paths for fast-folding proteins at an all-atom resolution.

## REPRODUCIBILITY STATEMENT

We provide complete details to enable reproduction of our results. The theoretical problem formulation and framework are described in Sections 3–4, with formal definitions and proofs in App C. Experiment details for cell perturbation modeling and transition path sampling are provided in App D and E. Evaluation metrics are listed in App D.2 and E.2. Tables 1, 4, 2 and 5 summarize the benchmarks and results. Algorithms 2 and 3 provide pseudocode for training and inference. Code is provided in the anonymous repository: https://anonymous.4open.science/r/EntangledSBManon. Together, these resources ensure full reproducibility.

## ETHICS STATEMENT

This work introduces a computational framework for modeling interacting multi-particle dynamics with applications to protein folding and cell state perturbations. The cell perturbation experiments (Section 5.1) use publicly available datasets from Tahoe-100M and related sources, cited with attribution and used under their original licenses. All biomolecular simulations (Section 5.2) were performed on standard test proteins (Alanine Dipeptide, Chignolin, Trp-cage) using publicly available force fields and simulation protocols, with no human or animal subjects involved. No proprietary or sensitive data were employed. The work does not provide experimental protocols or actionable therapeutic targets, and is restricted to methodological contributions in generative modeling. While such frameworks may be broadly useful for molecular or cellular design, the immediate scope of EntangledSBM is fundamental methodology, and the dual-use risk is minimal. We also report hardware, training steps, and hyperparameters (App D and E, Table 4 and 5) to encourage efficient reproduction and transparent assessment of computational cost.

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

## OUTLINE OF APPENDIX

In App A, we provide an overview and discussion of related works in off-policy learning, Schrödinger bridge matching, and transition path sampling. App B provides the theoretical basis of stochastic optimal control (SOC) theory used in our work and a theoretical motivation for our entangled bias force. App C provides the theoretical formulations and proofs for EntangledSBM. We provide details on the experimental setup and training hyperparameters for the cell perturbation experiment in App D and the transition path sampling (TPS) experiment in App E. We further compare the log-variance (LV) divergence and cross-entropy (CE) objectives in App F. Finally, we provide the pseudocode for the training and inference of EntangledSBM in App G.

**Notation**  In this work, we consider an $n$-particle system $\boldsymbol{X}_t = (\boldsymbol{R}_t, \boldsymbol{V}_t)$ where $\boldsymbol{R}_t \in \mathbb{R}^{n \times d}$ and $\boldsymbol{V}_t \in \mathbb{R}^{n \times d}$ denote the positions and velocities of the full system that evolves over the time horizon $t \in [0, T]$. The notation $\boldsymbol{X}$ indicates a random variable and $\boldsymbol{x}$ denotes a deterministic realization. Each particle $i$ is defined by its position $\boldsymbol{r}_t^i \in \mathbb{R}^d$ and velocity $\boldsymbol{v}_t^i \in \mathbb{R}^d$ which lie in $d$-dimensional Euclidean space. The base dynamics evolve according to a potential energy function denoted $U(\boldsymbol{R}_t) : \mathbb{R}^{n \times d} \to \mathbb{R}$ and the biased dynamics evolve with an additional parameterized bias force $\boldsymbol{b}_\theta(\boldsymbol{R}_t, \boldsymbol{V}_t) : \mathbb{R}^{n \times d} \times \mathbb{R}^{n \times d} \to \mathbb{R}^{n \times d}$ that captures the implicit dependencies within the system and controls the base dynamics towards a target distribution $\pi_{\mathcal{B}}(\boldsymbol{R}_T)$ from a state in the initial distribution $\boldsymbol{R}_{\mathcal{A}} \in \pi_{\mathcal{A}}(\boldsymbol{R}_0)$. The corresponding path measures over trajectories $\boldsymbol{X}_{0:T} := (\boldsymbol{X}_t)_{t \in [0,T]}$ are denoted $\mathbb{P}^0$ for the base process and $\mathbb{P}^{b_\theta}$ for the biased process. The optimal path measure that solves the EntangledSB problem is denoted $\mathbb{P}^\star \equiv \mathbb{P}^{b^\star}$ with bias force $\boldsymbol{b}^\star(\boldsymbol{R}_t, \boldsymbol{V}_t)$.

## A  RELATED WORKS

**Off-Policy Learning**  Importance-weighted cross-entropy (CE) objectives have previously been used to sample the target distribution $g^\star \propto h f_u$ by aligning a proposal distribution $f_v$ to the intractable target with $\mathrm{KL}(g^\star \| f_v)$ (Kappen & Ruiz, 2016). The main difference is that our approach aims to match a **controlled path measure** $\mathbb{P}^{b_\theta}$ with an energy-minimizing path measure $\mathbb{P}^\star$ rather than a static target distribution. Furthermore, we adapt our path CE objective to apply for discrete-time paths, similar to the approach taken by Seong et al. (2025), which discretizes the log-variance (LV) divergence (Nüsken & Richter, 2021). While the LV objective has theoretical optimality guarantees, it is non-convex in $\mathbb{P}^{b_\theta}$. Since the CE objective is convex in $\mathbb{P}^{b_\theta}$, it yields an ideal optimization landscape for $\theta$ and demonstrates improved performance as shown in Sec 5.1. We also highlight that our optimization scheme is a form of *off-policy* learning, as we minimize the objective with respect to the path distribution generated from an auxiliary control $\boldsymbol{v}$ which we set to the non-gradient-tracking bias force $\boldsymbol{v} := \bar{\boldsymbol{b}} = \mathtt{stopgrad}(\boldsymbol{b}_\theta)$. This objective is inspired by recent work that leverages off-policy reinforcement learning in the discrete state space for discrete diffusion sampling (Zhu et al., 2025) and fine-tuning (Tang et al., 2025b).

**Transition Path Sampling**  To overcome the challenge of generating feasible transition paths across high energy barriers for transition path sampling, several non-ML and ML-based approaches have been introduced (Bolhuis et al., 2002; Dellago et al., 1998; Vanden-Eijnden et al., 2010). Non-ML approaches often rely on collective variables (CVs), which reduce the dimensionality of the molecular conformation to a function of coordinates that are known to be involved in transition states (Hooft et al., 2021). These include steered MD (Schlitter et al., 1994; Izrailev et al., 1999), umbrella sampling (Torrie & Valleau, 1977; Kästner, 2011), meta-dynamics (Laio & Parrinello, 2002; Ensing et al., 2006; Branduardi et al., 2012; Bussi & Branduardi, 2015), adaptive biasing force (Comer et al., 2015), and on-the-fly probability-enhanced sampling (Invernizzi & Parrinello, 2020). To overcome the lack of well-understood CVs for large biomolecular systems, ML-based methods have been explored for determining or constructing CVs (Sultan & Pande, 2018; Rogal et al., 2019; Chen & Ferguson, 2018; Sun et al., 2022; Sipka et al., 2023b), but modeling transition paths at all-atom resolution is still desirable due to systems of increased complexity, and CVs are uncertain.

Several ML-based approaches for simulating transition paths have been developed (Singh & Limmer, 2023; Yan et al., 2022; Sipka et al., 2023a; Das et al., 2021). Notably, Path Integral Path Sampling (PIPS; Holdijk et al. (2023)) learns a bias force with stochastic optimal control theory, Doob's Lagrangian Du et al. (2024) defines an optimal transition path with a Lagrangian objective that solves

the Doob's $h$-transform, Transition Path Sampling with Diffusion Path Samplers (TPS-DPS; Seong et al. (2025)) trains a bias force with an off-policy log-variance objective, and Action Minimization with Onsager-Machlup (OM) Functional (Raja et al., 2025) generates a discrete interpolation between endpoints by minimizing the OM functional. Furthermore, Blessing et al. (2025) introduces a trust-region-constrained SOC optimization algorithm, which is applied to TPS. While these approaches demonstrate the promise of ML-based TPS, their applicability remains limited to small-scale biomolecular systems (Alanine Dipeptide, tetrapeptides, etc.) for all-atom simulation (Holdijk et al., 2023; Du et al., 2024) or require coarse-grained representations when scaling up to larger fast-folding proteins (Raja et al., 2025). We directly compare our method against Seong et al. (2025) and Blessing et al. (2025), and achieve state-of-the-art performance on all-atom simulations of larger, fast-folding proteins, and Blessing et al. (2025). Furthermore, we highlight that our framework extends beyond molecular dynamics (MD) simulations, with applications to single-cell simulation and generalization to unseen target distributions.

**Modeling Cell Dynamics Under Perturbation** Predicting the dynamics of heterogeneous cell populations under perturbations such as treatment with a drug candidate, gene editing, and knockouts, or protein expression, has critical applications in drug discovery and predictive medicine (Shalem et al., 2014; Kramme et al., 2021; Dixit et al., 2016; Gavriilidis et al., 2024; Zhang et al., 2025a; Kobayashi et al., 2022; Smela et al., 2023a; Pierson Smela et al., 2025; Yeo et al., 2021). Generative modeling frameworks including flow matching (Zhang et al., 2025b; Rohbeck et al., 2025; Atanackovic et al., 2024; Tong et al., 2024; Wang et al., 2025), Schrödinger Bridge Matching (Alatkar & Wang, 2025; Tong et al., 2024; Kapuśniak et al., 2024; Tang et al., 2025a), and stochastic optimal control (Zhang et al., 2025c;d) have enabled significant advances in parameterizing velocities that evolve dynamically over time on non-linear energy manifolds by learning on static snapshots across the trajectory. However, these models often require *explicit training on a specific target distribution* and fail to generalize to unseen target distributions at inference, especially if they diverge from those seen in the training data, which limits their scalability to cell types and perturbations that are not seen during training. Furthermore, many of these approaches simulate cells as populations of particles that evolve independently from each other, which fails to capture the diverse intercellular signaling and interactions that occur naturally and under perturbations. Although there have been some early developments in learning these interactions from data (Zhang et al., 2025d), they operate under the mean-field assumption, where each cell takes the average effect of all surrounding cells rather than simulating individual pairwise interactions. This assumption does not account for the complex interactions between cells and limits its applicability to heterogeneous cell populations.

## B EXTENDED THEORETICAL PRELIMINARIES

In this section, we provide relevant theoretical background in stochastic optimal control (SOC) theory for path measures. We use the terms *bias force* and *optimal control* interchangeably, denoted with $\boldsymbol{b}$, which is less common in general SOC literature where $\boldsymbol{u}$ is often used. For a more comprehensive analysis of SOC theory, we refer the reader to Nüsken & Richter (2021).

### B.1 STOCHASTIC OPTIMAL CONTROL

**Controlled Path Measures** First, we consider the controlled path measure $\mathbb{P}^b$ defined by the SDE:

$$d\boldsymbol{X}_t = [\boldsymbol{f}(\boldsymbol{X}_t, t) + \boldsymbol{\Sigma}\boldsymbol{b}(\boldsymbol{X}_t, t)]dt + \boldsymbol{\Sigma}d\boldsymbol{W}_t \tag{18}$$

where $\boldsymbol{b} : [0, T] \times \mathcal{X} \to \mathcal{X} \in \mathcal{U}$ is known as the **control**[1] that tilts the path measure from the unconditional dynamics $\mathbb{P}^0$. Now, we will define **Radon-Nikodym derivative (RND)** of the path measures corresponding to the controlled and unconditional SDEs, which is crucial for defining our objective.

---

[1] $\mathcal{U}$ is the set of admissible controls continuously differentiable in $(\boldsymbol{x}, t)$ with bounded linear growth in $\boldsymbol{x}$

**Lemma B.1** (Radon-Nikodym Derivative). *Given a pair of SDEs with and without a control drift $b(X_t, t)$ on $t \in [0, T]$ with the same diffusion $\Sigma$ defined as*

$$\mathbb{P}^0 : dX_t = f(X_t, t)dt + \Sigma dW_t, \qquad\qquad X_0 = x_0 \qquad (19)$$

$$\mathbb{P}^b : dX_t^b = \left[ f(X_t, t) + \Sigma b(X_t, t) \right] dt + \Sigma dW_t, \qquad X_0^b = x_0 \qquad (20)$$

*Then, the Radon-Nikodym derivative satisfies*

$$\log \frac{d\mathbb{P}^b}{d\mathbb{P}^0}(X_{0:T}) = \int_0^T (b^\top \Sigma^{-1})(X_t, t)dX_t - \int_0^T (\Sigma^{-1} f \cdot b)(X_t, t)dt - \frac{1}{2} \int_0^T \|b(X_t, t)\|^2 dt$$

*or equivalently:*

$$\log \frac{d\mathbb{P}^b}{d\mathbb{P}^0}(X_{0:T}) = \int_0^T b(X_t, t)^\top dW_t - \frac{1}{2} \int_0^T \|b(X_t, t)\|^2 dt \qquad (21)$$

*Proof.* First, we use Girsanov's theorem to get the Radon-Nikodym derivative with respect to the zero-drift reference path measure $\mathbb{Q}$ defined by the SDE $dX_t = \Sigma(X_t, t)dW_t$, we have

$$\log \frac{\mathbb{P}^0}{\mathbb{Q}}(X_{0:T}) = \int_0^T \left( f(X_t, t) \cdot \Sigma^{-2}(X_t, t) \right) dX_t - \frac{1}{2} \int_0^T \left( f \cdot \Sigma^{-2} f \right)(X_t, t)dt \qquad (22)$$

and for the controlled path measure $\mathbb{P}^b$, we have

$$\log \frac{\mathbb{P}^b}{\mathbb{Q}}(X_{0:T}) = \int_0^T (f + \Sigma b)(X_t, t) \cdot \Sigma^{-2}(X_t, t)dX_t$$

$$- \frac{1}{2} \int_0^T \left( (f + \Sigma b) \cdot \Sigma^{-2}(f + \Sigma b) \right)(X_t, t)dt \qquad (23)$$

Then, using the identity

$$\log \frac{d\mathbb{P}^b}{d\mathbb{P}^0} = \log \frac{d\mathbb{P}^b}{d\mathbb{Q}} \frac{d\mathbb{Q}}{d\mathbb{P}^0} = \log \frac{d\mathbb{P}^b}{d\mathbb{Q}} + \log \frac{d\mathbb{Q}}{d\mathbb{P}^0} = \log \frac{d\mathbb{P}^b}{d\mathbb{Q}} - \log \frac{d\mathbb{P}^0}{d\mathbb{Q}} \qquad (24)$$

Substituting in (22) and (23) and canceling terms, we get

$$\log \frac{d\mathbb{P}^b}{d\mathbb{P}^0}(X_{0:T}) = \int_0^T (b^\top \Sigma^{-1})(X_t, t)dX_t - \int_0^T (\Sigma^{-1} f \cdot b)(X_t, t)dt - \frac{1}{2} \int_0^T \|b(X_t, t)\|^2 dt \quad (25)$$

Equivalently, since we can write $dW_t = \Sigma^{-1}(X_t, t)(dX_t + b(X_t, t)dt)$, we can write

$$\log \frac{d\mathbb{P}^b}{d\mathbb{P}^0}(X_{0:T}) = \int_0^T b(X_t, t)^\top dW_t - \frac{1}{2} \int_0^T \|b(X_t, t)\|^2 dt \qquad (26)$$

which concludes the proof. $\qquad\qquad\square$

**Remark B.1.** *In the underdamped Langevin model, the diffusion acts only on the velocity coordinates, so the full diffusion matrix on $X_t = (R_t, V_t)$ is degenerate, i.e., $\Sigma = \begin{bmatrix} 0 & 0 \\ 0 & \Sigma_V \end{bmatrix}$. We therefore apply Girsanov's theorem on the velocity SDE only, where the diffusion $\Sigma_V$ is invertible, and treat positions as deterministic integrals of velocity (i.e., $R_t = R_0 + \int_0^t V_s ds$). Therefore, Girsanov's theorem is first defined on velocity paths as*

$$\log \frac{d\mathbb{P}_V^b}{d\mathbb{P}_V^0}(X_{0:T}) = \int_0^T b(X_t, t)^\top dW_t - \frac{1}{2} \int_0^T \|b(X_t, t)\|^2 dt \qquad (27)$$

*and pushed forward to the full path measure via $(R_t, V_t) = (R_0 + \int_0^t V_s ds, V_t)$. For simplicity in notation, we define $\Sigma^{-1} \equiv \Sigma_V^{-1}$ in the theoretical proofs.*

**Stochastic Optimal Control**   The reward-guided **stochastic optimal control (SOC)** problem aims to determine the **optimal control** $\boldsymbol{b}^\star = \min_{\boldsymbol{b} \in \mathcal{U}} J(\boldsymbol{b}; \boldsymbol{x}, t)$ that minimizes the **cost functional** $J(\boldsymbol{b}_\theta; \boldsymbol{x}, t)$ with reward $r : \mathcal{X} \to \mathbb{R}$ given by

$$J(\boldsymbol{b}; \boldsymbol{x}, t) = \mathbb{E}_{p_t}\left[\int_t^T \left(\boldsymbol{f}(\boldsymbol{X}_s, s) + \frac{1}{2}\|\boldsymbol{b}(\boldsymbol{X}_s, s)\|^2\right) ds - r(\boldsymbol{X}_T)\Big| \boldsymbol{X}_t = \boldsymbol{x}\right] \qquad (28)$$

which measures the *cost-to-go* from the current state $\boldsymbol{X}_t = \boldsymbol{x}$ at time $t$ under the controlled dynamics defined in (18) to the terminal state $\boldsymbol{X}_T$. The infimum of the cost functional is defined as

$$J(\boldsymbol{b}^\star; \boldsymbol{x}, t) = \inf_{\boldsymbol{b} \in \mathcal{U}} J(\boldsymbol{b}; \boldsymbol{x}, t) =: V_t(\boldsymbol{x}) \qquad (29)$$

which is also referred to as the *value function* $V_t(\boldsymbol{x})$. To derive the path integral representation of the optimal control $\boldsymbol{b}^\star$, we introduce the *work functional* which computes the running cost subtracted by the terminal reward of a state path $\boldsymbol{X}_{0:T}$,

$$\mathcal{W}(\boldsymbol{X}_{t:T}, t) := \int_t^T \boldsymbol{f}(\boldsymbol{X}_s, s)ds - r(\boldsymbol{X}_T) \qquad (30)$$

We can now define a series of statements that connect the optimal drift $\boldsymbol{b}^\star$ and the value function $V_t$.

**Theorem B.1.** *Let $V_t$ be the value function and $\boldsymbol{b}^\star$ be the optimal control. Then, the following are true:*

(a) *The optimal control satisfies $\boldsymbol{b}^\star(\boldsymbol{x}) = -\boldsymbol{\Sigma}^\top \nabla_{\boldsymbol{x}} V_t(\boldsymbol{x})$*

(b) *The Radon-Nikodym derivative of the optimal path measure $\mathbb{P}^\star$ can be defined by taking state paths from the unconditional path measure $\mathbb{P}^0$ and reweighting them with the output of the work functional $\mathcal{W}(\boldsymbol{X}_{0:T}, 0)$.*

$$\frac{d\mathbb{P}^\star}{d\mathbb{P}^0}(\boldsymbol{X}_{0:T}) = \frac{1}{Z}e^{-\mathcal{W}(\boldsymbol{X}_{0:T}, 0)}, \quad Z = \mathbb{E}_{\boldsymbol{X}_{0:T} \sim \mathbb{P}^0}\left[e^{-\mathcal{W}(\boldsymbol{X}_{0:T}, 0)}\right] \qquad (31)$$

(c) *For any $(\boldsymbol{x}, t) \in \mathcal{X} \times [0, T]$, the value function can be written in path integral form by taking the expectation over the uncontrolled path measure $\mathbb{P}^0$ as*

$$V_t(\boldsymbol{x}, t) = -\log \mathbb{E}\left[e^{-\mathcal{W}(\boldsymbol{X}_{t:T}, t)}\big|\boldsymbol{X}_t = \boldsymbol{x}\right] \qquad (32)$$

(d) *Combining (a) and (c), we can write the optimal control as*

$$\boldsymbol{b}^\star(\boldsymbol{x}) = \boldsymbol{\Sigma}^\top \nabla_{\boldsymbol{x}} \log \mathbb{E}_{\boldsymbol{X}_{t:T} \sim \mathbb{P}^0}\left[e^{-\mathcal{W}(\boldsymbol{X}_{t:T}, t)}\big|\boldsymbol{X}_t = \boldsymbol{x}\right] \qquad (33)$$

While Theorem B.1 provides a closed-form solution to the optimal control $\boldsymbol{b}^\star$, it remains impractical to compute as $\mathbb{P}^0$ contains an infinite number of paths from any point $\boldsymbol{x}$. This problem motivates parameterizing $\boldsymbol{b}^\star$ with a neural network $\boldsymbol{b}_\theta$. The natural objective for obtaining an accurate approximation of $\boldsymbol{b}^\star$ that induces the path measure $\mathbb{P}^\star$ is the KL-divergence $\mathcal{L}(\mathbb{P}^\star, \mathbb{P}^{b_\theta}) := D_{\mathrm{KL}}(\mathbb{P}^{b_\theta}\|\mathbb{P}^\star)$ as its minimizer is exactly $\mathbb{P}^\star$.

However, taking the gradient $\nabla_\theta D_{\mathrm{KL}}(\mathbb{P}^{b_\theta}\|\mathbb{P}^\star)$ requires differentiating through the full stochastic trajectories $\boldsymbol{X}_{0:T} \sim \mathbb{P}^{b_\theta}$ generated by the Euler-Maruyama SDE solver due to the expectation over $\mathbb{P}^{b_\theta}$, resulting in a significant computational bottleneck. To overcome this bottleneck, previous work (Nüsken & Richter, 2021; Seong et al., 2025) leverages alternative path-measure objectives that do not involve expectations over $\mathbb{P}^{b_\theta}$ while preserving optimality guarantees. We focus on two objectives: the **log-variance divergence** used in Seong et al. (2025) and the **cross-entropy objective**, which we formalize in the present work for solving the EntangledSB problem.

**Log-Variance Objective**   Here, we describe the **log-variance (LV) divergence** which is introduced in Nüsken & Richter (2021) and applied to transition path sampling (TPS) in TPS-DPS (Seong et al., 2025). Given the path measure $\mathbb{P}^{b_\theta}$ generated with the parameterized bias force $\boldsymbol{b}_\theta$ and the target path measure $\mathbb{P}^\star$, the log-variance divergence $\mathcal{L}_{\mathrm{LV}}$ is defined as

$$\mathcal{L}_{\mathrm{LV}}(\mathbb{P}^\star, \mathbb{P}^{b_\theta}) := \mathrm{Var}_{\mathbb{P}^v}\left[\log \frac{d\mathbb{P}^\star}{d\mathbb{P}^{b_\theta}}\right] = \mathbb{E}_{\mathbb{P}^v}\left[\left(\log \frac{d\mathbb{P}^\star}{d\mathbb{P}^{b_\theta}} - \mathbb{E}_{\mathbb{P}^v}\left[\log \frac{d\mathbb{P}^\star}{d\mathbb{P}^{b_\theta}}\right]\right)^2\right] \qquad (34)$$

where $\mathbb{P}^v$ can be defined as any arbitrary path measure generated from a control $v$. To allow simulated trajectories to be used across multiple training iterations, it is common to define $v$ as the bias force with frozen parameters $v = \bar{b} := \texttt{stopgrad}(b_\theta)$. Using Lemma B.1, we define $\mathcal{L}_{\mathrm{LV}}(\mathbb{P}^\star, \mathbb{P}^{b_\theta}) = \mathbb{E}_{\mathbb{P}^v}[(\mathcal{F}_{b_\theta,v} - \mathbb{E}_{\mathbb{P}^v}[\mathcal{F}_{b_\theta,v}])^2]$, where $\mathcal{F}_{b_\theta,v}$ is defined as

$$\mathcal{F}_{b_\theta,v} := \log \frac{d\mathbb{P}^\star}{d\mathbb{P}^{b_\theta}} = \log \frac{d\mathbb{P}^\star}{d\mathbb{P}^0} \frac{d\mathbb{P}^0}{d\mathbb{P}^{b_\theta}}$$

$$= r(\boldsymbol{X}_T) + \frac{1}{2} \int_0^T \|\boldsymbol{b}_\theta(\boldsymbol{X}_t)\|^2 dt - \int_0^T (\boldsymbol{b}_\theta^\top \boldsymbol{v})(\boldsymbol{X}_t) dt - \int_0^T \boldsymbol{b}_\theta^\top d\boldsymbol{W}_t \tag{35}$$

using similar justification as in App C.4. Since the expectation $\mathbb{E}_{\mathbb{P}^v}[\mathcal{F}_{b_\theta,v}]$ is computationally intractable, Seong et al. (2025) introduces a scalar parameter $w$ that is jointly optimized with $\theta$ such that $\arg\min_w \mathcal{L}_{\mathrm{LV}}(\theta, w) = \mathbb{E}_{\mathbb{P}^v}[\mathcal{F}_{b_\theta,v}]$. Substituting in $w$ for $\mathbb{E}_{\mathbb{P}^v}[\mathcal{F}_{b_\theta,v}]$ and taking the discretization over $K$ steps, the LV objective becomes

$$\mathcal{L}_{\mathrm{LV}}(\theta, w) = \mathbb{E}_{\boldsymbol{X}_{0:K} \sim \mathbb{P}^v}\left[(\mathcal{F}_{b_\theta,v}(\boldsymbol{X}_{0:K}) - w)^2\right] = \mathbb{E}_{\boldsymbol{x}_{0:K} \sim \mathbb{P}^v}\left[\left(\log \frac{p^0(\boldsymbol{x}_{0:K})\exp(r(\boldsymbol{X}_K))}{p^{b_\theta}(\boldsymbol{x}_{0:K})} - w\right)^2\right]$$

where $v = \bar{b} := \texttt{stopgrad}(b_\theta)$.

## B.2 JUSTIFICATION FOR ENTANGLED BIAS FORCES

**Proposition B.1** (Monotone Optimality of Entangled Bias Forces). *Let $\mathcal{F}_{ind}$ be the hypothesis class of bias forces that depend only on particle positions $\boldsymbol{b}_\theta^i := \boldsymbol{b}_\theta^i(\boldsymbol{R}_t)$ and $\mathcal{F}_{ent}$ be the hypothesis class of entangled bias forces $\boldsymbol{b}_\theta^i := \boldsymbol{b}_\theta^i(\boldsymbol{R}_t, \boldsymbol{V}_t)$. Then, under the cross-entropy objective,*

$$\inf_{\boldsymbol{b}_\theta \in \mathcal{F}_{ent}} \mathcal{L}_{CE}(\boldsymbol{b}_\theta) \leq \inf_{\boldsymbol{b}_\theta \in \mathcal{F}_{ind}} \mathcal{L}_{CE}(\boldsymbol{b}_\theta) \tag{36}$$

*with strict improvement when the optimal control is non-factorizable.*

*Proof.* Let $\mathbb{P}^\star$ denote the optimal path measure that solves the EntangledSB problem from (3.1) and $\mathbb{P}^{b_\theta}$ be the path measure induced by the bias force $\boldsymbol{b}_\theta$. Let $\mathcal{F}_{\mathrm{ind}}$ be the hypothesis class of bias forces independent of the full system velocities $\boldsymbol{b}_\theta^i := \boldsymbol{b}_\theta^i(\boldsymbol{R}_t)$ and $\mathcal{F}_{\mathrm{ent}}$ be the hypothesis class of entangled bias forces $\boldsymbol{b}_\theta^i := \boldsymbol{b}_\theta^i(\boldsymbol{R}_t, \boldsymbol{V}_t)$. Clearly, we have $\mathcal{F}_{\mathrm{ind}} \subseteq \mathcal{F}_{\mathrm{ent}}$. It follows that

$$\{\mathbb{P}^{b_\theta} : \boldsymbol{b}_\theta \in \mathcal{F}_{\mathrm{ind}}\} \subseteq \{\mathbb{P}^{b_\theta} : \boldsymbol{b}_\theta \in \mathcal{F}_{\mathrm{ent}}\} \tag{37}$$

Taking the infima over the KL-divergence functional $D_{\mathrm{KL}}(\mathbb{P}^\star\|\cdot) \geq 0$ over a larger set of functions cannot increase the value.

$$\inf_{\boldsymbol{b}_\theta \in \mathcal{F}_{\mathrm{ent}}} D_{\mathrm{KL}}(\mathbb{P}^\star\|\mathbb{P}^{b_\theta}) \leq \inf_{\boldsymbol{b}_\theta \in \mathcal{F}_{\mathrm{ind}}} D_{\mathrm{KL}}(\mathbb{P}^\star\|\mathbb{P}^{b_\theta}) \tag{38}$$

Furthermore, if $\mathbb{P}^\star \notin \overline{\{\mathbb{P}^{b_\theta} : \boldsymbol{b}_\theta \in \mathcal{F}_{\mathrm{ind}}\}}$ which denotes the closure of the set of path measures induced by $\mathcal{F}_{\mathrm{ind}}$ and there exists $\boldsymbol{b}^\star \in \overline{\mathcal{F}_{\mathrm{ent}}}$ where $\mathbb{P}^{\boldsymbol{b}^\star} = \mathbb{P}^\star$, then we have

$$\inf_{\boldsymbol{b}_\theta \in \mathcal{F}_{\mathrm{ent}}} D_{\mathrm{KL}}(\mathbb{P}^\star\|\mathbb{P}^{b_\theta}) = 0 \quad \text{while} \quad \inf_{\boldsymbol{b}_\theta \in \mathcal{F}_{\mathrm{ind}}} D_{\mathrm{KL}}(\mathbb{P}^\star\|\mathbb{P}^{b_\theta}) > 0 \tag{39}$$

yielding strict improvement. By Proposition 4.2, we have that $\mathbb{P}^{\boldsymbol{b}^\star} = \mathbb{P}^\star$ is the unique minimizer of $D_{\mathrm{KL}}(\mathbb{P}^\star\|\cdot)$. $\qquad\square$

# C THEORETICAL PROOFS

For notational simplicity, we will drop the explicit dependence on $(\boldsymbol{R}_t, \boldsymbol{V}_t)$ and simply denote the stochastic path of the system as $(\boldsymbol{X}_t)_{t \in [0,T]}$.

## C.1   PROOF OF PROPOSITION 3.1

**Lemma C.1.** *The bias force $\boldsymbol{b}^\star$ that minimizes the SOC objective $J(\boldsymbol{b}_\theta; \boldsymbol{x}, t)$ generates the path measure $\mathbb{P}^b$ that minimizes the KL-divergence with the optimal tilted path measure $\mathbb{P}^\star$ that satisfies*

$$\mathbb{P}^\star = \frac{1}{Z}\mathbb{P}^0 e^{r(\boldsymbol{X}_T)}, \quad Z = \mathbb{E}_{\mathbb{P}^0_T}\left[e^{r(\boldsymbol{X}_T)}\right] \tag{40}$$

*where $\mathbb{P}^0$ is the base path measure.*

*Proof.* Let the dynamics of $\boldsymbol{X}_t = (\boldsymbol{R}_t, \boldsymbol{V}_t)$ under the base path measure $\mathbb{P}^0$ be defined as

$$d\boldsymbol{X}_t = \boldsymbol{f}(\boldsymbol{X}_t)dt + \boldsymbol{\Sigma} d\boldsymbol{W}_t \tag{41}$$

and the dynamics under the biased path measure $\mathbb{P}^{b_\theta}$ be defined as

$$d\boldsymbol{X}_t = \left[\boldsymbol{f}(\boldsymbol{X}_t) + \boldsymbol{\Sigma}\boldsymbol{b}_\theta(\boldsymbol{X}_t)\right]dt + \boldsymbol{\Sigma} d\boldsymbol{W}_t \tag{42}$$

By Lemma B.1, we have that the logarithm of the Radon-Nikodym derivative of the biased measure $\mathbb{P}^{b_\theta}$ with respect to the base measure $\mathbb{P}^0$ is given by

$$\log\frac{d\mathbb{P}^{b_\theta}}{d\mathbb{P}^0}(\boldsymbol{X}_{0:T}) = \int_0^T \boldsymbol{b}_\theta(\boldsymbol{X}_t)^\top d\boldsymbol{W}_t - \frac{1}{2}\int_0^T \|\boldsymbol{b}_\theta(\boldsymbol{X}_t)\|^2 dt \tag{43}$$

Now, taking the expectation with respect to $\mathbb{P}^{b_\theta}$, we derive the expression for the KL-divergence

$$D_{\mathrm{KL}}(\mathbb{P}^{b_\theta}\|\mathbb{P}^0) = \mathbb{E}_{\mathbb{P}^{b_\theta}}\left[\log\frac{d\mathbb{P}^{b_\theta}}{d\mathbb{P}^0}\right]$$

$$= \mathbb{E}_{\mathbb{P}^{b_\theta}}\left[\int_0^T \boldsymbol{b}_\theta(\boldsymbol{X}_t)^\top d\boldsymbol{W}_t - \frac{1}{2}\int_0^T \|\boldsymbol{b}_\theta(\boldsymbol{X}_t)\|^2 dt\right]$$

$$= \mathbb{E}_{\mathbb{P}^{b_\theta}}\left[\int_0^T \boldsymbol{b}_\theta(\boldsymbol{X}_t)^\top (d\boldsymbol{W}_t^b + \boldsymbol{b}_\theta(\boldsymbol{X}_t)dt) - \frac{1}{2}\int_0^T \|\boldsymbol{b}_\theta(\boldsymbol{X}_t)\|^2 dt\right]$$

$$= \underbrace{\mathbb{E}_{\mathbb{P}^{b_\theta}}\left[\int_0^T \boldsymbol{b}_\theta(\boldsymbol{X}_t)^\top d\boldsymbol{W}_t^b\right]}_{=0} + \mathbb{E}_{\mathbb{P}^{b_\theta}}\left[\int_0^T \|\boldsymbol{b}_\theta(\boldsymbol{X}_t)\|^2 dt - \frac{1}{2}\int_0^T \|\boldsymbol{b}_\theta(\boldsymbol{X}_t)\|^2 dt\right]$$

$$= \frac{1}{2}\mathbb{E}_{\mathbb{P}^{b_\theta}}\int_0^T \|\boldsymbol{b}_\theta(\boldsymbol{X}_t)\|^2 dt \tag{44}$$

Substituting this into the SOC objective in (3.1), we get

$$J(\boldsymbol{b}_\theta) = \mathbb{E}_{\mathbb{P}^{b_\theta}}\left[\int_0^T \frac{1}{2}\|\boldsymbol{b}_\theta(\boldsymbol{X}_t)\|^2 dt - r(\boldsymbol{X}_T)\right]$$

$$= D_{\mathrm{KL}}(\mathbb{P}^{b_\theta}\|\mathbb{P}^0) - \mathbb{E}_{\mathbb{P}^{b_\theta}}[r(\boldsymbol{X}_T)] \tag{45}$$

By adding $\log Z$ on both sides, we do not change the minimizer of $J(\boldsymbol{b}_\theta)$ and the SOC objective becomes the KL-divergence between the controlled path measure $\mathbb{P}^{b_\theta}$ and the optimal path measure $\mathbb{P}^\star$:

$$J(\boldsymbol{b}_\theta) + \log Z = D_{\mathrm{KL}}(\mathbb{P}^{b_\theta}\|\mathbb{P}^0) - \mathbb{E}_{\mathbb{P}^{b_\theta}}[r(\boldsymbol{X}_T)] + \log Z$$

$$= \underbrace{\mathbb{E}_{\mathbb{P}^{b_\theta}}\left[\log\frac{d\mathbb{P}^{b_\theta}}{d\mathbb{P}^0} - r(\boldsymbol{X}_T) + \log Z\right]}_{D_{\mathrm{KL}}(\mathbb{P}^{b_\theta}\|\mathbb{P}^\star)} \tag{46}$$

This shows that the minimizer $\boldsymbol{b}^\star$ to the SOC objective also generates the path measure $\mathbb{P}^{b^\star}$ that is closest in KL-divergence to the optimal path measure $\mathbb{P}^\star$. Therefore, the optimal control solution satisfies

$$\log\frac{d\mathbb{P}^\star}{d\mathbb{P}^0}(\boldsymbol{X}_{0:T}) = r(\boldsymbol{X}_T) - \log Z \iff \mathbb{P}^\star = \frac{1}{Z}\mathbb{P}^0 e^{r(\boldsymbol{X}_T)} \tag{47}$$

which concludes our proof. □

**Proposition 3.1** (Solving EntangledSB with Stochastic Optimal Control). *We can solve the EntangledSB problem with the stochastic optimal control (SOC) objective given by*

$$\boldsymbol{b}^\star = \arg\min_{\boldsymbol{b}_\theta} \mathbb{E}_{\boldsymbol{X}_{0:T} \sim \mathbb{P}^{b_\theta}} \left[ \int_0^T \frac{1}{2} \|\boldsymbol{b}_\theta(\boldsymbol{R}_t, \boldsymbol{V}_t)\|^2 dt - r(\boldsymbol{X}_T) \right] \quad s.t. \quad (6) \tag{8}$$

*where $r(\boldsymbol{X}_T) := \log \pi_{\mathcal{B}}(\boldsymbol{R}_T)$ is the terminal reward that measures the log-probability under the target distribution.*

We aim to recover the optimal path measure $\mathbb{P}^\star$ that minimizes the KL divergence from the base dynamics $\mathbb{P}^0$ while matching the terminal distribution $\pi_{\mathcal{B}}$ defined as

$$\mathbb{P}^\star(\boldsymbol{X}_{0:T}) = \frac{1}{Z} \mathbb{P}^0 \pi_{\mathcal{B}}(\boldsymbol{X}_T) \tag{48}$$

From Lemma C.1, we have that the bias force $\boldsymbol{b}_\theta$ that minimizes the SOC objective defined as

$$\boldsymbol{b}^\star = \arg\min_{\boldsymbol{b}_\theta} \mathbb{E}_{\boldsymbol{X}_{0:T} \sim \mathbb{P}^{b_\theta}} \left[ \int_0^T \frac{1}{2} \|\boldsymbol{b}_\theta(\boldsymbol{X}_t)\|^2 dt - r(\boldsymbol{X}_T) \right] \tag{49}$$

also generates the path measure $\mathbb{P}^{b^\star}$ that minimizes the KL-divergence $D_{\mathrm{KL}}(\mathbb{P}^{b_\theta} \| \mathbb{P}^\star)$ with the optimal path measure $\mathbb{P}^\star$ defined as

$$\mathbb{P}^\star = \frac{1}{Z} \mathbb{P}^0 e^{r(\boldsymbol{X}_T)} \tag{50}$$

Therefore, setting $r(\boldsymbol{X}_T) := \log \pi_{\mathcal{B}}(\boldsymbol{X}_T)$ recovers the solution to the EntangledSB problem. □

## C.2 PROOF OF PROPOSITION 4.1

**Proposition 4.1** (Non-Increasing Distance from Target Distribution). *For small enough $\Delta t$, the distance from some target state $\boldsymbol{R}_{\mathcal{B}} \in \pi_{\mathcal{B}}$ state after an update with the bias force $\boldsymbol{b}_\theta^i(\boldsymbol{R}_t, \boldsymbol{V}_t)$ defined in (10) is non-increasing, such that*

$$\exists \boldsymbol{R}_{\mathcal{B}} \in \pi_{\mathcal{B}} \ \ s.t. \ \ \|\boldsymbol{R}_{t+\Delta t} - \boldsymbol{R}_{\mathcal{B}}\| \leq \|\boldsymbol{R}_t - \boldsymbol{R}_{\mathcal{B}}\| \tag{11}$$

*where $\boldsymbol{R}_{t+\Delta t} = \boldsymbol{R}_t + (\boldsymbol{b}_\theta^i(\boldsymbol{R}_t, \boldsymbol{V}_t)/m_i)\Delta t$.*

*Proof.* We aim to show that our parameterization of the bias force in (10) ensures that the distance between the current position $\boldsymbol{R}_t$ and some target state $\boldsymbol{R}_T \sim \pi_{\mathcal{B}}$ in the target distribution is *non-increasing*. For each atom $i \in \{1, \ldots, n\}$, we define the bias force as

$$\boldsymbol{b}_\theta^i := \alpha_\theta^i \hat{\boldsymbol{s}}_i + \left( \boldsymbol{I} - \hat{\boldsymbol{s}}_i \hat{\boldsymbol{s}}_i^\top \right) \boldsymbol{h}_\theta^i, \qquad \alpha_\theta^i \geq 0, \ \ \hat{\boldsymbol{s}}_i = \frac{\nabla_{\boldsymbol{r}_t^i} \log \pi_{\mathcal{B}}}{\|\nabla_{\boldsymbol{r}_t^i} \log \pi_{\mathcal{B}}\|} \tag{51}$$

After the update in the direction of the potential energy $-\nabla_{\boldsymbol{r}_t^i} U(\boldsymbol{R}_t)$, the Euler update to the position with time step $\Delta t$ using the bias force is

$$\boldsymbol{r}_{t+\Delta t}^i = \boldsymbol{r}_t^i + \frac{\Delta t}{m_i} \left( \alpha_\theta^i \hat{\boldsymbol{s}}_i + \left( \boldsymbol{I} - \hat{\boldsymbol{s}}_i \hat{\boldsymbol{s}}_i^\top \right) \boldsymbol{h}_\theta^i \right) = \boldsymbol{r}_t^i + \frac{\Delta t}{m_i} \boldsymbol{b}_\theta^i \tag{52}$$

Let $\boldsymbol{r}_{\mathcal{B}}^i \in \mathrm{supp}(\pi_{\mathcal{B}})$ be a target point chosen on the ascent ray of $\hat{\boldsymbol{s}}_i$ given by

$$\boldsymbol{r}_{\mathcal{B}}^i \in \left\{ \boldsymbol{r}_t^i + \lambda \hat{\boldsymbol{s}}_i : \ \lambda \geq 0 \right\} \cap \mathrm{supp}(\pi_{\mathcal{B}}) \tag{53}$$

We denote displacement to this target as $\boldsymbol{d}_i := \boldsymbol{r}_{\mathcal{B}}^i - \boldsymbol{r}_t^i = \rho_i \hat{\boldsymbol{d}}_i$ with $\hat{\boldsymbol{d}}_i := \frac{\boldsymbol{d}_i}{\|\boldsymbol{d}_i\|}$ and $\rho_i := \|\boldsymbol{d}_i\| \geq 0$. By construction of the ray, $\hat{\boldsymbol{d}}_i = \hat{\boldsymbol{s}}_i$ (i.e., $\boldsymbol{d}_i$ is parallel with $\hat{\boldsymbol{s}}_i$). After a single step, the squared

distance is given by

$$\|\boldsymbol{r}_\mathcal{B}^i - \boldsymbol{r}_{t+\Delta t}^i\|^2 = \left\|\boldsymbol{r}_\mathcal{B}^i - \left(\boldsymbol{r}_t^i + \frac{\Delta t}{m_i}\boldsymbol{b}_\theta^i\right)\right\|^2$$

$$= \left\|\boldsymbol{d}_i - \frac{\Delta t}{m_i}\boldsymbol{b}_\theta^i\right\|^2$$

$$= \|\boldsymbol{d}_i\|^2 - \frac{2\Delta t}{m_i}\langle\boldsymbol{d}_i, \boldsymbol{b}_\theta^i\rangle + \frac{\Delta t^2}{m_i^2}\|\boldsymbol{b}_\theta^i\|^2 \tag{54}$$

Using the decomposition of $\boldsymbol{b}_\theta^i$ and the fact that $\boldsymbol{d}_i \parallel \hat{\boldsymbol{s}}_i$ and $(\boldsymbol{I} - \hat{\boldsymbol{s}}_i\hat{\boldsymbol{s}}_i^\top)\boldsymbol{h}_\theta^i \perp \hat{\boldsymbol{s}}_i$, we obtain

$$\langle\boldsymbol{d}_i, \boldsymbol{b}_\theta^i\rangle = \alpha_\theta^i\langle\boldsymbol{d}_i, \hat{\boldsymbol{s}}_i\rangle + \langle\boldsymbol{d}_i, \underbrace{(\boldsymbol{I} - \hat{\boldsymbol{s}}_i\hat{\boldsymbol{s}}_i^\top)\boldsymbol{h}_\theta^i}_{\perp\,\hat{\boldsymbol{s}}_i}\rangle = \alpha_\theta^i\langle\rho_i\hat{\boldsymbol{s}}_i, \hat{\boldsymbol{s}}_i\rangle = \alpha_\theta^i\rho_i \geq 0 \tag{55}$$

Thus, the first-order term in (54) always decreases (or preserves) the distance, and the orthogonal correction cannot increase it at first order because it is orthogonal to $\boldsymbol{d}_t^i$. When $\Delta t \to 0$, the second-order term becomes negligible and the first-order term dominates. We determine the exact threshold for $\Delta t$ that guarantees a non-increasing distance to be

$$\frac{2\Delta t}{m_i}\alpha_\theta^i\rho_i \geq \frac{\Delta t^2}{m_i^2}\|\boldsymbol{b}_\theta^i\|^2 \implies \Delta t \leq \frac{2m_i\alpha_\theta^i\rho_i}{\|\boldsymbol{b}_\theta^i\|^2} \tag{56}$$

under which the decrease due to the first-order term dominates the quadratic increase, yielding

$$\|\boldsymbol{r}_{t+\Delta t}^i - \boldsymbol{r}_\mathcal{B}^i\| \leq \|\boldsymbol{r}_t^i - \boldsymbol{r}_\mathcal{B}^i\| \tag{57}$$

In the continuous-time limit as $\Delta t \to 0$, the non-increasing guarantee follows immediately from (55):

$$\frac{\partial}{\partial t}\|\boldsymbol{r}_\mathcal{B}^i - \boldsymbol{r}_t^i\|^2 = \frac{\partial}{\partial t}\|\boldsymbol{d}_i\|^2 = -\frac{2}{m_i}\langle\boldsymbol{d}_t^i, \boldsymbol{b}_\theta^i\rangle = -\frac{2}{m_i}\alpha_\theta^i\rho_i \leq 0 \tag{58}$$

Summing over $i = 1, \ldots, n$ (or equivalently using the concatenated coordinates), we conclude that there exists a target configuration $\boldsymbol{R}_\mathcal{B} \in \pi_\mathcal{B}$ such that the distance to $\boldsymbol{R}_\mathcal{B}$ is *non-increasing* under the bias force update, which proves the claim. $\qquad\square$

### C.3 PROOF OF PROPOSITION 4.2

**Proposition 4.2** (Convexity and Uniqueness of Cross-Entropy Objective). *The cross-entropy objective $\mathcal{L}_{CE}$ is convex in $\mathbb{P}^{b_\theta}$ and there exists a unique minimizer $\boldsymbol{b}^\star$ that is the solution to the EntangledSOC problem in Proposition 3.1.*

*Proof.* First, we show that the cross-entropy objective is convex in the path measure $\mathbb{P}^{b_\theta}$. The functional $\mathbb{P}^{b_\theta} \mapsto D_{\mathrm{KL}}(\mathbb{P}^\star\|\mathbb{P}^{b_\theta})$ is convex with respect to its second argument, since $x \mapsto -\log x$ is convex. Since $\mathcal{L}_{\mathrm{CE}}(\mathbb{P}^\star, \mathbb{P}^{b_\theta}) := D_{\mathrm{KL}}(\mathbb{P}^\star\|\mathbb{P}^{b_\theta})$ which is strictly convex in $\mathbb{P}^{b_\theta}$, minimizing $\mathcal{L}_{\mathrm{CE}}$ yields a unique minimizer $\mathbb{P}^{b^\star} \equiv \mathbb{P}^\star$. From Proposition 3.1, we have that the unique minimizer of $\mathcal{L}_{\mathrm{CE}}$ is exactly the solution to the EntangledSB problem. While this does not necessarily imply convexity in the neural network parameters $\theta$, it yields a more favorable optimization objective. $\qquad\square$

### C.4 PROOF OF PROPOSITION 4.3

**Proposition 4.3** (Equivalence of Variational and Path Integral Objectives). *The cross-entropy objective can be expressed in path-integral form as*

$$\mathcal{L}_{CE}(\theta) = \mathbb{E}_{\mathbb{P}^v}\left[w^\star(\boldsymbol{X}_{0:T})\mathcal{F}_{\boldsymbol{b}_\theta, \boldsymbol{v}}(\boldsymbol{X}_{0:T})\right] \tag{14}$$

$$\begin{cases} w^\star(\boldsymbol{X}_{0:T}) = \frac{\mathrm{d}\mathbb{P}^\star}{\mathrm{d}\mathbb{P}^v}(\boldsymbol{X}_{0:T}) = \frac{e^{r(\boldsymbol{X}_T)}}{Z}\frac{\mathrm{d}\mathbb{P}^0}{\mathrm{d}\mathbb{P}^v}(\boldsymbol{X}_{0:T}) \\ \mathcal{F}_{\boldsymbol{b}_\theta, \boldsymbol{v}}(\boldsymbol{X}_{0:T}) = \frac{1}{2}\int_0^T\|\boldsymbol{b}_\theta(\boldsymbol{R}_t, \boldsymbol{V}_t)\|^2 - \int_0^T(\boldsymbol{b}_\theta^\top\boldsymbol{v})(\boldsymbol{R}_t, \boldsymbol{V}_t)dt - \int_0^T\boldsymbol{b}_\theta(\boldsymbol{R}_t, \boldsymbol{V}_t)^\top d\boldsymbol{W}_t \end{cases} \tag{15}$$

*where we define the reference measure $\boldsymbol{v} = \bar{\boldsymbol{b}} := \mathtt{stopgrad}(\boldsymbol{b}_\theta)$ is the off-policy control drift from the previous iteration.*

From the cross-entropy objective, we have

$$\mathcal{L}_{\text{CE}}(\mathbb{P}^{\star}, \mathbb{P}^{b_\theta}) := D_{\text{KL}}(\mathbb{P}^{\star}\|\mathbb{P}^{b_\theta}) = \mathbb{E}_{\mathbb{P}^{\star}}\left[\log \frac{d\mathbb{P}^{\star}}{d\mathbb{P}^{b_\theta}}\right] = \mathbb{E}_{\mathbb{P}^{\bar{b}}}\left[\frac{d\mathbb{P}^{\star}}{d\mathbb{P}^{\bar{b}}}\log\frac{d\mathbb{P}^{\star}}{d\mathbb{P}^{b_\theta}}\right] \tag{59}$$

Setting $w^{\star} := \frac{d\mathbb{P}^{\star}}{d\mathbb{P}^{\bar{b}}}$, we can derive

$$w^{\star}(\boldsymbol{X}_{0:T}) = \frac{d\mathbb{P}^{\star}}{d\mathbb{P}^{\bar{b}}}(\boldsymbol{X}_{0:T}) = \frac{d\mathbb{P}^{\star}}{d\mathbb{P}^0}\frac{d\mathbb{P}^0}{d\mathbb{P}^{\bar{b}}}(\boldsymbol{X}_{0:T}) \tag{60}$$

From Lemma C.1, we have

$$w^{\star}(\boldsymbol{X}_{0:T}) = \frac{e^{r(\boldsymbol{X}_T)}}{Z}\frac{d\mathbb{P}^0}{d\mathbb{P}^{\bar{b}}}(\boldsymbol{X}_{0:T}) \tag{61}$$

which is our definition for $w^{\star}$. Now, expanding $\log\frac{d\mathbb{P}^{\star}}{d\mathbb{P}^{b_\theta}}$, we have

$$\log\frac{d\mathbb{P}^{\star}}{d\mathbb{P}^{b_\theta}} = \log\frac{d\mathbb{P}^{\star}}{d\mathbb{P}^0} + \log\frac{d\mathbb{P}^0}{d\mathbb{P}^{b_\theta}}$$

$$= \log\frac{e^{r(\boldsymbol{X}_T)}}{Z} - \log\frac{d\mathbb{P}^{b_\theta}}{d\mathbb{P}^0}$$

$$= r(\boldsymbol{X}_T) - \log Z - \log\frac{d\mathbb{P}^{b_\theta}}{d\mathbb{P}^0} \tag{62}$$

Applying Girsanov's theorem from Lemma B.1, we expand the second term $\log\frac{d\mathbb{P}^{b_\theta}}{d\mathbb{P}^0}$ as

$$\log\frac{d\mathbb{P}^{b_\theta}}{d\mathbb{P}^0}(\boldsymbol{X}_{0:T}) = \int_0^T (\boldsymbol{b}_\theta(\boldsymbol{X}_t)^{\top}\boldsymbol{\Sigma}^{-1})d\boldsymbol{X}_t - \int_0^T (\boldsymbol{\Sigma}^{-1}\boldsymbol{f}(\boldsymbol{X}_t))^{\top}\boldsymbol{b}_\theta(\boldsymbol{X}_t)dt - \frac{1}{2}\int_0^T \|\boldsymbol{b}_\theta(\boldsymbol{X}_t)\|^2 dt$$

Given that $d\boldsymbol{X}_t$ evolves via the SDE

$$d\boldsymbol{X}_t = (\boldsymbol{f}(\boldsymbol{X}_t) + \boldsymbol{\Sigma}\bar{\boldsymbol{b}}(\boldsymbol{X}_t))dt + \boldsymbol{\Sigma}d\boldsymbol{W}_t \tag{63}$$

Applying $(\boldsymbol{b}_\theta^{\top}\boldsymbol{\Sigma}^{-1})$ to both sides of the equation, we have

$$(\boldsymbol{b}_\theta^{\top}\boldsymbol{\Sigma}^{-1})d\boldsymbol{X}_t = (\boldsymbol{b}_\theta^{\top}\boldsymbol{\Sigma}^{-1})(\boldsymbol{f}(\boldsymbol{X}_t) + \boldsymbol{\Sigma}\bar{\boldsymbol{b}}(\boldsymbol{X}_t))dt + (\boldsymbol{b}_\theta^{\top}\boldsymbol{\Sigma}^{-1})\boldsymbol{\Sigma}d\boldsymbol{W}_t$$

$$= \boldsymbol{b}_\theta^{\top}(\boldsymbol{\Sigma}^{-1}\boldsymbol{f})(\boldsymbol{X}_t)dt + (\boldsymbol{b}_\theta^{\top}\boldsymbol{\Sigma}^{-1}\boldsymbol{\Sigma}\bar{\boldsymbol{b}})(\boldsymbol{X}_t)dt + (\boldsymbol{b}_\theta^{\top}\boldsymbol{\Sigma}^{-1}\boldsymbol{\Sigma})d\boldsymbol{W}_t$$

$$= (\boldsymbol{\Sigma}^{-1}\boldsymbol{f}(\boldsymbol{X}_t))^{\top}\boldsymbol{b}_\theta(\boldsymbol{X}_t)dt + (\boldsymbol{b}_\theta^{\top}\bar{\boldsymbol{b}})(\boldsymbol{X}_t)dt + \boldsymbol{b}_\theta^{\top}d\boldsymbol{W}_t \tag{64}$$

Substituting this into (63), we get

$$\log\frac{d\mathbb{P}^{b_\theta}}{d\mathbb{P}^0}(\boldsymbol{X}_{0:T}) = \int_0^T (\boldsymbol{b}_\theta(\boldsymbol{X}_t)^{\top}\boldsymbol{\Sigma}^{-1})d\boldsymbol{X}_t - \int_0^T (\boldsymbol{\Sigma}^{-1}\boldsymbol{f}(\boldsymbol{X}_t))^{\top}\boldsymbol{b}_\theta(\boldsymbol{X}_t)dt - \frac{1}{2}\int_0^T \|\boldsymbol{b}_\theta(\boldsymbol{X}_t)\|^2 dt$$

$$= \int_0^T \left((\boldsymbol{\Sigma}^{-1}\boldsymbol{f}(\boldsymbol{X}_t))^{\top}\boldsymbol{b}_\theta(\boldsymbol{X}_t)dt + (\boldsymbol{b}_\theta^{\top}\bar{\boldsymbol{b}})(\boldsymbol{X}_t)dt + \boldsymbol{b}_\theta^{\top}d\boldsymbol{W}_t\right)$$

$$\quad - \int_0^T (\boldsymbol{\Sigma}^{-1}\boldsymbol{f}(\boldsymbol{X}_t))^{\top}\boldsymbol{b}_\theta(\boldsymbol{X}_t)dt - \frac{1}{2}\int_0^T \|\boldsymbol{b}_\theta(\boldsymbol{X}_t)\|^2 dt$$

$$= \int_0^T (\boldsymbol{\Sigma}^{-1}\boldsymbol{f}(\boldsymbol{X}_t))^{\top}\boldsymbol{b}_\theta(\boldsymbol{X}_t)dt + \int_0^T (\boldsymbol{b}_\theta^{\top}\bar{\boldsymbol{b}})(\boldsymbol{X}_t)dt + \int_0^T \boldsymbol{b}_\theta^{\top}d\boldsymbol{W}_t$$

$$\quad - \int_0^T (\boldsymbol{\Sigma}^{-1}\boldsymbol{f}(\boldsymbol{X}_t))^{\top}\boldsymbol{b}_\theta(\boldsymbol{X}_t)dt - \frac{1}{2}\int_0^T \|\boldsymbol{b}_\theta(\boldsymbol{X}_t)\|^2 dt$$

$$= \int_0^T (\boldsymbol{b}_\theta^{\top}\bar{\boldsymbol{b}})(\boldsymbol{X}_t)dt + \int_0^T \boldsymbol{b}_\theta^{\top}d\boldsymbol{W}_t - \frac{1}{2}\int_0^T \|\boldsymbol{b}_\theta(\boldsymbol{X}_t)\|^2 dt \tag{65}$$

Putting everything together into the expression for the cross-entropy objective, we get

$$\mathcal{L}_{\text{CE}}(\boldsymbol{b}_\theta) = \mathbb{E}_{\mathbb{P}^v}\left[w^{\star}(\boldsymbol{X}_{0:T})\left(r(\boldsymbol{X}_T) - \log Z - \log\frac{d\mathbb{P}^{b_\theta}}{d\mathbb{P}^0}(\boldsymbol{X}_{0:T})\right)\right]$$

$$= \mathbb{E}_{\mathbb{P}^v}\left[w^{\star}(\boldsymbol{X}_{0:T})\left(r(\boldsymbol{X}_T) - \log Z - \left[\int_0^T (\boldsymbol{b}_\theta^{\top}\bar{\boldsymbol{b}})(\boldsymbol{X}_t)dt + \int_0^T \boldsymbol{b}_\theta^{\top}d\boldsymbol{W}_t - \frac{1}{2}\int_0^T \|\boldsymbol{b}_\theta(\boldsymbol{X}_t)\|^2 dt\right]\right)\right]$$

$$= \mathbb{E}_{\mathbb{P}^v}\left[w^{\star}(\boldsymbol{X}_{0:T})(r(\boldsymbol{X}_T) - \log Z) - w^{\star}(\boldsymbol{X}_{0:T})\left(\int_0^T (\boldsymbol{b}_\theta^{\top}\bar{\boldsymbol{b}})(\boldsymbol{X}_t)dt + \int_0^T \boldsymbol{b}_\theta^{\top}d\boldsymbol{W}_t - \frac{1}{2}\int_0^T \|\boldsymbol{b}_\theta(\boldsymbol{X}_t)\|^2 dt\right)\right]$$

$$= \mathbb{E}_{\mathbb{P}^v}\Bigg[\underbrace{w^{\star}(\boldsymbol{X}_{0:T})(r(\boldsymbol{X}_T) - \log Z)}_{\text{independent of }\theta} + w^{\star}(\boldsymbol{X}_{0:T})\underbrace{\left(\frac{1}{2}\int_0^T \|\boldsymbol{b}_\theta(\boldsymbol{X}_t)\|^2 dt - \int_0^T (\boldsymbol{b}_\theta^{\top}\bar{\boldsymbol{b}})(\boldsymbol{X}_t)dt - \int_0^T \boldsymbol{b}_\theta^{\top}d\boldsymbol{W}_t\right)}_{:=\mathcal{F}_{\boldsymbol{b}_\theta,\bar{\boldsymbol{b}}}(\boldsymbol{X}_{0:T})}\Bigg]$$

Since $w^\star(\boldsymbol{X}_{0:T})\,(r(\boldsymbol{X}_T) - \log Z)$ is independent of $\theta$, we can drop it and write the cross-entropy objective as

$$\mathcal{L}_{\text{CE}}(\theta) = \mathbb{E}_{\mathbb{P}^{\bar{b}}}\left[ w^\star(\boldsymbol{X}_{0:T})\mathcal{F}_{\boldsymbol{b}_\theta,\bar{\boldsymbol{b}}}(\boldsymbol{X}_{0:T})\right] \tag{66}$$

where we define $\bar{\boldsymbol{b}} := \texttt{stopgrad}(\boldsymbol{b}_\theta)$. $\qquad\square$

### C.5 Proof of Proposition 4.4

**Proposition 4.4** (Discretized Cross-Entropy). *Given the discretized $\hat{\mathcal{F}}_{\boldsymbol{b}_\theta,\bar{\boldsymbol{b}}}(\boldsymbol{X}_{0:K})$ in (16), we can derive a simplified loss function as*

$$\hat{\mathcal{L}}_{CE}(\theta) = \mathbb{E}_{\boldsymbol{X}_{0:K}\sim\mathbb{P}^{\bar{b}}}\left[ \underbrace{\frac{d\mathbb{P}^\star}{d\mathbb{P}^{\bar{b}}}(\boldsymbol{X}_{0:K})}_{w^\star(\boldsymbol{X}_{0:K})}\log\left(\frac{p^0(\boldsymbol{X}_{0:K})\exp(r(\boldsymbol{X}_K))}{p^{b_\theta}(\boldsymbol{X}_{0:K})}\right)\right] \tag{17}$$

*where $w^\star(\boldsymbol{X}_{0:K})$ is the importance weight of the discrete time trajectory $\boldsymbol{X}_{0:K}$.*

We will show that the discretized expression for $\hat{\mathcal{F}}_{\boldsymbol{b}_\theta,\hat{\boldsymbol{b}}}(\boldsymbol{X}_{0:K})$ can be written as

$$\hat{\mathcal{F}}_{\boldsymbol{b}_\theta,\boldsymbol{v}}(\boldsymbol{X}_{0:K}) = \log\left(\frac{p^\star(\boldsymbol{X}_{0:K})}{p^{b_\theta}(\boldsymbol{X}_{0:K})}\right) = \log\left(\frac{p^0(\boldsymbol{X}_{0:K})\exp(r(\boldsymbol{X}_K))}{p^{b_\theta}(\boldsymbol{X}_{0:K})}\right) \tag{67}$$

First, we expand the expression as

$$\log\left(\frac{p^0(\boldsymbol{X}_{0:K})\exp(r(\boldsymbol{X}_K))}{p^{b_\theta}(\boldsymbol{X}_{0:K})}\right) = \log p^0(\boldsymbol{X}_{0:K}) + r(\boldsymbol{X}_K) - \log p^{b_\theta}(\boldsymbol{X}_{0:K})$$

$$= \sum_{k=0}^{K-1}\left[\log p^\star(\boldsymbol{X}_{k+1}|\boldsymbol{X}_k) - \log p^{b_\theta}(\boldsymbol{X}_{k+1}|\boldsymbol{X}_k)\right] + r(\boldsymbol{X}_K) \tag{68}$$

The distribution of the position $\boldsymbol{R}_{k+1}$ under the uncontrolled dynamics is a Gaussian with mean $\boldsymbol{R}_k$ and co-variance $\boldsymbol{\Sigma}^\top\boldsymbol{\Sigma}\Delta t$. The log-density becomes

$$\log p^0(\boldsymbol{X}_{k+1}|\boldsymbol{X}_k) = \log\mathcal{N}\left(\boldsymbol{X}_{k+1}\big|\boldsymbol{X}_k, \boldsymbol{\Sigma}^\top\boldsymbol{\Sigma}\Delta t\right)$$

$$= -\frac{1}{2}(\boldsymbol{X}_{k+1} - \boldsymbol{X}_k)^\top(\boldsymbol{\Sigma}^\top\boldsymbol{\Sigma}\Delta t)^{-1}(\boldsymbol{X}_{k+1} - \boldsymbol{X}_k) \tag{69}$$

Similarly, the log-density under the bias force $\boldsymbol{b}_\theta$ is given by

$$\log p^{b_\theta}(\boldsymbol{X}_{k+1}|\boldsymbol{X}_k) = \log\mathcal{N}\left(\boldsymbol{X}_{k+1}\big|\boldsymbol{X}_k + \boldsymbol{\Sigma}\boldsymbol{b}_\theta(\boldsymbol{X}_k)\Delta t, \boldsymbol{\Sigma}^\top\boldsymbol{\Sigma}\Delta t\right)$$

$$= -\frac{1}{2}(\boldsymbol{X}_{k+1} - \boldsymbol{X}_k - \boldsymbol{\Sigma}\boldsymbol{b}_\theta(\boldsymbol{X}_k)\Delta t)^\top(\boldsymbol{\Sigma}^\top\boldsymbol{\Sigma}\Delta t)^{-1}(\boldsymbol{X}_{k+1} - \boldsymbol{X}_k - \boldsymbol{\Sigma}\boldsymbol{b}_\theta(\boldsymbol{X}_k)\Delta t) \tag{70}$$

Now, we write the increment from time $k$ to $k+1$ as

$$\boldsymbol{X}_{k+1} = \boldsymbol{X}_k + \boldsymbol{\Sigma}\bar{\boldsymbol{b}}(\boldsymbol{X}_k)\Delta t + \boldsymbol{\Sigma}\Delta\boldsymbol{W}_k$$

$$\boldsymbol{X}_{k+1} - \boldsymbol{X}_k = \boldsymbol{\Sigma}\bar{\boldsymbol{b}}(\boldsymbol{X}_k)\Delta t + \boldsymbol{\Sigma}\Delta\boldsymbol{W}_k \tag{71}$$

and substitute into (69) to get

$$\log p^0(\boldsymbol{X}_{k+1}|\boldsymbol{X}_k) - \log p^{b_\theta}(\boldsymbol{X}_{k+1}|\boldsymbol{X}_k)$$

$$= -\frac{1}{2}\big(\boldsymbol{\Sigma}\bar{\boldsymbol{b}}(\boldsymbol{X}_k)\Delta t + \boldsymbol{\Sigma}\Delta\boldsymbol{W}_k\big)^\top\big(\boldsymbol{\Sigma}^\top\boldsymbol{\Sigma}\Delta t\big)^{-1}\big(\boldsymbol{\Sigma}\bar{\boldsymbol{b}}(\boldsymbol{X}_k)\Delta t + \boldsymbol{\Sigma}\Delta\boldsymbol{W}_k\big)$$

$$+ \frac{1}{2}\big(\boldsymbol{\Sigma}\bar{\boldsymbol{b}}(\boldsymbol{X}_k)\Delta t + \boldsymbol{\Sigma}\Delta\boldsymbol{W}_k - \boldsymbol{\Sigma}\boldsymbol{b}_\theta(\boldsymbol{X}_k)\Delta t\big)^\top\big(\boldsymbol{\Sigma}^\top\boldsymbol{\Sigma}\Delta t\big)^{-1}\big(\boldsymbol{\Sigma}\bar{\boldsymbol{b}}(\boldsymbol{X}_k)\Delta t + \boldsymbol{\Sigma}\Delta\boldsymbol{W}_k - \boldsymbol{\Sigma}\boldsymbol{b}_\theta(\boldsymbol{X}_k)\Delta t\big)$$

Given that $\left(\mathbf{\Sigma}^\top \mathbf{\Sigma} \Delta t\right)^{-1} = \frac{1}{\Delta t}(\mathbf{\Sigma}^\top \mathbf{\Sigma})^{-1} = \frac{1}{\Delta t}(\mathbf{\Sigma}^{-1}\mathbf{\Sigma}^{-\top})$, and rewriting $\boldsymbol{a} = \bar{\boldsymbol{b}}\Delta t + \Delta \boldsymbol{W}_k - \boldsymbol{b}_\theta \Delta t$ and $\bar{\boldsymbol{a}} = \bar{\boldsymbol{b}}\Delta t + \Delta \boldsymbol{W}_k$, we have

$$
\log p^0(\boldsymbol{X}_{k+1}|\boldsymbol{X}_k) - \log p^{b_\theta}(\boldsymbol{X}_{k+1}|\boldsymbol{X}_k)
$$

$$
= \frac{1}{2\Delta t}\left[ -(\mathbf{\Sigma}\bar{\boldsymbol{a}})^\top (\mathbf{\Sigma}^{-1}\mathbf{\Sigma}^{-\top})(\mathbf{\Sigma}\bar{\boldsymbol{a}}) + (\mathbf{\Sigma}\boldsymbol{a})^\top (\mathbf{\Sigma}^{-1}\mathbf{\Sigma}^{-\top})(\mathbf{\Sigma}\boldsymbol{a}) \right]
$$

$$
= \frac{1}{2\Delta t}\left[ -\bar{\boldsymbol{a}}^\top \left(\mathbf{\Sigma}^\top \mathbf{\Sigma}^{-1}\mathbf{\Sigma}^{-\top}\mathbf{\Sigma}\right)\bar{\boldsymbol{a}} + \boldsymbol{a}^\top \left(\mathbf{\Sigma}^\top \mathbf{\Sigma}^{-1}\mathbf{\Sigma}^{-\top}\mathbf{\Sigma}\right)\boldsymbol{a} \right]
$$

$$
= \frac{1}{2\Delta t}\left[ -\|\bar{\boldsymbol{a}}\|^2 + \|\boldsymbol{a}\|^2 \right]
$$

$$
= \frac{1}{2\Delta t}\left[ -\|\bar{\boldsymbol{b}}\Delta t + \Delta \boldsymbol{W}_k\|^2 + \|\bar{\boldsymbol{b}}\Delta t + \Delta \boldsymbol{W}_k - \boldsymbol{b}_\theta \Delta t\|^2 \right]
$$

$$
= \frac{1}{2\Delta t}\left[ -\|\bar{\boldsymbol{b}}\Delta t + \Delta \boldsymbol{W}_k\|^2 + \|\bar{\boldsymbol{b}}\Delta t + \Delta \boldsymbol{W}_k\|^2 - 2(\boldsymbol{b}_\theta \Delta t)\left(\bar{\boldsymbol{b}}\Delta t + \Delta \boldsymbol{W}_k\right) + \|\boldsymbol{b}_\theta \Delta t\|^2 \right]
$$

$$
= \frac{1}{2\Delta t}\left[ -2(\boldsymbol{b}_\theta \cdot \bar{\boldsymbol{b}})(\Delta t)^2 - 2(\boldsymbol{b}_\theta \cdot \Delta \boldsymbol{W}_k)\Delta t + \|\boldsymbol{b}_\theta\|^2(\Delta t)^2 \right]
$$

$$
= -(\boldsymbol{b}_\theta \cdot \bar{\boldsymbol{b}})\Delta t - (\boldsymbol{b}_\theta \cdot \Delta \boldsymbol{W}_k) + \frac{1}{2}\|\boldsymbol{b}_\theta\|^2 \Delta t \tag{72}
$$

Summing over all $K$ steps and adding $r(\boldsymbol{X}_K)$, we have

$$
\sum_{k=0}^{K-1}\left[ \log p^{b_\theta}(\boldsymbol{X}_{k+1}|\boldsymbol{X}_k) - \log p^0(\boldsymbol{X}_{k+1}|\boldsymbol{X}_k) \right] + r(\boldsymbol{X}_K)
$$

$$
= \underbrace{\frac{1}{2}\sum_{k=0}^{K-1}\|\boldsymbol{b}_\theta(\boldsymbol{X}_k)\|^2 \Delta t - \sum_{k=0}^{K-1}(\boldsymbol{b}_\theta \cdot \bar{\boldsymbol{b}})(\boldsymbol{X}_k)\Delta t - \sum_{k=0}^{K-1}\boldsymbol{b}_\theta(\boldsymbol{X}_k) \cdot \Delta \boldsymbol{W}_k + r(\boldsymbol{X}_K)}_{:=\hat{\mathcal{F}}_{\boldsymbol{b}_\theta, \bar{\boldsymbol{b}}}(\boldsymbol{X}_{0:K})} \tag{73}
$$

Therefore, we can write $\hat{\mathcal{F}}_{\boldsymbol{b}_\theta, \bar{\boldsymbol{b}}}(\boldsymbol{X}_{0:K})$ equal to

$$
\hat{\mathcal{F}}_{\boldsymbol{b}_\theta, \bar{\boldsymbol{b}}}(\boldsymbol{X}_{0:K}) = \sum_{k=0}^{K-1} \log \frac{p^0(\boldsymbol{X}_{k+1}|\boldsymbol{X}_k)}{p^{b_\theta}(\boldsymbol{X}_{k+1}|\boldsymbol{X}_k)} + r(\boldsymbol{X}_K) = \log\left( \frac{p^0(\boldsymbol{X}_{0:K})\exp(r(\boldsymbol{X}_K))}{p^{b_\theta}(\boldsymbol{X}_{0:K})} \right) \tag{74}
$$

where $r(\boldsymbol{X}_K) := \log \pi_{\mathcal{B}}(\boldsymbol{X}_K)$. $\qquad\square$

# D CELL PERTURBATION EXPERIMENT DETAILS

## D.1 EXPERIMENT SETUP

**Data Processing**  For this experiment, we extract the cell perturbation data from the Tahoe-100M dataset consists of 50 cell lines and over 1000 different drug-dose conditions (Zhang et al., 2025a). Specifically, we use data on a single cell line (A-549) under two drug perturbation conditions: Clonidine at $5\mu M$ and Trametinib at $5\mu M$. The initial distribution is defined as the DMSO-treated control cells. Using centroid-based sampling, we obtain balanced training sets of 1033 cells per cluster (Figure 5). We split the cells into training and validation sets with a 0.9/0.1 ratio. All final visualizations are plotted with the first two principal components, but the cell state trajectories are simulated for $d \in \{50, 100, 150\}$ for Clonidine and $d = 50$ for Trametinib.

Table 3: **Training cluster cell counts for perturbation experiments.**

|  | Clonidine | | Trametinib | | |
| --- | --- | --- | --- | --- | --- |
|  | Cluster 1 | Cluster 2 | Cluster 1 | Cluster 2 | Cluster 3 |
| Original Cell Count | 1675 | 1033 | 1622 | 686 | 381 |

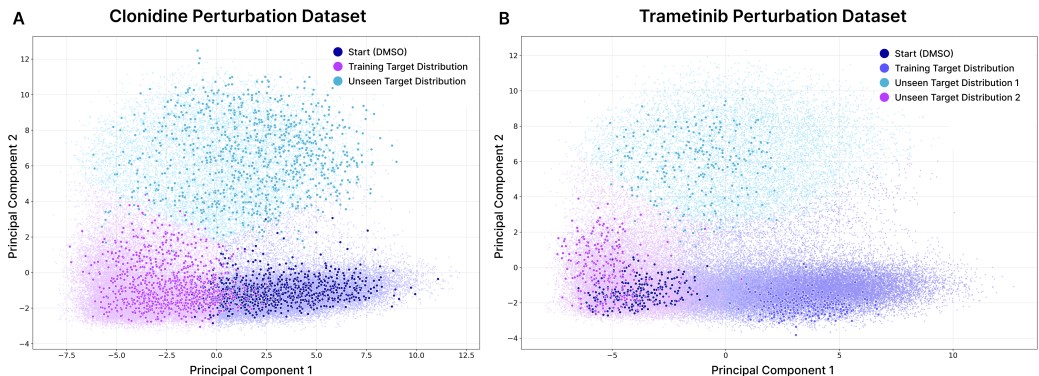

Figure 5: **Training and held-out cell clusters for the cell perturbation experiment.** All cells are plotted with the top 2 PCs. **(A)** Clonidine ($5\mu L$) perturbation data containing the initial DMSO-treated control cells (dark blue), perturbed cluster for training (magenta), and held-out validation cluster (turquoise). **(B)** Trametinib ($5\mu L$) perturbation data containing the initial DMSO-treated control cells (navy blue), the perturbed cluster for training (purple), and two held-out perturbed clusters (magenta and turquoise).

**Model Architecture** To learn the dependencies across the $n$ particles, we use Transformer blocks to learn dependencies between the positions $\boldsymbol{R}_t \in \mathbb{R}^{n \times d}$ and velocities $\boldsymbol{V}_t \in \mathbb{R}^{n \times d}$ in the system. The input to the model is the concatenated position and velocity vectors $\mathrm{Cat}(\boldsymbol{R}_t, \boldsymbol{V}_t) \in \mathbb{R}^{n \times 6}$. The architecture consists of an input projection layer that projects the input to $d_{\text{hidden}} = 256$, a 4-layer Transformer Encoder (`gelu` activation, feedforward dimension $d_{\text{ff}} = 512$, 8 attention heads, and dropout 0.1), and two MLP decoder heads that predict the scaling factor $\alpha_t(\boldsymbol{R}_t, \boldsymbol{V}_t)$ and correction vector $\boldsymbol{h}_t^i(\boldsymbol{R}_t, \boldsymbol{V}_t)$ per atom. To ensure positive scaling, the `softplus` activation is applied to the output of the scalar MLP head. The correction vector is projected onto the orthogonal plane of the vector pointing in the direction of the target $\hat{\boldsymbol{s}}_i$, and the output bias force is the sum of the scaled directional unit vector $\alpha_\theta^i(\boldsymbol{R}_t, \boldsymbol{V}_t)\hat{\boldsymbol{s}}_i$ and the projected correction vector $(\boldsymbol{I} - \hat{\boldsymbol{s}}_i \hat{\boldsymbol{s}}_i^\top)\boldsymbol{h}_\theta^i(\boldsymbol{R}_t, \boldsymbol{V}_t)$.

**Terminal Reward** To solve the EntangledSB problem, we defined the terminal reward to be the log-probability under the target density $\pi_\mathcal{B}$ given by $r(\boldsymbol{X}_T) = \log \pi_\mathcal{B}(\boldsymbol{R}_T)$. We set the target density as the Gaussian centered around $\boldsymbol{R}_\mathcal{B}$ with radius $\sigma$.

$$\pi_\mathcal{B}(\boldsymbol{R}_T) = \frac{1}{(2\pi\sigma^2)^{\frac{d}{2}}} \exp\left(\frac{-\|\boldsymbol{R}_T - \boldsymbol{R}_\mathcal{B}\|_2^2}{2\sigma^2}\right) \tag{75}$$

Therefore, to train the bias force to reconstruct the input target state of a cell cluster $\boldsymbol{R}_\mathcal{B} \in \mathbb{R}^{n \times d}$, we define $r : \mathcal{X} \to \mathbb{R}$ as:

$$r(\boldsymbol{X}_T) := \frac{-\|\boldsymbol{R}_T - \boldsymbol{R}_\mathcal{B}\|_2^2}{2\sigma^2} \tag{76}$$

The specific values for $\sigma$ are provided in Table 4.

**Simulating Trajectories on the Data Manifold** To define an energy landscape on the data manifold, we use the RBF metric introduced in Kapuśniak et al. (2024), which is lower in magnitude when $\boldsymbol{x}$ is within the support of the data manifold and larger in magnitude when $\boldsymbol{x}$ moves away from the support of the dataset into sparse regions. First, we define the elements of a function $h_j^{\text{RBF}}$ that satisfies $h_j^{\text{RBF}}(\boldsymbol{x}) \approx 1$ on the data manifold as

$$h_j^{\text{RBF}}(\boldsymbol{x}) = \sum_{m=1}^{N_c} \omega_{m,j}(\boldsymbol{x}) \exp\left(-\frac{\lambda_m}{2}\|\boldsymbol{x} - \hat{\boldsymbol{x}}_n\|^2\right) \tag{77}$$

$$\lambda_m = \frac{1}{2}\left(\frac{\kappa}{|C_m|} \sum_{\boldsymbol{x} \in C_m} \|\boldsymbol{x} - \hat{\boldsymbol{x}}_m\|^2\right)^{-2} \tag{78}$$

where $\{\omega_{m,j}\}_{m=1}^{N_c}$ for each cluster and each coordinate are learned given a dataset $\mathcal{D}$ to enforce $h_j^{\text{RBF}}(\boldsymbol{x}) \approx 1$ for all $\boldsymbol{x} \in \mathcal{D}$ with the following loss function

$$\mathcal{L}_{\text{RBF}}(\{\omega_{m,j}\}) = \sum_{\boldsymbol{x}_i \in \mathcal{D}} (1 - h_j^{\text{RBF}}(\boldsymbol{x}_i))^2 \tag{79}$$

Then, we define the potential energy function of the base dynamics as

$$U(\boldsymbol{x}) = -\sum_{j=1}^{d} \log(M_j(\boldsymbol{x}) + \varepsilon), \quad \text{where} \quad M_j(\boldsymbol{x}) = \frac{1}{(h_j^{\text{RBF}}(\boldsymbol{x}) + \varepsilon)^{\alpha}} \tag{80}$$

where $M_j(\boldsymbol{x})$ is small in regions of high data density and large in regions of low data density. The resulting force $-\nabla_{\boldsymbol{x}} U(\boldsymbol{x})$ is used by the simulator as the natural-gradient step.

**Hyperparameters** We present the hyperparameters used for the cell perturbation modeling experiment in Table 4. For each perturbation, we conducted ablations while keeping all other parameters constant on: **(1)** using the log-variance divergence with the learnable control variate $\mathcal{L}_{\text{LV}}$ described in B.1 instead of the cross-entropy objective $\mathcal{L}_{\text{CE}}$ and **(2)** removing the dependency on velocities as a feature input to the bias force model $\boldsymbol{b}_{\theta}(\boldsymbol{R}_t)$. The hyperparameters of the Transformer architecture are given in App D.1 and are kept constant across all experiments. All models are trained with the Adam optimizer (Kingma & Ba, 2014) with learning rate $\eta = 0.0001$. Due to the small batch sizes used for training, we leverage the importance weights $w^{\star}(\boldsymbol{X}_{0:T})$ for categorical sampling from the replay buffer $\mathcal{R}$ as $\boldsymbol{X}_{0:T} \sim \texttt{Cat}(\texttt{softmax}_{\mathcal{R}}(w^{\star}(\boldsymbol{X}_{0:T})))$ to mimic the effect of the reweighting in the cross-entropy objective $\mathcal{L}_{\text{CE}}$ over a larger batch size.

Table 4: **Hyperparameter settings for cell perturbation experiment.** The Clonidine perturbation experiment is split into three columns for each of the three dimensions of principal components (PCs) used $d \in \{50, 100, 150\}$.

| Parameter | Clonidine | | | Trametinib |
|---|---|---|---|---|
| | 50 PCs | 100 PCs | 150 PCs | 50 PCs |
| number of rollouts $N_{\text{rollouts}}$ | 100 | 100 | 100 | 100 |
| trains per rollout $N_{\text{epochs}}$ | 1000 | 1000 | 1000 | 1000 |
| step size $\Delta t$ | 0.01 | 0.01 | 0.01 | 0.01 |
| total time steps $T$ | 100 | 100 | 100 | 100 |
| number of samples $M$ | 64 | 64 | 64 | 64 |
| number of particles $n$ | 16 | 16 | 16 | 16 |
| batch size $N_{\text{batch}}$ | 64 | 64 | 64 | 64 |
| buffer size $|\mathcal{R}|$ | 1000 | 1000 | 1000 | 1000 |
| radius $\sigma$ | 0.1 | 0.1 | 0.1 | 0.1 |
| friction $\gamma$ | 2.0 | 2.0 | 2.0 | 2.0 |
| learning rate | 0.0001 | 0.0001 | 0.0001 | 0.0001 |
| RBF $N_c$ | 150 | 300 | 300 | 150 |
| RBF $\kappa$ | 1.5 | 2.0 | 3.0 | 1.5 |

## D.2 EVALUATION METRICS

**Maximum Mean Discrepency (RBF-MMD)** We evaluate reconstruction accuracy of the target distribution and the distribution simulated with EntangledSBM using MMD with the RBF kernel (RBF-MMD) on all $d$ principal components used during training. For Clonidine, we evaluate $d \in \{50, 100, 150\}$ and for Trametinib, we evaluate $d = 50$. Given the simulated endpoints of $M$ paths for $M$ cell clusters $\{\boldsymbol{R}_T \in \mathbb{R}^{n \times d}\}_{j=1}^{M}$ and the target states $\{\boldsymbol{R}_{\mathcal{B}} \in \mathbb{R}^{n \times d}\}_{\ell=1}^{M}$, the RBF-MMD is calculated as

$$\text{RBF-MMD} = \frac{1}{M^2} \sum_{j=1}^{M} \sum_{\ell=1}^{M} k_{\text{mix}}(\boldsymbol{R}_T^j, \boldsymbol{R}_T^\ell) + \frac{1}{M^2} \sum_{j=1}^{M} \sum_{\ell=1}^{M} k_{\text{mix}}(\boldsymbol{R}_{\mathcal{B}}^j, \boldsymbol{R}_{\mathcal{B}}^\ell) - \frac{2}{M^2} \sum_{j=1}^{M} \sum_{\ell=1}^{M} k_{\text{mix}}(\boldsymbol{R}_T^j, \boldsymbol{R}_{\mathcal{B}}^\ell) \tag{81}$$

where we define the mixture of RBF kernel functions $k_{\text{mix}}(\cdot, \cdot)$ as

$$k_{\text{mix}}(\boldsymbol{R}, \boldsymbol{R}') = \frac{1}{|\Sigma|} \sum_{\sigma \in \Sigma} \exp\left(-\frac{\|\boldsymbol{R} - \boldsymbol{R}'\|^2}{2\sigma^2}\right) \tag{82}$$

for $\Sigma = \{0.01, 0.1, 1, 10, 100\}$.

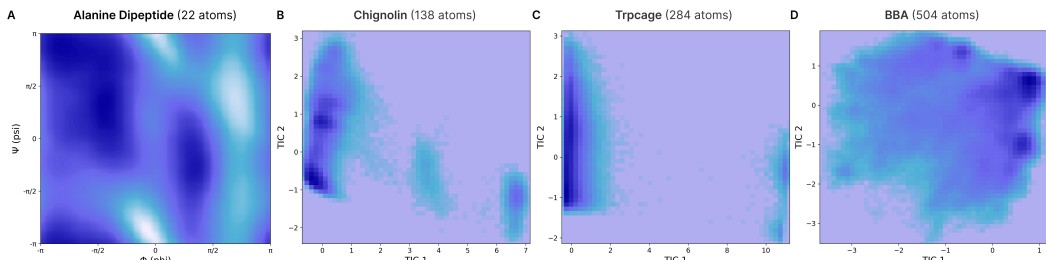

Figure 6: **Energy landscapes for transition path sampling experiments.** Potential energy plotted on dihedral angles $(\phi, \psi)$ for Alanine Dipeptide and the top two TICA components for fast-folding proteins.

**1-Wasserstein ($\mathcal{W}_1$) and 2-Wasserstein ($\mathcal{W}_2$) Distances**   We further compute the $\mathcal{W}_1$ and $\mathcal{W}_2$ distances for the top two PCs of the simulated endpoints of $M$ paths for $M$ cell clusters $\{\boldsymbol{R}_T \in \mathbb{R}^{n \times d}\}_{j=1}^M$ which form the predicted distribution $\pi_T$ and the *full distribution* that the target states are sampled from $\pi_{\mathcal{B}}$ since the $\mathcal{W}_1$ and $\mathcal{W}_2$ distances can be calculated for a pair of distributions with different sizes. Concretely, the $\mathcal{W}_1$ and $\mathcal{W}_2$ distances are calculated as

$$\mathcal{W}_1 = \left( \min_{\pi \in \Pi(\pi_T, \pi_{\mathcal{B}})} \int \|\boldsymbol{R}_T - \boldsymbol{R}_{\mathcal{B}}\|_2 d\pi(\boldsymbol{R}_T, \boldsymbol{R}_{\mathcal{B}}) \right) \tag{83}$$

$$\mathcal{W}_2 = \left( \min_{\pi \in \Pi(\pi_T, \pi_{\mathcal{B}})} \int \|\boldsymbol{R}_T - \boldsymbol{R}_{\mathcal{B}}\|_2^2 d\pi(\boldsymbol{R}_T, \boldsymbol{R}_{\mathcal{B}}) \right)^{1/2} \tag{84}$$

which quantify the minimal effort required to transform the simulated endpoint distribution into the true target distribution, demonstrating the ability of the model to capture the true perturbation dynamics.

# E   TRANSITION PATH SAMPLING EXPERIMENT DETAILS

## E.1   EXPERIMENT SETUP

**Model Architecture**   We use the same model architecture as the cell perturbation experiment described in Sec D, with the addition of Kabsch alignment (Kabsch, 1976). We define the **aligned frame** as the frame of the target coordinates and align the input positions and velocities of the *heavy atoms* (non-hydrogen atoms) to the aligned frame. The model then predicts the optimal bias force in the aligned frame, which we transform back to the original frame of the input positions.

**Terminal Reward**   Following the cell perturbation experiment, given the coordinates of a single target state $\boldsymbol{R}_{\mathcal{B}}$, we define $\pi_{\mathcal{B}}$ as the log-probability under the target Gaussian centered around the coordinates of $\boldsymbol{R}_{\mathcal{B}}$ with radius $\sigma$:

$$r(\boldsymbol{X}_T) := \exp \left( \frac{-\|\boldsymbol{R}_T - \boldsymbol{R}_{\mathcal{B}}\|_2^2}{2\sigma^2} \right) \tag{85}$$

The specific values for $\sigma$ are provided in Table 5.

**Molecular Dynamics Setup**   We closely follow the setup in (Seong et al., 2025). To simulate the position and velocity via the Langevin SDEs, we use the Velocity-Verlet with Velocity Randomization (VVVR) integrator (Sivak et al., 2014) in OpenMM (Eastman et al., 2017) that updates the state with a velocity-Verlet step for the deterministic forces and a stochastic Ornstein-Uhlenbeck step that randomizes velocities. We set the friction parameter to $\gamma = 1\text{ps}^{-1}$ and the time step size to $\Delta t = 1\text{fs}$ for a total time horizon of $T = 1000\text{fs}$ for Alanine Dipeptide and $T = 5000\text{fs}$ for the fast-folding proteins. Following (Seong et al., 2025), we leverage temperature annealing with a starting temperature of $\tau_{\text{start}} = 600\text{K}$ and a final temperature of $\tau_{\text{end}} = 300\text{K}$ for Alanine Dipeptide and Chignolin and $\tau_{\text{end}} = 400\text{K}$ for the remaining fast-folding proteins.

To simulate the unconditional dynamics following the potential energy landscape $U(\boldsymbol{R}_t)$ visualized in Fig. 6 and the corresponding force field $F = -\nabla_{\boldsymbol{R}_t} U(\boldsymbol{R}_t)$, we use the AMBER99SB force field

with the ILDN side-chain torsion corrections (amber99sbildn) force field (Lindorff-Larsen et al., 2010) for Alanine Dipeptide in a vacuum and the ff14SBonlysc forcefield (Maier et al., 2015) paired with the gbn2 solvation model (Nguyen et al., 2013) for the fast-folding proteins.

**Hyperparameters**    We present the hyperparameters used for the TPS experiment in Table 5. The hyperparameters of the Transformer architecture are given in App D.1 and are kept constant across all experiments. We run the benchmark against TPS-DPS (Seong et al., 2025) with the hyperparameters below to ensure fair comparison. All models are trained with the Adam optimizer (Kingma & Ba, 2014) with learning rate $\eta = 1 \times 10^{-4}$.

Table 5: **Hyperparameter settings for transition path sampling experiment.** The setup follows that of Seong et al. (2025) to ensure fair comparison.

| Parameter | Task | | | |
|---|---|---|---|---|
| | Alanine Dipeptide | Chignolin | Trp-cage | BBA |
| number of rollouts $N_{\text{rollouts}}$ | 100 | 100 | 100 | 100 |
| trains per rollout $N_{\text{epochs}}$ | 1000 | 1000 | 1000 | 1000 |
| step size $\Delta t$ | $1fs$ | $1fs$ | $1fs$ | $1fs$ |
| total time steps $T$ | 1000 | 5000 | 5000 | 5000 |
| number of samples $M$ | 64 | 64 | 64 | 64 |
| number of particles $n$ | 22 | 138 | 284 | 504 |
| batch size $N_{\text{batch}}$ | 16 | 1 | 1 | 1 |
| buffer size $|\mathcal{R}|$ | 1000 | 100 | 100 | 100 |
| starting temperature (Kelvin) | 600 | 600 | 600 | 600 |
| ending temperature (Kelvin) | 300 | 300 | 400 | 400 |
| radius $\sigma$ | 0.1 | 0.5 | 0.5 | 0.5 |
| friction $\gamma$ | 0.001 | 0.001 | 0.001 | 0.001 |
| learning rate | 0.0001 | 0.0001 | 0.0001 | 0.0001 |

### E.2    EVALUATION METRICS

We follow the evaluations of (Seong et al., 2025; Holdijk et al., 2023) and report three metrics: Root Mean Square Distance (RMSD), Target Hit Percentage (THP), and Energy of Transition State (ETS).

**Root Mean Square Distance (RMSD)**    RMSD ($\downarrow$) measures the distance between the *heavy atoms* (non-hydrogen atoms) of the final position of the system at time $\boldsymbol{R}_T$ and the target position $\boldsymbol{R}_{\mathcal{B}}$. To align the coordinate frames, we use the Kabsch algorithm (Kabsch, 1976), which determines the optimal rotation and translation such that two pairs of heavy atoms are aligned. The mean and standard deviation across the hits for $M$ paths are reported.

**Target Hit Percentage (THP)**    THP ($\uparrow$) measures the percentage of the total trajectories where the final position $\boldsymbol{R}_T$ reaches the vicinity of the target state $\boldsymbol{R}_{\mathcal{B}}$. Specifically, we consider a hit when the two backbone dihedral angles $(\phi, \psi)$ for Alanine Dipeptide or the first two TICA components for the fast-folding proteins (Chignolin, Trp-cage, BBA), denoted $\xi(\boldsymbol{R})$, are within the 0.75-radius sphere around the target defined as $\pi_{\mathcal{B}} = \{\boldsymbol{R} \mid \|\xi(\boldsymbol{R}) - \xi(\boldsymbol{R}_{\mathcal{B}})\| < 0.75\}$. For $M$ total paths, the THP is calculated as

$$\text{THP} = \frac{\sum_{i=1}^{M} \mathbf{1}[\boldsymbol{R}_T^i \in \pi_{\mathcal{B}}]}{M} \tag{86}$$

**Energy of Transition State (ETS)**    ETS ($\downarrow$) measures the maximum potential energy returned by $U : \mathcal{X} \to \mathbb{R}$ along the discrete transition path $\boldsymbol{R}_{0:K}$ in kJmol$^{-1}$. It is calculated for the trajectories that reach the vicinity of the target state $\boldsymbol{R}_T \in \pi_{\mathcal{B}}$ and is classified as a hit by the THP metric.

$$\text{ETS}(\boldsymbol{R}_{0:K}) = \max_{k \in \{1, \dots K\}} U(\boldsymbol{R}_k) \tag{87}$$

The mean and standard deviation across the hits for $M$ paths are reported.

### E.3 COMPUTE TIME

To assess the computational efficiency of our framework, we report the average training and sampling times per rollout across molecular dynamics systems of increasing atomic complexity in Table 6. The results demonstrate that both training and sampling times scale approximately linearly with the number of atoms, indicating that the cost of learning and simulating trajectories grows predictably with system size. For smaller systems, such as Alanine Dipeptide (22 atoms), each rollout completes within seconds, while larger, fast-folding proteins, such as BBA (504 atoms), remain tractable within a few minutes per rollout. These results confirm that EntangledSBM maintains practical scalability for all-atom molecular simulations across diverse molecular sizes.

Table 6: **Training and sampling times in seconds per rollout for molecular dynamics experiments on single NVIDIA B200 GPU.**

| Molecule | Sampling Time (s) | Training Time (s) |
|---|---|---|
| **Alanine Dipeptide** (22 atoms) | $31.469_{\pm 0.155}$ | $190.115_{\pm 0.475}$ |
| **Chignolin** (138 atoms) | $160.838_{\pm 0.375}$ | $437.232_{\pm 3.484}$ |
| **Trp-cage** (284 atoms) | $196.615_{\pm 7.354}$ | $946.982_{\pm 3.244}$ |
| **BBA** (504 atoms) | $291.682_{\pm 2.643}$ | $1956.893_{\pm 1.220}$ |

## F COMPARISON TO LOG-VARIANCE DIVERGENCE

In this section, we discuss the intuition behind the differences in cell state trajectories generated with the cross-entropy (CE) objective and log-variance (LV) objective for Clonidine, as shown in Fig. 7 and Table 7, and Trametinib as shown in Fig. 8 and Table 8.

We observe that the log-variance (LV) objective fails to accurately simulate the intermediate dynamics of cells following perturbation and generates nearly straight, abrupt trajectories connecting the initial and terminal populations. The trajectories generated from the LV-trained bias forces ignore the gradual, curved progression of cell states through intermediate data manifolds, in contrast to the trajectories generated from the cross-entropy (CE)-trained bias forces (Fig. 7, 8). While the LV-objective accurately reconstructs the target cell distribution, it fails to reconstruct the intermediate dynamics that arise from nonlinear coupling between position and velocity fields after perturbation (Tables 7, 8).

While the LV objective aims to make the log–likelihood ratio $\log \frac{\mathrm{d}\mathbb{P}^{\star}}{\mathrm{d}\mathbb{P}^{b_\theta}}(\boldsymbol{X}_{0:T})$ constant *in expectation* to minimize the variance, it is easiest to only maximize the terminal reward $r(\boldsymbol{X}_T)$ rather than intermediate path alignment with $\mathbb{P}^0$. In contrast, the CE objective is defined as the KL-divergence $D_{\mathrm{KL}}(\mathbb{P}^{\star}\|\mathbb{P}^{b_\theta})$ which expands into a path-integral action term $\frac{1}{2}\int_0^T \|\boldsymbol{b}_\theta(\boldsymbol{R}_t, \boldsymbol{V}_t)\|^2 dt$ using Girsanov's theorem. This explicitly regularizes the whole trajectory, rewarding gradual, smooth transport through intermediate states.

Table 7: **Full comparisons for simulating cell cluster dynamics under Clonidine perturbation with EntangledSBM.** We report RBF-MMD for all $d$ PCs with the target cluster and $\mathcal{W}_1$ and $\mathcal{W}_2$ distances of top 2 PCs against the full distribution of the perturbed cells for the seen and unseen populations after 100 simulation steps and cluster size set to $n = 16$. Mean and standard deviation of metrics from 5 independent simulations are reported. Comparisons include the base dynamics, without velocity conditioning, and EntangledSBM with the log-variance (LV) objective instead of cross-entropy (CE) for increasing principal component dimensions $d = \{50, 100, 150\}$.

| | Seen Target Distribution | | | Unseen Target Distribution | | |
|---|---|---|---|---|---|---|
| **Model** | RBF-MMD ($\downarrow$) | $\mathcal{W}_1$ ($\downarrow$) | $\mathcal{W}_2$ ($\downarrow$) | RBF-MMD ($\downarrow$) | $\mathcal{W}_1$ ($\downarrow$) | $\mathcal{W}_2$ ($\downarrow$) |
| **Base Dynamics** (50 PCs) | $0.677_{\pm 0.001}$ | $5.947_{\pm 0.005}$ | $6.015_{\pm 0.005}$ | $0.784_{\pm 0.001}$ | $8.217_{\pm 0.005}$ | $8.384_{\pm 0.005}$ |
| **EntangledSBM w/o Velocity Conditioning** | | | | | | |
| 50 PCs | $0.440_{\pm 0.000}$ | $1.741_{\pm 0.003}$ | $1.857_{\pm 0.004}$ | $0.478_{\pm 0.000}$ | $2.907_{\pm 0.006}$ | $3.022_{\pm 0.006}$ |
| 100 PCs | $0.494_{\pm 0.000}$ | $2.315_{\pm 0.004}$ | $2.423_{\pm 0.004}$ | $0.539_{\pm 0.000}$ | $4.110_{\pm 0.004}$ | $4.249_{\pm 0.003}$ |
| 150 PCs | $0.510_{\pm 0.000}$ | $2.497_{\pm 0.006}$ | $2.620_{\pm 0.006}$ | $0.560_{\pm 0.000}$ | $4.573_{\pm 0.006}$ | $4.716_{\pm 0.007}$ |
| **EntangledSBM w/ LV** | | | | | | |
| 50 PCs | $0.294_{\pm 0.000}$ | $0.205_{\pm 0.001}$ | $0.285_{\pm 0.001}$ | $0.290_{\pm 0.000}$ | $0.230_{\pm 0.001}$ | $0.382_{\pm 0.001}$ |
| 100 PCs | $0.330_{\pm 0.000}$ | $0.277_{\pm 0.001}$ | $0.332_{\pm 0.001}$ | $0.346_{\pm 0.000}$ | $0.358_{\pm 0.001}$ | $0.492_{\pm 0.001}$ |
| 150 PCs | $0.323_{\pm 0.000}$ | $0.128_{\pm 0.001}$ | $0.278_{\pm 0.000}$ | $0.327_{\pm 0.000}$ | $0.193_{\pm 0.001}$ | $0.425_{\pm 0.001}$ |
| **EntangledSBM w/ CE** | | | | | | |
| 50 PCs | $0.401_{\pm 0.000}$ | $0.342_{\pm 0.002}$ | $0.400_{\pm 0.001}$ | $0.419_{\pm 0.000}$ | $0.538_{\pm 0.013}$ | $0.705_{\pm 0.030}$ |
| 100 PCs | $0.455_{\pm 0.000}$ | $0.953_{\pm 0.025}$ | $1.015_{\pm 0.025}$ | $0.500_{\pm 0.001}$ | $0.899_{\pm 0.006}$ | $1.055_{\pm 0.008}$ |
| 150 PCs | $0.478_{\pm 0.000}$ | $0.753_{\pm 0.008}$ | $0.826_{\pm 0.007}$ | $0.506_{\pm 0.000}$ | $0.700_{\pm 0.009}$ | $0.811_{\pm 0.011}$ |

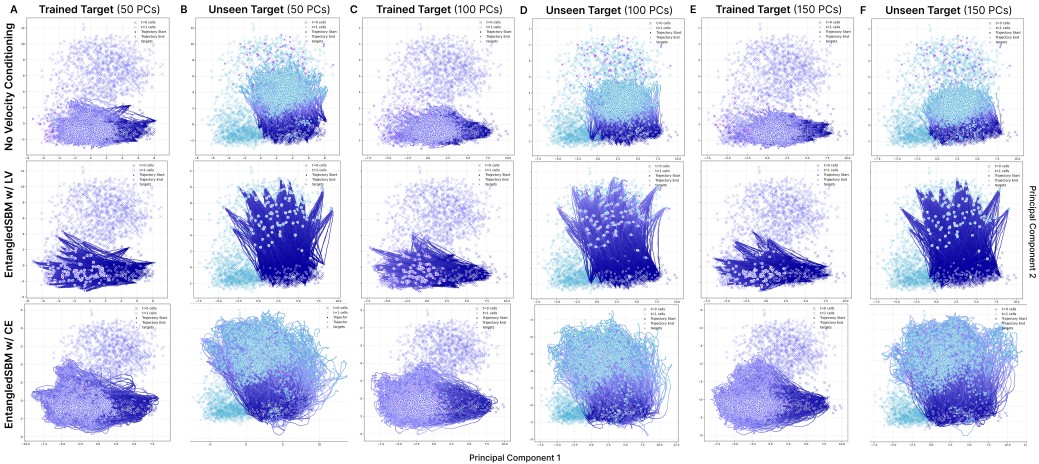

Figure 7: **Full visualization of simulated cell cluster dynamics with EntangledSBM under Clonidine perturbation.** Nearest neighbour cell clusters with $n = 16$ cells are simulated over 100 time steps. The gradient indicates the evolution of timesteps from the initial time $t = 0$ (navy) to the final time $t = T$ (purple or turquoise). 50 PCs simulated to **(A)** trained target distribution and **(B)** unseen target. 100 PCs simulated to **(C)** trained target distribution and **(D)** unseen target. 150 PCs simulated to **(E)** trained target distribution and **(F)** unseen target.

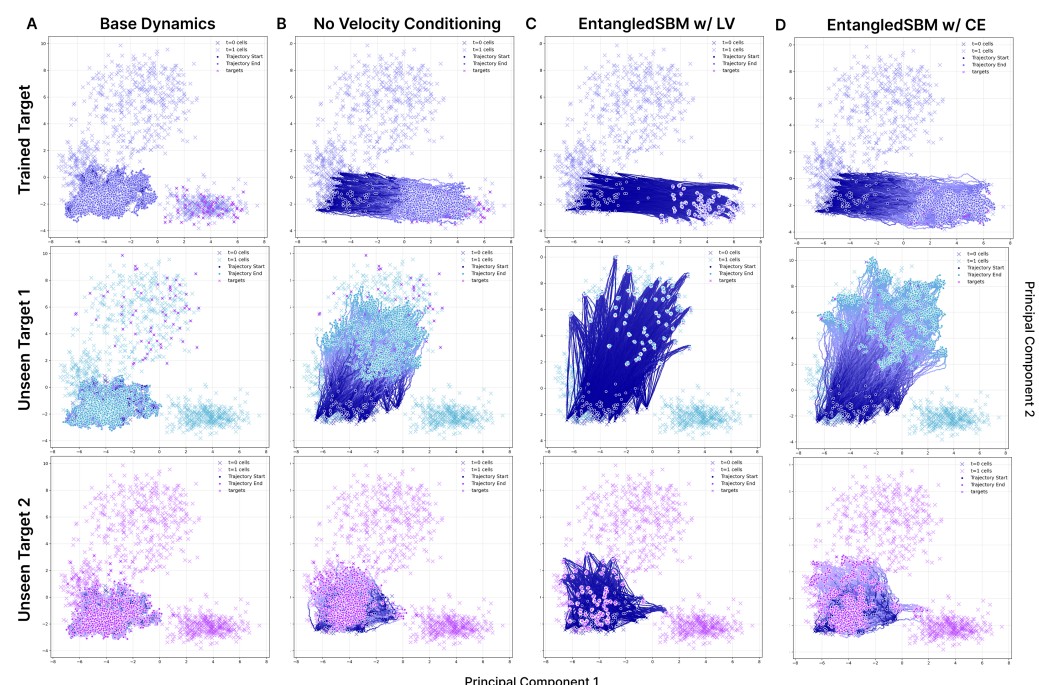

Figure 8: **Full visualization of simulated cell cluster dynamics with EntangledSBM under Trametinib perturbation.** All trajectories generated for $d = 50$ PCs. Nearest neighbour cell clusters with $n = 16$ cells are simulated over 100 time steps. The gradient indicates the evolution of timesteps from the initial time $t = 0$ (navy) to the final time $t = T$ (purple, turquoise, or magenta). **(A)** Base dynamics with no bias force. Dynamics with base and learned bias force trained with **(B)** no velocity conditioning, **(C)** log-variance (LV) objective, and **(D)** cross-entropy objective to the trained target distribution and two unseen target distributions.

Table 8: **Full comparisons for simulating cell cluster dynamics under Trametinib perturbation with EntangledSBM.** We report RBF-MMD for all $d = 50$ PCs with the target cluster and $\mathcal{W}_1$ and $\mathcal{W}_2$ distances of top 2 PCs against the full distribution of the perturbed cells for the seen and unseen populations after 100 simulation steps and cluster size set to $n = 16$. Mean and standard deviation of metrics from 5 independent simulations are reported. Comparisons include the base dynamics, without velocity conditioning, and EntangledSBM with the log-variance (LV) objective instead of cross-entropy (CE).

| Method | Seen Target Distribution | | | Unseen Target Distribution 1 | | | Unseen Target Distribution 2 | | |
|---|---|---|---|---|---|---|---|---|---|
| | RBF-MMD ($\downarrow$) | $\mathcal{W}_1$ ($\downarrow$) | $\mathcal{W}_2$ ($\downarrow$) | RBF-MMD ($\downarrow$) | $\mathcal{W}_1$ ($\downarrow$) | $\mathcal{W}_2$ ($\downarrow$) | RBF-MMD ($\downarrow$) | $\mathcal{W}_1$ ($\downarrow$) | $\mathcal{W}_2$ ($\downarrow$) |
| **Base Dynamics** | $0.938_{\pm 0.001}$ | $7.637_{\pm 0.005}$ | $7.653_{\pm 0.006}$ | $0.900_{\pm 0.000}$ | $7.766_{\pm 0.009}$ | $7.877_{\pm 0.009}$ | $0.754_{\pm 0.001}$ | $1.201_{\pm 0.009}$ | $1.455_{\pm 0.012}$ |
| **EntangledSBM w/o Velocity Conditioning** | $0.449_{\pm 0.000}$ | $1.506_{\pm 0.005}$ | $1.544_{\pm 0.005}$ | $0.476_{\pm 0.000}$ | $2.116_{\pm 0.005}$ | $2.197_{\pm 0.005}$ | $0.480_{\pm 0.000}$ | $0.505_{\pm 0.004}$ | $0.627_{\pm 0.005}$ |
| **EntangledSBM w/ LV** | $0.308_{\pm 0.000}$ | $0.175_{\pm 0.002}$ | $0.256_{\pm 0.001}$ | $0.302_{\pm 0.000}$ | $0.340_{\pm 0.001}$ | $0.565_{\pm 0.000}$ | $0.312_{\pm 0.000}$ | $0.198_{\pm 0.001}$ | $0.321_{\pm 0.001}$ |
| **EntangledSBM w/ CE** | $0.428_{\pm 0.000}$ | $0.392_{\pm 0.005}$ | $0.434_{\pm 0.006}$ | $0.409_{\pm 0.000}$ | $0.453_{\pm 0.008}$ | $0.561_{\pm 0.009}$ | $0.451_{\pm 0.000}$ | $0.394_{\pm 0.003}$ | $0.469_{\pm 0.004}$ |

# G    Algorithms

Here, we provide the pseudocode for training (Alg 2) and inference (Alg 3) with EntangledSBM.

---

**Algorithm 2** EntangledSBM **Training**

---

1: **Input:** Parameterized networks $\alpha_\theta(\boldsymbol{R}_t, \boldsymbol{V}_t) : \mathbb{R}^{n \times d} \times \mathbb{R}^{n \times d} \to \mathbb{R}^n$ and $\boldsymbol{h}_\theta(\boldsymbol{R}_t, \boldsymbol{V}_t) : \mathbb{R}^{n \times d} \times \mathbb{R}^{n \times d} \to \mathbb{R}^{n \times d}$, potential energy function $U(\boldsymbol{R}_t) : \mathbb{R}^{n \times d} \to \mathbb{R}$, distribution of target states $\pi_\mathcal{B}$, buffer size $|\mathcal{R}|$, batch size $N_{\text{batch}}$, number of samples $M$, number of rollouts $N_{\text{rollouts}}$, training steps per rollout $N_{\text{steps}}$, number of timesteps $T$
2: $\Delta t \leftarrow \frac{1}{T}$
3: $\mathcal{R} \leftarrow \{\}$                                                     ▷ *Initialize empty replay buffer*
4: **for** rollout in $1, \ldots, N_{\text{rollouts}}$ **do**
5:     **for** $t$ in $1, \ldots, T$ **do**
6:         Predict $\alpha_\theta^i(\boldsymbol{R}_t, \boldsymbol{V}_t) \in \mathbb{R}$ and $\boldsymbol{h}_\theta^i(\boldsymbol{R}_t, \boldsymbol{V}_t) \in \mathbb{R}^d$ with parameterized neural network where $\alpha_\theta^i(\boldsymbol{R}_t, \boldsymbol{V}_t) \geq 0$ is enforced with `softplus` activation
7:         $\boldsymbol{s}_i \leftarrow \nabla_{\boldsymbol{r}_t^i} \log \pi_\mathcal{B}, \hat{\boldsymbol{s}}_i \leftarrow \boldsymbol{s}_i / \|\boldsymbol{s}_i\|$
8:         Compute bias force

$$\boldsymbol{b}_\theta^i(\boldsymbol{R}_t, \boldsymbol{V}_t) \leftarrow \alpha_\theta^i(\boldsymbol{R}_t, \boldsymbol{V}_t)\hat{\boldsymbol{s}}_i + \left(\boldsymbol{I} - \hat{\boldsymbol{s}}_i \hat{\boldsymbol{s}}_i^\top\right) \boldsymbol{h}_\theta^i(\boldsymbol{R}_t, \boldsymbol{V}_t)$$

9:         Generate $M$ discrete trajectories $\{\boldsymbol{X}_{0:T}\}_{j=1}^M$ with current bias $\boldsymbol{b}_\theta^i$ following (6) with Euler-Maruyama integration for each particle $i$

$$\boldsymbol{r}_{t+1}^i = \boldsymbol{r}_t^i + \boldsymbol{v}_t^i(\boldsymbol{R}_t)\Delta t + \boldsymbol{\Sigma}\boldsymbol{b}_\theta^i(\boldsymbol{R}_t, \boldsymbol{V}_t)\Delta t + \boldsymbol{\Sigma}\boldsymbol{\epsilon}_t$$

10:     **end for**
11:     $\mathcal{B} \leftarrow \mathcal{B} \cup \{\boldsymbol{X}_{0:T}\}_{j=1}^M$                              ▷ *update buffer with discrete trajectories*
12:     **for** step in $1, \ldots, N_{\text{steps}}$ **do**
13:         Sample batch $\{\boldsymbol{X}_{0:T}\}_{j=1}^{N_{\text{batch}}}$ from buffer $\mathcal{R}$
14:         Compute cross-entropy objective $\mathcal{L}_{\text{CE}}$ with (13)
15:         Update $\theta$ with $\nabla_\theta \mathcal{L}_{\text{CE}}$
16:     **end for**
17: **end for**
18: **return** parameterized $\alpha_\theta^i(\boldsymbol{R}_t, \boldsymbol{V}_t)$ and $\boldsymbol{h}_\theta^i(\boldsymbol{R}_t, \boldsymbol{V}_t)$

---

---

**Algorithm 3** EntangledSBM **Inference**

---

1: **Input:** Trained networks $\alpha_\theta^i(\boldsymbol{R}_t, \boldsymbol{V}_t)$ and $\boldsymbol{h}_\theta^i(\boldsymbol{R}_t, \boldsymbol{V}_t)$, potential energy function $U(\boldsymbol{R}_t)$, initial state $\boldsymbol{X}_0 = (\boldsymbol{R}_0, \boldsymbol{V}_0)$, target state $\boldsymbol{X}_T = (\boldsymbol{R}_T, \boldsymbol{V}_T)$, time steps $T$, friction $\gamma$
2: $\Delta t \leftarrow \frac{1}{T}, \boldsymbol{R}_t \leftarrow \boldsymbol{R}_0, \boldsymbol{V}_t \leftarrow \boldsymbol{V}_0$
3: $\mathcal{P} \leftarrow \{\}$                                                       ▷ *initialize path*
4: **for** $t$ in $0, \ldots, T$ **do**
5:     Predict $\alpha_\theta^i(\boldsymbol{R}_t, \boldsymbol{V}_t)$ and $\boldsymbol{h}_\theta^i(\boldsymbol{R}_t, \boldsymbol{V}_t)$ with parameterized neural network where $\alpha_\theta^i(\boldsymbol{R}_t, \boldsymbol{V}_t) \geq 0$ is enforced with `softplus` activation
6:     $\boldsymbol{s}_i \leftarrow \nabla_{\boldsymbol{r}_t^i} \log \pi_\mathcal{B}, \hat{\boldsymbol{s}}_i \leftarrow \boldsymbol{s}_i / \|\boldsymbol{s}_i\|$
7:     Compute bias force

$$\boldsymbol{b}_\theta^i(\boldsymbol{R}_t, \boldsymbol{V}_t) \leftarrow \alpha_\theta^i(\boldsymbol{R}_t, \boldsymbol{V}_t)\hat{\boldsymbol{s}}_i + \left(\boldsymbol{I} - \hat{\boldsymbol{s}}_i \hat{\boldsymbol{s}}_i^\top\right) \boldsymbol{h}_\theta^i(\boldsymbol{R}_t, \boldsymbol{V}_t)$$

8:     Generate $M$ discrete trajectories $\{\boldsymbol{X}_{0:T}\}_{j=1}^M$ with current bias $\boldsymbol{b}_\theta^i$ following (6) with Euler-Maruyama integration for each particle $i$

$$\boldsymbol{r}_{t+1}^i = \boldsymbol{r}_t^i + \boldsymbol{v}_t^i(\boldsymbol{R}_t)\Delta t + \boldsymbol{\Sigma}\boldsymbol{b}_\theta^i(\boldsymbol{R}_t, \boldsymbol{V}_t)\Delta t + \boldsymbol{\Sigma}\boldsymbol{\epsilon}_t$$

9:     Append to path $\mathcal{P} \leftarrow \mathcal{P} \cup \{\boldsymbol{R}_t\}$
10: **end for**
11: **return** path $\mathcal{P}$

---

# H  USE OF LARGE LANGUAGE MODELS (LLMS)

We acknowledge the use of large language models (LLMs) to assist in polishing and editing parts of this manuscript. LLMs were used to refine phrasing, improve clarity, and ensure consistency of style across sections. All technical content, experiments, analyses, and conclusions were developed by the authors, with LLM support limited to language refinement and editorial improvements.

