# OpenReview forum: "Entangled Schrödinger Bridge Matching"
_ICLR.cc/2026/Conference — Submitted to ICLR 2026_

### Official Review · Reviewer_6fMu · 2025-10-17

**Soundness:** 3
**Presentation:** 3
**Contribution:** 3
**Rating:** 6
**Confidence:** 3

**Summary:**

This work extends Schrödinger Bridge Matching with a bias force that depends on the velocity of all particles (instead of just one) and is therefore “entangled”. The bias force is designed to always point “towards the data distribution” (i.e. have positive dot product with the score function) and trained via a proposed cross-entropy objective in path space. Experiments on cell perturbation experiments and transition path sampling for fast-folding proteins are presented showing the validity of the method on real-world data.

**Strengths:**

- The paper is overall easy to follow and well-presented/well-written
- Experiments are on two different real-world data sets and show that the methods can solve real-world tasks.
- The parameterizations of the bias force are sound and novel.
- All statements are supported by clear derivations and proofs.

**Weaknesses:**

- The experimental results are limited and benchmarks against prior methods are lacking. If this was not possible, it would be good to highlight why such comparisons were not done to make it understandable to the reader.
- The motivation of the work could be presented more clearly. While it is clear that the entangled bias force is a mathematical possibility, it would be great to have a clearer description of why we would want that. In short, what is the motivation behind introducing this? SBM does not learn any physical dynamics of interacting particles (this is purely data-driven), so it is not really clear to me what the motivation is. It would be great if this could be carved out more clearly.

**Questions:**

- Cell perturbation experiments: How do the interaction of cells represented by the model represent actual biological of cells in that data set? A random subset of n=10 cells is chosen. But how would you know that these cells have interacted in the physical/biological system? To my knowledge, you cannot. Therefore, it is not clear to me whether you can really speak here of an interacting particle system. I would appreciate a clarification, as I acknowledge that this could be limited by my understanding of the dataset.

- Line 63: It is defined how R_t and r_t^i are related (it is implicitly clear but this should be explicitly stated)
- In Proposition 4.1. The initial distribution is suddenly assumed to be a delta function. That should be stated as an assumption.
- Line 226: Kabasch -> Kabsch algorithm

---

> ### Author Response · Authors · 2025-11-16
> **Author Response to Reviewer (1/2)**
>
> **Dear Reviewer 6fMu,**
>
> Thank you for taking the time to carefully review our paper. We are encouraged by your positive evaluation of the presentation, novelty, and real-world applicability of EntangledSBM. Please see the general response comment above and our revised manuscript to view the key changes that we have made to improve the manuscript. Here, we highlight the changes that address each of your remaining concerns one by one:
>
> ---
>
> ### **Addressing Weaknesses**
>
> **W1: The experimental results are limited and benchmarks against prior methods are lacking.**
>
> Since the submission of our paper, we have extended the scale of our experiments to larger fast-folding proteins, including **BBA,** containing 28 amino acids and **504 atoms**. We also added **quantitative comparisons** against non-ML and ML baselines, including unbiased MD (UMD), steered MD (SMD), PIPS [1], and TPS-DPS [2]. **Notably, we show superior target hit percentage for all proteins against all baselines.** We have presented these results in Table 2 of the revised paper and below for your reference.
>
> **Table 2: Transition path sampling benchmarks with EntangledSBM.** Best values are **bolded**. All metrics are averaged over 64 paths. Unless specified in brackets, paths are generated at 300K for Alanine Dipeptide and Chignolin and 400K for the others. Hyperparameters and evaluation metrics are detailed in Appendix C.† denotes values from Seong et al. (2025) [2].
>
> |  |  | Alanine Dipeptide |  |  |  | Chignolin |  |
> | --- | --- | --- | --- | --- | --- | --- | --- |
> | **Method** | **RMSD (↓)** | **THP (↑)** | **ETS (↓)** | **Method** | **RMSD (↓)** | **THP (↑)** | **ETS (↓)** |
> | UMD † | 1.19 ± 0.32 | 6.25 | – | UMD † | 7.23 ± 0.93 | 1.56 | 388.17 |
> | SMD (10K) † | 0.86 ± 0.21 | 7.81 | 812.47 ± 148.80 | SMD (10K) † | 1.26 ± 0.31 | 6.25 | −527.95 ± 93.58 |
> | SMD (20K) † | 0.56 ± 0.27 | 54.69 | 78.40 ± 12.76 | SMD (20K) † | **0.85 ± 0.24** | 34.38 | 179.52 ± 138.87 |
> | PIPS † | 0.66 ± 0.15 | 43.75 | 28.17 ± 10.86 | PIPS † | 4.66 ± 0.17 | 0.00 | – |
> | TPS-DPS † | 0.25 ± 0.20 | 76.00 | **22.79 ± 13.57** | TPS-DPS † | 1.17 ± 0.66 | 59.38 | **−780.18 ± 216.93** |
> | **EntangledSBM** | **0.18 ± 0.07** | **92.19** | 47.91 ± 22.76 | **EntangledSBM** | 0.92 ± 0.13 | **64.06** | 2825.61 ± 318.94 |
>
> |  |  | Trp-cage |  |  |  | BBA |  |
> | --- | --- | --- | --- | --- | --- | --- | --- |
> | **Method** | **RMSD (↓)** | **THP (↑)** | **ETS (↓)** | **Method** | **RMSD (↓)** | **THP (↑)** | **ETS (↓)** |
> | UMD † | 8.27 ± 1.13 | 0.00 | – | UMD † | 10.81 ± 1.05 | 0.00 | – |
> | SMD (10K) † | 1.68 ± 0.23 | 3.12 | −312.54 ± 20.67 | SMD (10K) † | 2.89 ± 0.32 | 0.00 | – |
> | SMD (20K) † | 1.20 ± 0.20 | 42.19 | −226.40 ± 85.59 | SMD (20K) † | 1.66 ± 0.30 | 26.56 | −3104.95 ± 97.57 |
> | PIPS† | 7.47 ± 0.19 | 0.00 | – | PIPS (Force) † | 9.84 ± 0.18 | 0.00 | – |
> | TPS-DPS† | **0.76 ± 0.12** | 81.25 | **−317.61 ± 140.89** | TPS-DPS † | 1.21 ± 0.09 | 84.38 | **−3801.68 ± 139.38** |
> | **EntangledSBM** | 1.04 ± 0.22 | **82.81** | 765.74 ± 155.28 | **EntangledSBM** | **0.84 ± 0.08** | **96.88** | 1453.80 ± 367.84 |
>
> ---
>
> **W2: The motivation of the work could be presented more clearly.**
>
> In Section 3 of the updated manuscript, we have clearly stated the motivation behind our work before formally defining the **EntangledSB problem**. We further provide a **theoretical justification for the entangled bias force in Appendix B.2**.
>
> While the solution to the optimal Schrödinger Bridge can be defined as the optimal evolution of **independent particles** from the initial to target distribution that minimize the divergence from the base dynamics, for systems that follow Langevin dynamics with a defined potential energy function that depends on the joint positions of the particles in the system, there are inherent interactions between particles that govern the evolution along the energy landscape.
>
> The challenge is that many of these interactions are elusive even to experts, which motivates the parameterization of the entangled bias force that implicitly learns these interactions through solving the EntangledSB problem.

---

> ### Author Response · Authors · 2025-11-16
> **Author Response to Reviewer (2/2)**
>
> ### **Addressing Questions**
>
> **Q1: Cell perturbation experiments: How do the interactions of cells represented by the model represent actual biological processes of cells in that data set?**
>
> While it is not guaranteed that the random subset of cells interact in the dataset, it has been widely studied that cell state transition dynamics are governed by cell-cell interactions via signalling pathways **[3, 4]**. By training on simulated paths from $B$ batches of cell clusters, each with $n$ cells, we can implicitly capture the interaction dynamics between cells that prompt their intermediate trajectory to the target perturbed distribution.
>
> ---
>
> **Q2: It is defined how $\boldsymbol{R}_t$ and $\boldsymbol{r}_t^i$ are related (it is implicitly clear but this should be explicitly stated)**
>
> We have updated the description of Langevin dynamics to include the following:
>
> > A time-evolving $n$-particle molecular system can be represented as $\boldsymbol{X}_t=(\boldsymbol{R}_t, \boldsymbol{V}_t)$, where $\boldsymbol{R}_t=\\{\boldsymbol{r}^i_t\in \mathbb{R}^{d}\\}\_{i=1}^n$ denotes the set of coordinates $\boldsymbol{V}_t=\\{\boldsymbol{v}^i_t\in \mathbb{R}^{d}\\}\_{i=1}^n$ denotes the set of velocities of each particle $i$.
> >
>
> ---
>
> **Q3: In Proposition 4.1. The initial distribution is suddenly assumed to be a delta function. That should be stated as an assumption.**
>
> We have updated Prop 4.1 to read “*some target state*  $\boldsymbol{R}\_\mathcal{B}\in  \pi\_{\mathcal{B}}$” instead of defining $\pi\_{\mathcal{B}}$ as the delta function, which is also reflected in our proof in Appendix C.2:
>
> > **Proposition 4.1 (Non-Increasing Distance from Target Distribution):**
> >
> >
> > For small enough $\Delta t$, the distance from some target state $\boldsymbol{R}\_\mathcal{B}\in  \pi\_{\mathcal{B}}$ state after an update with the bias force $\boldsymbol{b}\_\theta^i(\boldsymbol{R}_t,\boldsymbol{V}_t)$ defined in (10) is non-increasing, such that
> >
> > $\exists \boldsymbol{R}\_\mathcal{B}\in \pi\_\mathcal{B}\quad\text{s.t.}\quad\\|\boldsymbol{R}\_{t+\Delta t}-\boldsymbol{R}_\mathcal{B}\\| \leq \\|\boldsymbol{R}\_{t}-\boldsymbol{R}\_\mathcal{B}\\|$
> >
> > where $\boldsymbol{R}\_{t+\Delta t}=\boldsymbol{R}\_{t}+(\boldsymbol{b}\_\theta^i(\boldsymbol{R}_t, \boldsymbol{V}_t)/m_i)\Delta t$.
> >
>
> ---
>
> **Q4: Line 226: Kabasch -> Kabsch algorithm**
>
> Thank you for bringing up this error. Since our original submission, we have rigorously refined the grammar and wording to ensure a coherent reading experience.
>
> ---
>
> **Citations**
>
> [1] Holdijk, Lars, et al. "Stochastic optimal control for collective variable free sampling of molecular transition paths." *Advances in Neural Information Processing Systems* 36 (2023): 79540-79556.
>
> [2] Seong, Kiyoung, et al. "Transition Path Sampling with Improved Off-Policy Training of Diffusion Path Samplers." *ICLR 2025* (2025).
>
> [3] Almet, Axel A., et al. "Inferring pattern-driving intercellular flows from single-cell and spatial transcriptomics." *Nature Methods* 21.10 (2024): 1806-1817.
>
> [4] Cang, Zixuan, et al. "Screening cell–cell communication in spatial transcriptomics via collective optimal transport." *Nature methods* 20.2 (2023): 218-228.
>
> ---
>
> If we have sufficiently addressed all your concerns, we kindly ask that you support the acceptance of our paper for poster presentation, and if concerns remain, we are happy to continue the discussion!
>
> Thank you,
>
> The Authors

---

> > ### Comment · Reviewer_6fMu · 2025-11-22
> >
> > Thank you for addressing my concerns in your revised manuscript and providing answers on my questions. I read the revised version and appreciate taking my concerns into account. As I stated above, I consider this paper above the acceptance threshold and a good contribution.

---

> > > ### Author Response · Authors · 2025-11-22
> > >
> > > We deeply appreciate your positive evaluation of our paper, and we are happy to hear that we have addressed your concerns! Your comments have been very helpful in improving our paper.

---

### Official Review · Reviewer_GHiq · 2025-10-29

**Soundness:** 2
**Presentation:** 2
**Contribution:** 3
**Rating:** 2
**Confidence:** 2

**Summary:**

The authors extended Schrodinger bridge problem when the particles are not independent but interacting and following interacted Langevin dynamics. The authors then test their algorithm in single cell sequencing and molecular dynamics examples.

**Strengths:**

The paper targeted on an important problem of considering SB problems when particles are not independent samples anymore but are interacting. The method is designed to more aligned with momentum SB type of methods rather than handling over-damped dynamics.

**Weaknesses:**

- The presentation on setups and notations is hard to follow when samples are from interacting particles.
    - The current score is mostly driven by this. I am open to change if this is clarified. See questions.
- The theoretical results are less careful in handling position part which has no diffusion on it.
- In experiments the authors seem did not compare to other baselines, e.g., vanilla SB.

Note: I tried to reread the paper several times with fresh eyes but I am really having a hard time understanding the setup hence my comments beyond setups and proof questions are less reliable. I am quite familiar with SB literature but I think the authors are more from physics side that contributed to my struggle. I am more than happy to be convinced that my current score is due to lack of understanding rather than issues of the work.

**Questions:**

## Setup
Overall I don't think I fully understood the setup, mostly on what should be considered a sample and if there is any independent samples. This can be well clarified if the author can provide an example in the setup.

- I am a bit confused by the notation $\mathbf{X}\_{t}$. Do we have multiple observation of  $\mathbf{X}\_{t}$ so that information about $\pi\_{\mathcal{A}}$ is available or we only have one observation $\mathbf{X}\_{t}$?

- When we are thinking two marginals, what are we having? Do we have 1) an ensemble of these multiple particle systems, but they are different ensembles at two ends, but we observe all particles within each ensemble? or 2) we only have one ensemble and at each end we see different part of it?

- Do we need to assume at the beginning and the end samples have same number of particles and they are all interacting?

- Do we know $\pi_{\mathcal{A}}$ and $\pi_{\mathcal{B}}$ or we only have sample to some extend.

- How is the setup different from an augmented vanilla SB whose state space is the concatenation of all particles?

- Use MD as an example, is the author saying all the molecules are interacting or the measured angle are interacting? Are there multiple molecules in the sample?

## Notation
- In section 2, the authors used some convention from physics like Boltzmann constant $k_B$ temperature $\tau$ and mass $m_i$. These are obvious for people with some level of physics background but less obvious for people in ML and it will be nice to briefly explain, or just absorb things into constants like $\gamma$.

## Proofs
- Using Girsanov: there are some conditions needed that are not spelled out. More importantly, the diffusion matrix $\Sigma$ on the position part are 0 since this is a Langevin rather than over-damped one, this makes $\Sigma$ not invertible. The conclusions *are* correct since the drift on locations is known but one need to be careful about these dynamics.

---

> ### Author Response · Authors · 2025-11-16
> **Author Response to Reviewer (1/3)**
>
> **Dear Reviewer GHiq,**
>
> Thank you for taking the time to carefully review our paper. Please see the **general response** comment above and our revised manuscript to view the **key changes** that we have made to improve the manuscript. Here, we highlight the changes that address each of your concerns one by one:
>
> ---
> ### **Addressing Weaknesses**
> **W1: The presentation on setups and notations is hard to follow when samples are from interacting particles.**
>
> Since the initial submission, we have carefully reviewed the manuscript to improve clarity and readability. The revised version ensures that the preliminaries and setup are clear and easy to follow, and we have addressed your specific questions regarding notation in the “**Addressing Questions**” section below. We appreciate your feedback and believe these clarifications resolve the noted ambiguities.
>
> ---
> **W2: The theoretical results are less careful in handling the position part, which has no diffusion on it.**
>
> To resolve your concern, we have added **Remark B.1** to the introduction of Girsanov’s theorem for Langevin dynamics, which directly clarifies how to correctly apply Girsanov’s theorem in the case of underdamped dynamics:
>
> > **Remark B.1:** In the underdamped Langevin model, the diffusion acts only on the velocity coordinates, so the full diffusion matrix on $\boldsymbol{X}_t=(\boldsymbol{R}_t, \boldsymbol{V}_t)$ is degenerate, i.e., $\boldsymbol{\Sigma}=\begin{bmatrix}
> 0&0\\
> 0&\boldsymbol{\Sigma}_V
> \end{bmatrix}$. We therefore apply Girsanov’s theorem on the velocity SDE only, where the diffusion $\boldsymbol{\Sigma}_V$ is invertible, and treat positions as deterministic integrals of velocity (i.e., $\boldsymbol{R}_t=\boldsymbol{R}_0+\int_0^t\boldsymbol{V}_sds$). Therefore, Girsanov's theorem is first defined on velocity paths as
> >
> >
> > $\log\frac{\mathrm{d}\mathbb{P}_V^b}{\mathrm{d}\mathbb{P}_V^0}(\boldsymbol{X}\_{0:T})=\int_0^T\boldsymbol{b}(\boldsymbol{X}_t,t)^\top d\boldsymbol{W}_t-\frac{1}{2}\int_0^T\\|\boldsymbol{b}(\boldsymbol{X}_t,t)\\|^2dt$
> >
> > and pushed forward to the full path measure via $(\boldsymbol{R}_t, \boldsymbol{V}_t)=(\boldsymbol{R}_0+\int_0^t\boldsymbol{V}_sds, \boldsymbol{V}_t)$. For simplicity in notation, we define $\boldsymbol{\Sigma}^{-1}\equiv\boldsymbol{\Sigma}^{-1}_V$ in the theoretical proofs.
> >

---

> ### Author Response · Authors · 2025-11-16
> **Author Response to Reviewer (2/3)**
>
> **W3: In experiments, the authors did not compare to other baselines, e.g., vanilla SB.**
>
> Since the submission of our paper, we have extended the scale of our experiments to larger fast-folding proteins, including **BBA,** containing 28 amino acids and **504 atoms**. We also added **quantitative comparisons** against non-ML and ML baselines, including unbiased MD (UMD), steered MD (SMD), PIPS [1], and TPS-DPS [2]. **Notably, we show superior target hit percentage for all proteins against all baselines.** We have presented these results in Table 2 of the revised paper and below for your reference.
>
> **Table 2: Transition path sampling benchmarks with EntangledSBM.** Best values are **bolded**. All metrics are averaged over 64 paths. Unless specified in brackets, paths are generated at 300K for Alanine Dipeptide and Chignolin and 400K for the others. Hyperparameters and evaluation metrics are detailed in Appendix C.† denotes values from Seong et al. (2025) [2].
>
> |  |  | Alanine Dipeptide |  |  |  | Chignolin |  |
> | --- | --- | --- | --- | --- | --- | --- | --- |
> | **Method** | **RMSD (↓)** | **THP (↑)** | **ETS (↓)** | **Method** | **RMSD (↓)** | **THP (↑)** | **ETS (↓)** |
> | UMD † | 1.19 ± 0.32 | 6.25 | – | UMD † | 7.23 ± 0.93 | 1.56 | 388.17 |
> | SMD (10K) † | 0.86 ± 0.21 | 7.81 | 812.47 ± 148.80 | SMD (10K) † | 1.26 ± 0.31 | 6.25 | −527.95 ± 93.58 |
> | SMD (20K) † | 0.56 ± 0.27 | 54.69 | 78.40 ± 12.76 | SMD (20K) † | **0.85 ± 0.24** | 34.38 | 179.52 ± 138.87 |
> | PIPS † | 0.66 ± 0.15 | 43.75 | 28.17 ± 10.86 | PIPS † | 4.66 ± 0.17 | 0.00 | – |
> | TPS-DPS † | 0.25 ± 0.20 | 76.00 | **22.79 ± 13.57** | TPS-DPS † | 1.17 ± 0.66 | 59.38 | **−780.18 ± 216.93** |
> | **EntangledSBM** | **0.18 ± 0.07** | **92.19** | 47.91 ± 22.76 | **EntangledSBM** | 0.92 ± 0.13 | **64.06** | 2825.61 ± 318.94 |
>
> |  |  | Trp-cage |  |  |  | BBA |  |
> | --- | --- | --- | --- | --- | --- | --- | --- |
> | **Method** | **RMSD (↓)** | **THP (↑)** | **ETS (↓)** | **Method** | **RMSD (↓)** | **THP (↑)** | **ETS (↓)** |
> | UMD † | 8.27 ± 1.13 | 0.00 | – | UMD † | 10.81 ± 1.05 | 0.00 | – |
> | SMD (10K) † | 1.68 ± 0.23 | 3.12 | −312.54 ± 20.67 | SMD (10K) † | 2.89 ± 0.32 | 0.00 | – |
> | SMD (20K) † | 1.20 ± 0.20 | 42.19 | −226.40 ± 85.59 | SMD (20K) † | 1.66 ± 0.30 | 26.56 | −3104.95 ± 97.57 |
> | PIPS† | 7.47 ± 0.19 | 0.00 | – | PIPS (Force) † | 9.84 ± 0.18 | 0.00 | – |
> | TPS-DPS† | **0.76 ± 0.12** | 81.25 | **−317.61 ± 140.89** | TPS-DPS † | 1.21 ± 0.09 | 84.38 | **−3801.68 ± 139.38** |
> | **EntangledSBM** | 1.04 ± 0.22 | **82.81** | 765.74 ± 155.28 | **EntangledSBM** | **0.84 ± 0.08** | **96.88** | 1453.80 ± 367.84 |
>
> **In Section 5.1 Table 1**, we show ablations with no velocity conditioning (analogous to vanilla SB) to show that our parameterization performs much better at both generating the intermediate trajectory along the potential energy landscape and reconstructing the target state or distribution.
>
> ---
>
> ### **Addressing Questions**
>
> **Q1: Do we have multiple observations of $\boldsymbol{X}_t$ so that information about $\pi\_{\mathcal{A}}$ is available or we only have one observation $\boldsymbol{X}_t$?**
>
> To clarify $\boldsymbol{X}_t:=(\boldsymbol{R}_t, \boldsymbol{V}_t)$ denotes the position $\boldsymbol{R}_t$ and velocity $\boldsymbol{V}_t$ of the $n$-particle system at time $t$.
>
> **For transition path sampling,** $\pi\_\mathcal{A}$ is the Dirac delta at a defined initial state $\boldsymbol{R}\_{\mathcal{A}}$, so we have one observation of the initial state, which is simulated by many stochastic trajectories $(\boldsymbol{X}_t)\_{t\in [0,1]}$ to approximate the optimal bridge measure $\mathbb{P}^\star$.
>
> For **cell cluster modelling**, we have a distribution of unperturbed cells $\pi\_\mathcal{A}$ from the dataset, from which we sample a cluster of $n$ cells $\boldsymbol{R}_0\sim \pi\_{\mathcal{A}}$ as the initial state. In this case, we train the bias force to be able to simulate perturbations from any arbitrary cluster of cells sampled from $\pi\_\mathcal{A}$.
>
> We note that we have no prior knowledge of any intermediate states, and they are determined only during simulation, which follows the bias force added to the natural gradient of the potential energy landscape.

---

> ### Author Response · Authors · 2025-11-16
> **Author Response to Reviewer (3/3)**
>
> **Q2: When we are thinking two marginals, what are we having?**
>
> The initial and target marginal is a distribution of observed states of the system. For the **MD experiment**, the initial and target distribution is a Dirac delta at a single initial state $\pi\_\mathcal{A}=\boldsymbol{1}\_{\mathcal{A}}(\boldsymbol{R}\_\mathcal{A})$ and target state  $\pi\_\mathcal{B}=\boldsymbol{1}\_{\mathcal{A}}(\boldsymbol{B}\_\mathcal{B})$ of the same biomolecular system.
>
> For the **cell perturbation modelling experiment**, the initial and target distributions are the observed population of untreated cells and the population of perturbed cells in gene expression space. In the cell modelling experiment, we sample random clusters from the control and perturbed distributions to capture the dynamics that map untreated cells to perturbed cells.
>
> ---
>
> **Q3: Do we need to assume at the beginning and the end samples have the same number of particles and they are all interacting?**
>
> **Yes, we assume that the number of particles is conserved throughout the trajectory, which is the case for most biomolecular systems with fixed atomistic composition** (e.g., peptides, proteins, and small molecules) and cell clusters.
>
> We do not need to assume that the particles are all interacting, as the parameterization naturally learns the interactions and dynamic dependencies between the pairwise positions and velocities in the system that approximate the optimal bridge measure (minimizing the potential energy of the path and maximizing the terminal reward defined by the alignment with the target state).
>
> ---
>
> **Q4: Do we know $\pi_\mathcal{A}$ and $\pi_\mathcal{B}$ or we only have sample to some extent.**
>
> **See our answer to Q2 above.**
>
> ---
>
> **Q5: How is the setup different from an augmented vanilla SB whose state space is the concatenation of all particles?**
>
> Unlike a vanilla SB that treats the concatenated state as a single high-dimensional feature with independent dynamics, **EntangledSBM introduces structured interactions via the second-order dynamic bias force and Transformer architecture**. Each particle’s bias force explicitly depends on the positions and velocities of *all* other particles **as they evolve temporally**, enabling dynamic coupling rather than factorized or mean-field evolution.
>
> ---
>
> **Q6: Use MD as an example, is the author saying all the molecules are interacting or the measured angles are interacting? Are there multiple molecules in the sample?**
>
> In the MD experiments, each system represents a **single molecule** modelled at the **all-atom resolution**, where atoms (particles) evolve via the potential energy function $U(\boldsymbol{R}_t)$. The entangled bias force adds a learned correction that captures collective coupling between atomic velocities, which captures the **influence of all system interactions on the trajectory of each particle**, not just the angle interactions.
>
> ---
>
> **Q7: In section 2, the authors used some convention from physics […]**
>
> Thank you for bringing this up. We have **clarified the physics notation in Section 2 as follows:**
>
> > where $m_i$ is the mass of each particle, $\gamma$ is the friction coefficient, $k_B$ is the Boltzmann constant, $\tau$ is the temperature, and $d\boldsymbol{W}_t$ is standard Brownian motion
> >
>
> In the proofs, we absorb the physics variables into the diffusion coefficient $\boldsymbol{\Sigma}d\boldsymbol{W}_t$ for simplicity of notation and for easier reading experience to those with non-physics background.
>
> ---
>
> **Q8: Using Girsanov, there are some conditions needed that are not spelled out.**
>
> **See our response to W2 above.**
>
> ---
>
> **Citation:**
>
> [1] Holdijk, Lars, et al. "Stochastic optimal control for collective variable free sampling of molecular transition paths." *Advances in Neural Information Processing Systems* 36 (2023): 79540-79556.
>
> [2] Seong, Kiyoung, et al. "Transition Path Sampling with Improved Off-Policy Training of Diffusion Path Samplers." *ICLR 2025* (2025).
>
> ---
>
> If we have sufficiently addressed all your concerns, we kindly ask that you support the acceptance of our paper for poster presentation, and if concerns remain, we are happy to continue the discussion!
>
> Thank you,
>
> The Authors

---

> > ### Comment · Reviewer_GHiq · 2025-11-23
> >
> > I appreciate the authors' revision and answers especially the part treating Girsanov more carefully addressed my concern there.
> >
> > However I am still struggling in understanding the dependency in the setup. Let me try ask some follow up of Q5 and Q6 that hopefully would clear the confusion.
> >
> >  - MD experiment
> >    - Should I understand that particles are atom and there is only one molecule so the distribution in SB is over these atoms and there is only one molecule thus the distribution is a delta measure at the first observed configuration? And I would argue this is different from single cell case where we observe many independent cells (as well kill them to sequence so early cells cannot interact with later ones) and there is indeed a population of cells. In this example is $\mathbf{r}_t^i$ the state of $i$th atom at time $t$? Is it always the same atoms?
> >
> > -  Difference between the proposed method and vanilla SB with augmented state space
> >    - From the stochastic control view of SB (e.g., [1], eq. 14 and Sec IV) learning a biased forces to the potential can be seen as solving SB with a reference driven by that potential so appears to me the problem can be solved by seeing it as a vanilla SB problem with some inductive bias on the learning target $b$. Is the main difference transformer architecture? Connect back to Q5, if both initial and end in MD experiments are delta measures, we have a reference of potential field, appears to me the SB solution might even be analytical as it reduces to a Brownian bridge via a transform of measure to accommodate the potential field.
> >
> >
> > [1] Caluya, K.F. and Halder, A., 2021. Wasserstein proximal algorithms for the Schrödinger bridge problem: Density control with nonlinear drift. IEEE Transactions on Automatic Control, 67(3), pp.1163-1178.

---

> > > ### Author Response · Authors · 2025-11-24
> > > **Author Response to Follow-Up Questions (1/2)**
> > >
> > > **Dear Reviewer GHiq,**
> > >
> > > We greatly appreciate your deep engagement with our paper and for bringing up these crucial points of discussion! We hope to clear up your remaining concerns by carefully answering each of your remaining questions one by one:
> > >
> > > ---
> > >
> > > > **Should I understand that particles are atom and there is only one molecule so the distribution in SB is over these atoms and there is only one molecule thus the distribution is a delta measure at the first observed configuration? In this example, is $\boldsymbol{r}^i_t$the state of the atom at time $t$? Is it always the same atoms?**
> > > >
> > >
> > > Yes, in the case of MD, the **particles** — with position and velocity at time $t$ denoted **$(\boldsymbol{r}^i_t, \boldsymbol{v}^i_t)$** indexed by $i$ — are **single atoms**, and the **system** — with position and velocity at time $t$ denoted $(\boldsymbol{R}_t, \boldsymbol{V}_t)$ — is the **molecule** (Alanine Dipeptide or fast folding proteins in our experiments). Since the initial state is defined as a single configuration (stored as a PDB file), we define it as the delta distribution at that state.
> > >
> > > > **And I would argue this is different from single cell case where we observe many independent cells (as well kill them to sequence so early cells cannot interact with later ones) and there is indeed a population of cells.**
> > > >
> > >
> > > In the single cell case, the **particles** are **single cells,** and the **system** is a **cluster of cells sampled from a cell population that evolves through cell-cell interactions rather than independently**. While it is true that in single cell analysis, the cells in the first timepoint are not the same cells and cannot interact with the cells in the final timepoint, it is a common assumption that the single cell measurements are **snapshots of the cell dynamics over time**, which is the motivation of many prior work applying optimal transport and flow matching to model cellular dynamics (See for example [1], [2]).
> > >
> > > Following this perspective, we assume that the cells interact dynamically across time via cell-cell signalling, which **guides the evolution** in cell state over time points. Therefore, even though we do not track the same physical cells across time, the population snapshot assumption allows us to treat the observed cells as samples from an underlying **dynamical population process** in which cell–cell signalling shapes the evolution of cellular states, which is standard in the single-cell literature.
> > >
> > > ---
> > >
> > > **Citations:**
> > >
> > > [1] Tong, Alexander, et al. "Trajectorynet: A dynamic optimal transport network for modeling cellular dynamics." *International conference on machine learning*. PMLR, 2020.
> > >
> > > [2] Zhang, Zhenyi, Tiejun Li, and Peijie Zhou. "Learning stochastic dynamics from snapshots through regularized unbalanced optimal transport." (2024).

---

> > > > ### Author Response · Authors · 2025-11-24
> > > > **Author Response to Follow-Up Questions (2/2)**
> > > >
> > > > > **Difference between the proposed method and vanilla SB with augmented state space**
> > > > >
> > > >
> > > > We agree that, in classical Schrödinger bridge settings where the dynamics **follow an elliptic**, **first-order**, **fully diffusive SDE** with a fixed potential and delta endpoint constraints, the optimal bridge can reduce to a generalized Brownian bridge obtained by a Doob transform. However, we emphasize that this analytical simplification **does not hold in our case for the following reasons:**
> > > >
> > > > 1. **We assume particle dynamics are non-factorizable.** This means that we cannot reduce the joint dynamics into independent single-particle paths and diffusion cannot be applied independently on each particle without structured interactions with the other particles. Therefore, we need to define the bias force such that it depends on the collective evolution of particles in the system.
> > > > 2. **The marginal second-order particle position dynamics are non-Markovian.** For the vanilla SB solution to hold, the position is assumed to evolve via a Markov process. However, the second-order Langevin dynamics defined as:
> > > >
> > > >     $$
> > > >     d\boldsymbol{R}_t=\boldsymbol{V}_tdt, \quad d\boldsymbol{V}_t=\frac{-\nabla\_{\boldsymbol{R}_t }U(\boldsymbol{R}_t)}{m}dt-\gamma \boldsymbol{V}_tdt+\sqrt{\frac{2\gamma k_B\tau}{m}}d\boldsymbol{W}_t\tag{$\star$}
> > > >     $$
> > > >
> > > >     The position coordinates evolve via the **second-order dynamics** that depend on **both** the current position $\boldsymbol{R}_t$ and current velocity $\boldsymbol{V}_t$, **not just the current position as in vanilla SB**:
> > > >
> > > >     $$
> > > >     \frac{d^2\boldsymbol{R}_t}{dt^2}=\frac{d\boldsymbol{V}_t}{dt}=\frac{-\nabla U(\boldsymbol{R}_t)}{m}-\gamma \boldsymbol{V}_t+\text{noise}
> > > >     $$
> > > >
> > > >     Our parameterization is naturally suited to model these dynamics by injecting dependence on the velocity $\boldsymbol{V}_t$.
> > > >
> > > > 3. **The underdamped Langevin dynamics are not fully diffusive.** The vanilla “Brownian-bridge with a Doob transform” construction defined in Eq. 14 of [3] assumes a first-order, fully diffusive process written as:
> > > >
> > > > $$
> > > > \inf\_{\boldsymbol{u}\in \mathcal{U}}\mathbb{E}\{\int_0^1\frac{1}{2}\\|\boldsymbol{u}(\boldsymbol{x}, t)]\\|_2^2dt\} \quad\text{s.t.}\quad\begin{cases}d\boldsymbol{x}=\boldsymbol{f}(\boldsymbol{x},t)dt+\boldsymbol{B}(t)\boldsymbol{u}(\boldsymbol{x},t)dt+\sqrt{2\epsilon}\boldsymbol{B}(t)d\boldsymbol{w}(t)\\\\\boldsymbol{x}(t=0)\sim \rho_0(\boldsymbol{x}), \quad \boldsymbol{x}(t=1)\sim \rho_1(\boldsymbol{x})\end{cases}
> > > > $$
> > > >
> > > > which is subject to a **first-order SDE** with full diffusion on **all coordinates** of the state space $\boldsymbol{x}$. In this case, the optimal SB solution can **indeed be defined in closed form with the Doob transform,** given in Eq. (21) of [3] as:
> > > >
> > > > $$\boldsymbol{u}^\star(\boldsymbol{x}, t)=\boldsymbol{B}(t)^\top\nabla\psi(\boldsymbol{x},t)$$
> > > >
> > > > In our case, the reference dynamics is defined as **underdamped Langevin dynamics on the phase space $(\boldsymbol{R}_t, \boldsymbol{V}_t)$ as shown in Eq. ($\star$) above, which injects noise only in the velocity coordinates as you correctly pointed out in your initial comment.** In this case, the optimal control cannot be solved analytically.
> > > >
> > > > Only in very special, **linear–Gaussian, first-order** cases, the optimal SB solution can be solved analytically; however, in most real-world cases with non-linear drift and high dimensional state spaces, or in our case underdamped Langevin dynamics on the phase space $(\boldsymbol{R}_t, \boldsymbol{V}_t)$, there is **no closed-form solution** for the optimal control force. This is the motivation behind ML-based work in approximating SB solutions, such as [4], and the motivation for our present work, which models the **joint phase space dynamics with parameterized bias forces.**
> > > >
> > > >
> > > > > **Is the main difference transformer architecture?**
> > > > >
> > > >
> > > > While the Transformer architecture enables efficient learning of dependencies across particles, the key novelty lies in the **entangled formulation** of the bias force itself and how it **expands the class of representable control fields** and yields strictly greater expressiveness when the optimal SB control is **non-factorizable,** as **justified in App. B.2**. The Transformer is used in practice to capture this dependence by allowing each particle to attend to the positions and velocities of all other particles.
> > > >
> > > > ---
> > > >
> > > > **Citations:**
> > > >
> > > > [3] Caluya, K.F. and Halder, A., 2021. Wasserstein proximal algorithms for the Schrödinger bridge problem: Density control with nonlinear drift. IEEE Transactions on Automatic Control, 67(3), pp.1163-1178.
> > > >
> > > > [4] Liu, Guan-Horng, et al. "Deep generalized schrödinger bridge." *Advances in Neural Information Processing Systems* 35 (2022): 9374-9388.
> > > >
> > > > ---
> > > >
> > > > Thank you for thoughtfully reading our paper and bringing up these important clarifications. We hope we have cleared up your remaining confusion, but we are happy to answer any further questions you may have!

---

> ### Comment · Reviewer_GHiq · 2025-11-25
>
> First let me thank the authors for bearing with me trying to understand the paper. I think it is addressing an important problem, albeit it is a bit different from my current mental picture. I want to emphasis that all of the back and forth are for me to understand the setup better so I can confidently say whether I think this is a good paper or not and I think it will also help other readers with similar background as I do --- so thank again for the authors responses. I am more than happy to change the score after the left over confusions I have got cleaned, we are getting there.
>
> - I think I understand the MD examples better now. To test that let me ask this: Let us focus on the $1$st atom, let's say it is a hydrogen at the N' end of the protein, with position and velocity at time $0$ being $(\mathbf{r}\_{1, 0}, \mathbf{v}\_{1, 0})$. At time $t=1$, will $(\mathbf{r}\_{1, 1}, \mathbf{v}\_{1, 1})$ be the same hydrogen atom? Or the other way to ask is whether the particles are **paired** at time 0 and 1? If they are we are working with the same set of particles (atoms) in the entire time interval and interactions between atom 1 and atom 2 is never ambiguous, there is a fixed number of atoms interaction in the entire time [0, 1] and I think I understand the value of modeling such interactions in SB. I suspect this is the original motivating example of this approach. If I indeed understand it correctly, I am quite convinced this is a realistic application and the approach is natural and seems work well.
>
> - I am more familiar with the single cell setup but this is where I got most confused. Again let us consider the 1st cell in both time 0 and time 1 samples, since we have to kill the cells to see them, the 1st cell in time 0 is *not the same cell* as the 1st cell in time 1, and the trajectory is ended at time 0 for the 1st cell sampled at time 0 so it cannot interact with the 1st cell that is sampled at time 1. Here is where I got confused about the interaction: the notion of cell 1 and cell 2's interaction is ambiguous since there is no consistent enumeration across time, the 1st cell we see in sample time 0 is dead after we see it. It can no longer interact with any cell after it was seen, i.e., it cannot join any interactions in time in (0, 1]. I can see that we can consider the hypothetical case of "what if cell 1 at time 0 is not dead", I am not sure in the proposed method how many cells are interacting, say we sample $N\_0$ cells in time 0 and $N\_1$ cells in time 1, are we modeling $N\_0$ cells interacting, $N\_1$ cell interacting or $N\_0+N\_1$ cell interacting? What if in the petri dish there are more cells we just did not take for measurements, should they join the interactions?

---

> > ### Author Response · Authors · 2025-11-25
> > **Author Response to Additional Questions**
> >
> > Dear Reviewer GHiq,
> >
> > We are more than happy to continue the discussion to ensure you fully understand the setup  of the experiments we chose to validate our proposed framework, and we will carefully address your questions for both the MD and single-cell experiments below:
> >
> > ---
> >
> > ### **MD Experiment**
> >
> > You are completely correct in your interpretation of the MD experiment!
> >
> > > **At time $t=1$, will $(\boldsymbol{r}\_{1,1}, \boldsymbol{v}\_{1,1})$ be the same hydrogen atom? Or the other way to ask is whether the particles are paired at time 0 and 1?**
> > >
> >
> > Yes, among the $n$ atoms indexed $i=\\{1, \dots, n\\}$ are **paired at all times $t\in [0,T]$,** where $t=T$ is the final time point in our notation.
> >
> > You’re correct — the atom indexed $i=1$ with state $(\boldsymbol{r}\_{1,0}, \boldsymbol{v}\_{1,0})$ at $t=0$ **is indeed the same atom with state** $(\boldsymbol{r}\_{1,1}, \boldsymbol{v}\_{1,1})$ at $t=1$. Accordingly, the atom indexed $i=n$ at with state $(\boldsymbol{r}\_{n,0}, \boldsymbol{v}\_{n,0})$ at $t=0$ **is also the same atom with state** $(\boldsymbol{r}\_{n,1}, \boldsymbol{v}\_{n,1})$ at $t=1$.
> >
> > > **If we are working with the same set of particles (atoms) in the entire time interval and interactions between atom 1 and atom 2 is never ambiguous, there is a fixed number of atoms interaction in the entire time [0, 1] and I think I understand the value of modeling such interactions in SB.**
> > >
> >
> > Thank you for acknowledging the value of modelling these paired interactions, and you are fully correct in your interpretation that we are modelling pairwise interactions **among a fixed set of atoms.**
> >
> > > **I suspect this is the original motivating example of this approach.**
> > >
> >
> > You’re again correct! The problem of simulating the transition dynamics for high-dimensional all-atom MD systems over difficult energy landscapes was indeed the **core motivation behind our approach**. As atoms within a molecule are naturally heterogeneous particles that interact via interatomic forces, we sought to develop a framework that accounts for these dynamic interactions to enhance the expressivity and accuracy of ML-based alternatives for costly MD simulations.
> >
> > ---
> > ### **Single-Cell Experiment**
> >
> > We agree that the single-cell setup is more ambiguous and understand your confusion. We hope to clarify them below and provide some context into why we believe our framework is applicable in modelling single-cell dynamics.
> >
> > > **the notion of cell 1 and cell 2's interaction is ambiguous since there is no consistent enumeration across time, the 1st cell we see in sample time 0 is dead after we see it. It can no longer interact with any cell after it was seen, i.e., it cannot join any interactions in time in (0, 1].  I can see that we can consider the hypothetical case of "what if cell 1 at time 0 is not dead”**
> > >
> >
> > We agree that this idea can be confusing, but we assume that the measurements taken at the snapshot following perturbation with a drug is a representation of **the same cell population** if it were hypothetically able to evolve over the time interval. Under this assumption, cell 1 and cell 2 (and every pair of cells in the sample) **can interact over the time interval non-ambiguously and cell 1 at time $t=0$ is the same as cell 1 at time $t=1$.** With current single-cell RNA sequencing techniques and datasets, this does not exactly represent the reality of the cells being measured, but we believe this is a biologically motivated assumption that can help us model the real interactions of perturbed cell populations that are allowed to evolve naturally without destructive measurement.
> >
> > > **I am not sure in the proposed method how many cells are interacting, say we sample $N_0$ cells at time 0 and $N_1$ cells in time 1, are we modeling $N_0$ cells interacting, $N_1$ cell interacting or $N_0+N_1$ cell interacting? What if in the petri dish there are more cells we just did not take for measurements, should they join the interactions?**
> > >
> >
> > In our setup, we assume that the number of cells in the cluster is the same across time, or $N_0=N_1$. So, there are $n=N_0=N_1$ cells interacting over the time interval. We acknowledge that this does not exactly correspond to the reality of cell populations, which can grow and shrink in population size and have a non-constant number of cells interacting over time. This is a simplifying assumption that we make, which assumes that population-level dynamics can be implicitly learned by training the bias forces over many such constant-sized cell clusters sampled from the dataset of unperturbed and perturbed cells.
> >
> > We agree that the single-cell setup is simplifying the reality of interacting cells and believe that future work can leverage the entangled parameterization to learn growth rates that capture the growth and death of cells as they interact within the cluster.
> >
> > ---
> >
> > We really appreciate your efforts to fully understand our paper, and hope our clarifications are helpful!

---

### Official Review · Reviewer_UM1c · 2025-10-30

**Soundness:** 3
**Presentation:** 2
**Contribution:** 3
**Rating:** 4
**Confidence:** 3

**Summary:**

The paper introduces Entangled Schrödinger Bridge Matching (EntangledSBM), a framework designed to model the second-order Langevin dynamics of multi-particle systems with entangled bias forces. The EntangledSBM framework solves the Entangled Schrödinger Bridge (EntangledSB) problem using stochastic optimal control theory and further enables conditional sampling during inference. Experiments on cell cluster dynamics and molecular systems demonstrate the effectiveness of the proposed framework

**Strengths:**

- The paper is well structured and easy to follow.
- The paper provides clear, step-by-step theoretical derivations.

**Weaknesses:**

- **Limited experimental scale.**
  The experiments are restricted to relatively small-scale systems. Specifically, they are limited to small, fast-folding proteins (fewer than 20 amino acids), which are significantly smaller than those encountered in real-world applications. In addition, quantitative comparisons with relevant baselines such as [1, 2, 3] are necessary for the transition path sampling experiments.

- **Lack of efficiency and scalability analysis.**
  The practical training objective in Eqn. (19) depends on the number of simulation steps, which should have a noticeable impact on performance. Additional analyses on training and inference efficiency, as well as scalability, are needed to demonstrate that the proposed framework is computationally efficient.

**Minor issues:**

- There are several writing issues, such as the misspelling of “initial” in Line 93, the redundant “where” in Lemma 1 (Lines 821–826), and the same issue in Eqn. (12), Line 198.
- Eqn. (6) is somewhat confusing, as it appears to be a discrete-time expression while still containing the term $dt$.
- The notation \(X_{0:K}\) in Eqn. (18) has not been introduced previously. It likely represents the discretized trajectory, and additional explanations regarding this discretization should be provided.


[1]. Lars Holdijk, Yuanqi Du, Ferry Hooft, Priyank Jaini, Berend Ensing, and Max Welling. Stochastic optimal control for collective variable free sampling of molecular transition paths. Advances in Neural Information Processing Systems, 36:79540–79556, 2023

[2]. Kiyoung Seong, Seonghyun Park, Seonghwan Kim, Woo Youn Kim, and Sungsoo Ahn. Transition path sampling with improved off-policy training of diffusion path samplers. In The Thirteenth International Conference on Learning Representations, 2025.

[3]. Yuanqi Du, Michael Plainer, Rob Brekelmans, Chenru Duan, Frank Noe, Carla P Gomes, Alan Aspuru-Guzik, and Kirill Neklyudov. Doob’s lagrangian: A sample-efficient variational approach to transition path sampling. Advances in Neural Information Processing Systems, 37:65791–65822, 2024.

**Questions:**

- Could you provide a more detailed comparison with the prior work [2]? It appears that the main differences lie in the use of a cross-entropy (CE) training objective and the replacement of the MLP architecture with a transformer.

---

> ### Author Response · Authors · 2025-11-16
> **Author Response to Reviewer (1/3)**
>
> **Dear Reviewer UM1c,**
>
> Thank you for taking the time to carefully review our paper. Please see the **general response** comment above and our revised manuscript to view the **key changes** that we have made to improve the manuscript. Here, we highlight the changes that address each of your concerns one by one:
>
> ---
> ### Addressing Weaknesses
>
> **W1: Limited experimental scale. The experiments are restricted to relatively small-scale systems. Specifically, they are limited to small, fast-folding proteins (fewer than 20 amino acids), which are significantly smaller than those encountered in real-world applications. In addition, quantitative comparisons with relevant baselines such as [1, 2, 3] are necessary for the transition path sampling experiments.**
>
> Since the submission of our paper, we have extended the scale of our experiments to larger fast-folding proteins, including **BBA,** containing 28 amino acids and **504 atoms**. We have also conducted rigorous experiments and provided **quantitative comparisons** against the baselines that you mention, including **PIPS [1] and TPS-DPS [2], in addition to unbiased MD (UMD) and steered MD (SMD)**, and show superior performance across all benchmarks in Target Hit Percentage (THP). We have presented these results in **Table 2 of the revised paper** and below for your reference.
>
> **Table 2: Transition path sampling benchmarks with EntangledSBM.** Best values are **bolded**. All metrics are averaged over 64 paths. Unless specified in brackets, paths are generated at 300K for Alanine Dipeptide and Chignolin and 400K for the others. Hyperparameters and evaluation metrics are detailed in Appendix C.† denotes values from Seong et al. (2025) [2].
>
> |  |  | Alanine Dipeptide |  |  |  | Chignolin |  |
> | --- | --- | --- | --- | --- | --- | --- | --- |
> | **Method** | **RMSD (↓)** | **THP (↑)** | **ETS (↓)** | **Method** | **RMSD (↓)** | **THP (↑)** | **ETS (↓)** |
> | UMD † | 1.19 ± 0.32 | 6.25 | – | UMD † | 7.23 ± 0.93 | 1.56 | 388.17 |
> | SMD (10K) † | 0.86 ± 0.21 | 7.81 | 812.47 ± 148.80 | SMD (10K) † | 1.26 ± 0.31 | 6.25 | −527.95 ± 93.58 |
> | SMD (20K) † | 0.56 ± 0.27 | 54.69 | 78.40 ± 12.76 | SMD (20K) † | **0.85 ± 0.24** | 34.38 | 179.52 ± 138.87 |
> | PIPS † | 0.66 ± 0.15 | 43.75 | 28.17 ± 10.86 | PIPS † | 4.66 ± 0.17 | 0.00 | – |
> | TPS-DPS † | 0.25 ± 0.20 | 76.00 | **22.79 ± 13.57** | TPS-DPS † | 1.17 ± 0.66 | 59.38 | **−780.18 ± 216.93** |
> | **EntangledSBM** | **0.18 ± 0.07** | **92.19** | 47.91 ± 22.76 | **EntangledSBM** | 0.92 ± 0.13 | **64.06** | 2825.61 ± 318.94 |
>
> |  |  | Trp-cage |  |  |  | BBA |  |
> | --- | --- | --- | --- | --- | --- | --- | --- |
> | **Method** | **RMSD (↓)** | **THP (↑)** | **ETS (↓)** | **Method** | **RMSD (↓)** | **THP (↑)** | **ETS (↓)** |
> | UMD † | 8.27 ± 1.13 | 0.00 | – | UMD † | 10.81 ± 1.05 | 0.00 | – |
> | SMD (10K) † | 1.68 ± 0.23 | 3.12 | −312.54 ± 20.67 | SMD (10K) † | 2.89 ± 0.32 | 0.00 | – |
> | SMD (20K) † | 1.20 ± 0.20 | 42.19 | −226.40 ± 85.59 | SMD (20K) † | 1.66 ± 0.30 | 26.56 | −3104.95 ± 97.57 |
> | PIPS† | 7.47 ± 0.19 | 0.00 | – | PIPS (Force) † | 9.84 ± 0.18 | 0.00 | – |
> | TPS-DPS† | **0.76 ± 0.12** | 81.25 | **−317.61 ± 140.89** | TPS-DPS † | 1.21 ± 0.09 | 84.38 | **−3801.68 ± 139.38** |
> | **EntangledSBM** | 1.04 ± 0.22 | **82.81** | 765.74 ± 155.28 | **EntangledSBM** | **0.84 ± 0.08** | **96.88** | 1453.80 ± 367.84 |
>
> We believe that Doob’s Lagrangian [3] is not comparable to our method as it studies a fundamentally different problem where intermediate states are sampled after rigidly defining the initial state $\boldsymbol{R}\_{0}$ and target state $\boldsymbol{R}_{\mathcal{B}}$, making the THP and RMSD metrics invalid. Furthermore, [3] limited their quantitative evaluation to Alanine Dipeptide, indicating that the method is limited in its scalability to larger, real-world systems.

---

> ### Author Response · Authors · 2025-11-16
> **Author Response to Reviewer (2/3)**
>
> **W2: Lack of efficiency and scalability analysis. The practical training objective in Eqn. (19) depends on the number of simulation steps, which should have a noticeable impact on performance. Additional analyses on training and inference efficiency, as well as scalability, are needed to demonstrate that the proposed framework is computationally efficient.**
>
> While Eq. 19 is computed over the full simulated trajectories, we note that its efficacy is not directly dependent on the number of simulation steps. The number of steps chosen during training would directly determine the number of steps required to reach the target state during inference; however, this number can be selected to simulate coarser trajectories if memory or runtime constraints are a concern.
>
> In addition, the number of simulation steps does not necessarily scale with the size or complexity of the biological system that is being modelled, but rather the precision with which we aim to model the dynamics and the estimated timescale of the biological event (1 ps for Alanine Dipeptide and 5 ps for fast-folding proteins). For other events where the estimated timescale is unknown, such as cell state transitions after perturbation, an arbitrary number of simulation steps can be defined to interpolate between the initial and target distributions.
>
> To further clarify computational efficiency, **Table A below summarizes the average training and sampling times per rollout across molecular dynamics experiments**, demonstrating that the proposed framework scales favourably with increasing system complexity.
>
> **Table A: Training and sampling times in seconds per rollout for molecular dynamics experiments on a single NVIDIA BH200 GPU.**
> | Molecule | Sampling Time (s) | Training Time (s) |
> | --- | --- | --- |
> | **Alanine Dipeptide (22 atoms)** | 31.469 ± 0.155 | 190.115 ± 0.475 |
> | **Chignolin (138 atoms)** | 160.838 ± 0.375 | 437.232 ± 3.484 |
> | **Trp-cage (284 atoms)** | 196.615 ± 7.354 | 946.982 ± 3.244 |
> | **BBA (504 atoms)** | 291.682 ± 2.643 | 1956.893 ± 1.220 |
>
> ---
>
> ### **Addressing Minor Issues**
>
> We appreciate your attention to detail. Since the initial submission, we have carefully reviewed the manuscript to correct grammatical inconsistencies and improve clarity and readability. The revised version addresses all minor issues raised.
>
> ---
>
> **I1: The notation $\boldsymbol{X}_{0:K}$ in Eqn. (18) has not been introduced previously. It likely represents the discretized trajectory, and additional explanations regarding this discretization should be provided.**
>
> We have made the following clarification in the revised version under “**Discrete Time Objective**” of our manuscript:
>
> > Since we want to train on off-policy trajectories from previous iterations to dynamically update the learned bias force given the velocities of the remaining particles, we require storing the simulated trajectories in a discretized form $\boldsymbol{X}\_{0:K}:=(\boldsymbol{X}_k)\_{k\in \{0, \dots, K\}}$ with step size $\Delta t$.
> >

---

> ### Author Response · Authors · 2025-11-16
> **Author Response to Reviewer (3/3)**
>
> ### **Addressing Questions**
>
> **Q1: Could you provide a more detailed comparison with the prior work [2]? It appears that the main differences lie in the use of a cross-entropy (CE) training objective and the replacement of the MLP architecture with a transformer.**
>
> First, we emphasize that the **key difference in our bias force parameterization is the dependence on the full system positions and velocities to capture the interacting dynamics of the system**. We demonstrate the empirical significance of the entangled bias force in Sec. 5.1, where we compare the performance of the bias force with no velocity conditioning to the velocity-dependent bias force (**EntangledSBM**) in reconstructing the target distribution and generalizing to distributions not seen in the dataset.
>
> Our parameterization learns to navigate complex energy landscapes and interactions given a target state as input, enabling generalization to unseen pairs of states, **which is not possible with the approach presented in [2]** as stated explicitly by the authors in the conclusion of their paper:
>
> > “Our method does not generalize across unseen pairs of meta-stable states or different molecular systems. These points to an interesting venue for future research, which would be more appealing for practical applications in drug discovery or material design.” (Seong et al. 2024 [2])
> >
>
> To demonstrate the significance of the **cross-entropy (CE) objective that we formalize in Sec. 4.2**, we provide additional background on the alternative log-variance (LV) objective in App. B.1 and a rigorous evaluation of the theoretical and empirical benefits of the CE objective over the LV objective in **App F, which we repeat below as reference:**
>
> > In this section, we discuss the intuition behind the differences in cell state trajectories generated with the cross-entropy (CE) objective and log-variance (LV) objective for Clonidine, as shown in Fig. 7 and Table 6, and Trametinib, as shown in Fig. 8 and Table 7.
> >
> >
> > We observe that the log-variance (LV) objective **fails to accurately simulate the intermediate dynamics of cells following perturbation and generates nearly straight, abrupt trajectories connecting the initial and terminal populations.** The trajectories generated from the LV-trained bias forces ignore the gradual, curved progression of cell states through intermediate data manifolds, in contrast to the trajectories generated from the cross-entropy (CE)-trained bias forces (Fig. 7, 8). While the LV-objective accurately reconstructs the target cell distribution, it fails to reconstruct the intermediate dynamics that arise from nonlinear coupling between position and velocity fields after perturbation (Tables 6, 7).
> >
> >
> > While the LV objective aims to make the log–likelihood ratio $\log \frac{\mathrm{d}\mathbb{P}^\star}{\mathrm{d}\mathbb{P}^{b\_\theta}}(\boldsymbol{X}\_{0:T})$ *constant in expectation* to minimize the variance, it is easiest to only maximize the terminal reward $r(\boldsymbol{X}_T)$ rather than intermediate path alignment with $\mathbb{P}^0$. In contrast, the CE objective is defined as the KL-divergence $D\_{\text{KL}}(\mathbb{P}^\star\|\mathbb{P}^{b\theta})$ which expands into a path-integral action term $\frac{1}{2}\int_0^T\||\boldsymbol{b}\_\theta(\boldsymbol{R}_t, \boldsymbol{V}_t)\||^2dt$ using Girsanov's theorem. This explicitly regularizes the whole trajectory, rewarding gradual, smooth transport through intermediate states.
> >
>
> ---
>
> If we have sufficiently addressed all your concerns, we kindly ask that you support the acceptance of our paper for poster presentation, and if concerns remain, we are happy to continue the discussion!
>
> Thank you,
>
> The Authors

---

### Author Response · Authors · 2025-11-16
**General Response to Reviewers and Overview of Key Updates to Manuscript**

# General Response

We sincerely thank all reviewers for their valuable and insightful comments. Since the original submission of our manuscript, we have made the following **key updates** to improve both the presentation and empirical validation of our paper.

### Overview of Key Updates

- In Sec. 5.2, we provide **quantitative** and **qualitative** experiments on Alanine Dipeptide and three fast folding proteins: **Chignolin** (10 amino acids; 138 atoms), **Trpcage** (284 atoms; 20 amino acids), and **BBA** (504 atoms; 28 amino acids). We compare against non-ML and ML baselines, including unbiased MD (UMD), steered MD (SMD), PIPS (Holdijk et al. 2023), and TPS-DPS (Seong et al. 2024). **Notably, we show superior target hit percentage for all proteins against all baselines.**
- Instead of using a target-aware base dynamics for the cell perturbation modelling experiment, we define the base dynamics as just the **natural gradient of the data manifold** that points in the direction of the largest increase in data density. This allows us to rigorously evaluate the ability of EntangledSBM to **fully capture the cellular dynamics as a result of the drug perturbation.**
- In Sec. 3 and App. B.2, we provide a deeper, intuitive and theoretical justification for integrating dependencies on joint particle positions and velocities.
- To demonstrate the significance of the **cross-entropy (CE) objective that we formalize in Sec. 4.2**, we provide additional background on the alternative log-variance (LV) objective in App. B.1 and a rigorous evaluation of the theoretical and empirical benefits of the CE objective over the LV objective in **App. F**.
- To assess the computational efficiency of our framework, we report the average training and sampling times per rollout across molecular dynamics systems of increasing atomic complexity in **Table 6 in App. E.3**.
- We provide the full codebase for our models and experiments in the anonymous repository: [https://anonymous.4open.science/r/EntangledSBManon](https://anonymous.4open.science/r/EntangledSBManon/README.md)

---

### Author Response · Authors · 2025-11-29
**Summary of Reviewer Discussion and Revisions for Reassigned AC (1/2)**

**Dear Area Chair,**

First of all, we sincerely thank all of our reviewers for their constructive and valuable feedback, as they have truly improved our paper. Given the AC reassignment, we would like to provide our new AC with a **concise summary of the outcomes of the reviewer discussion for our paper and the improvements we have made to our manuscript since the original submission**.

---

### **Outcomes of Reviewer Discussion**

In an effort to make the decision process as easy as possible for the new AC, we will provide the **original scores of each reviewer** and a **summary of our discussion with each reviewer below:**

- **Reviewer UM1c [Score 4; Confidence 3]** highlighted that our paper is **well-structured** and **provides clear theoretical derivations**. They gave a borderline score with primary concerns including limited experimental scale and lack of efficiency analysis, and we have added several experiments and **revised our paper to address each of their concerns fully as highlighted in the following points:**
    - **Addressing limited experimental scale and lack of relevant baselines:** In Sec 5.2 of the revised paper, we included significant additional quantitative results on simulating **all-atom transition paths** on larger fast-folding (up to 504 atoms), and demonstrate SOTA performance in target hit percentage and RMSD against the baselines mentioned by the reviewer, including PIPS (Holdijk et al. 2023) and TPS-DPS (Seong et al. 2025).
    - **Addressing lack of efficiency analysis:** We add **Table 6 in App. E.3, which summarizes the average training and sampling times per rollout across molecular dynamics experiments**, demonstrating that the proposed framework scales favourably with increasing system complexity.

    **Following our response, there has been no further discussion, and no further concerns were raised.** Given that we have addressed each concern raised with additional experiments with positive results, we believe that if the reviewer discussion were allowed to continue, our discussion with Reviewer UM1c would have resulted in a positive score.

- **Reviewer GHiq [Score 2; Confidence 2]** gave us an original score of 2 because they had a “**hard time understanding the setup hence my comments beyond setups and proof questions are less reliable**” but mentioned they were “**more than happy to be convinced that [their] current score is due to lack of understanding rather than issues of the work**.” Their primary confusion was in our experimental setup and why our method was uniquely positioned to solve the transition path sampling problem in MD and perturbation modelling in single-cells, and we were really happy to have had an engaging discussion with this reviewer, as we believe it could be helpful to clear up any confusion on our experimental setup for other readers.
    - Following our initial response to their concerns, they stated, “I appreciate the authors' revision and answers, especially the part treating Girsanov more carefully addressed my concern there.” However, they had follow-up questions regarding the **setup of the experiments** to which we responded with small clarifications on notation and a thorough explanation of how our method is uniquely positioned to simulate **underdamped Langevin dynamics on the phase space**, which cannot be solved analytically or with vanilla SB.
    - **After our response to their follow-up, their most recent comment stated:**

        > I think it is addressing an important problem […] I want to emphasis that all of the back and forth are for me to understand the setup better so I can confidently say whether I think this is a good paper or not and I think it will also help other readers with similar background as I do --- so thank again for the authors responses. I am more than happy to change the score after the left over confusions I have got cleaned, we are getting there.”
        >

        **They also highlighted the significance of our approach for MD simulation by saying:**

        > If I indeed understand it correctly, I am quite convinced this is a realistic application and the approach is natural and seems to work well.
        >

        Since then, we have carefully addressed their final confusions regarding the experimental setup, which mainly concerned the applicability of our method in single-cell simulation, and no further questions were raised. **We believe that if the reviewer discussion were allowed to continue, this discussion would result in a positive score.**

---

> ### Author Response · Authors · 2025-11-29
> **Summary of Reviewer Discussion and Revisions for Reassigned AC (2/2)**
>
> - **Reviewer 6fMu [Score 6; Confidence 3]** had an positive evaluation of our paper, and highlighted that the paper was “**well-written**,” “**parameterizations of the bias force are sound and novel**”, our “**method can solve real-world tasks**”, and that “**all statements are supported by clear derivations and proofs.**” They raised minor concerns and questions which we responded to with our expanded experimental results against baselines in Sec 5.2 and a clearer description of the motivation of our work in Sec 3. **Their final response maintained a positive evaluation, and wrote**:
>
>     > I read the revised version and appreciate taking my concerns into account. As I stated above, I consider this paper **above the acceptance threshold and a good contribution.**
>     >
>
> ---
>
> ### **Summary of Key Updates**
>
> All our reviewers’ feedback contributed significantly to the improvement of our paper, from adding empirical evidence of scalability to increasingly complex bimolecular systems and superior performance against baseline methods to strengthening the theoretical justifications of our framework. **We provide a concise overview of the key updates to our paper in the points below, and we would greatly appreciate it if the AC could evaluate the current version of our manuscript, which incorporates these improvements:**
>
> - In Sec. 5.2, we provide **quantitative** and **qualitative** experiments on Alanine Dipeptide and three fast folding proteins: **Chignolin** (10 amino acids; 138 atoms), **Trpcage** (284 atoms; 20 amino acids), and **BBA** (504 atoms; 28 amino acids). We compare against non-ML and ML baselines, including unbiased MD (UMD), steered MD (SMD), PIPS (Holdijk et al. 2023), and TPS-DPS (Seong et al. 2024). **Notably, we show superior target hit percentage for all proteins against all baselines.**
> - Instead of using a target-aware base dynamics for the cell perturbation modelling experiment, we define the base dynamics as just the **natural gradient of the data manifold** that points in the direction of the largest increase in data density. This allows us to rigorously evaluate the ability of EntangledSBM to **fully capture the cellular dynamics as a result of the drug perturbation.**
> - In Sec. 3 and App. B.2, we provide a deeper, intuitive and theoretical justification for integrating dependencies on joint particle positions and velocities.
> - To demonstrate the significance of the **cross-entropy (CE) objective that we formalize in Sec. 4.2**, we provide additional background on the alternative log-variance (LV) objective in App. B.1 and a rigorous evaluation of the theoretical and empirical benefits of the CE objective over the LV objective in **App. F**.
> - To assess the computational efficiency of our framework, we report the average training and sampling times per rollout across molecular dynamics systems of increasing atomic complexity in **Table 6 in App. E.3**.
> - We provide the full codebase for our models and experiments in the anonymous repository: [https://anonymous.4open.science/r/EntangledSBManon](https://anonymous.4open.science/r/EntangledSBManon/README.md)
>
> ---
>
> We would like to thank our original reviewers again for their time and effort in reviewing our paper. We regret that we can no longer engage in discussion with our reviewers, and would like to assure them that each of their reviews has improved our paper in many aspects.
>
> We kindly ask that our new AC acknowledge the contributions of our paper and our efforts to thoroughly address each of our reviewer comments and support our paper for acceptance to ICLR.
>
> Thank you,
>
> The Authors

---

### Meta-Review · Area_Chair_59wP · 2026-01-07

**Summary:**

This paper defines a formulation of the Schrödinger bridge problem for multi-particle systems, where the dynamics of the particles are given by drift-diffusion processes in (position, momentum) space with coupling imposed on the second-order (force) term, and proposes to use stochastic optimal control algorithms (using two possible objectives) to solve it.

The main points made by the reviewers are the following.

Strengths:
1. Good writing and mathematical exposition (UM1c, 6fMu)
2. Interesting/useful/important application (GHiq, 6fMu)

Weaknesses/concerns:
1. Scale of experiments (UM1c, 6fMu)
2. Lack of comparison to baselines, novelty over prior work (UM1c, 6fMu)
3. Analysis requested on the computational efficiency (UM1c)
4. Paper difficult to follow, even by a reviewer familiar with SB; many questions about setup, especially the multi-particle formulation (GHiq, some from 6fMu)
5. Question about Girsanov conditions (GHiq)

As detailed below, many of these concerns have been at least partially addressed in the rebuttal. However, some issues remain, and it should be noted that none of the reviewers had a high confidence in their initial score. Overall, the paper is below the acceptance threshold in my view. I would encourage the authors to address the concerns about comparisons to past work, ablations of method choices, and clarity of exposition in a future resubmission.

**Reviewer Concerns:**

1. This is addressed by larger-scale experiments added to the rebuttal; 6fMu considers this resolved and UM1c's questions appear to have been addressed.
2. This is partially resolved: the authors clarified the novelty in the rebuttal and gave baseline comparisons; however, I find some unusual claims about comparisons and novelty in both the rebuttal and the paper.
  - The claim "Furthermore, [3] limited their quantitative evaluation to Alanine Dipeptide, indicating that the method is limited in its scalability to larger, real-world systems" does not seem to do justice to [3], which also considers chignolin but does not perform quantitative evaluation apparently *because suitable learning-based baselines were not available*.
  - The LV objective for diffusion-based sampling is said to originate in two 2025 works applying it to discrete-space problems (pdf line 846), which ignores the large volume of work using this objective on continuous spaces as considered by the present paper (e.g., [Richter et al., arXiv:2307.01198, ICLR'24], [Vargas et al., arXiv:2307.01050, ICLR'24], [Sendera et al., arXiv:2402.05098, NeurIPS'24], [Blessing et al., arXiv:2503.01006, ICLR'25] (underdamped case), inter alia). Note also that while the importance-weighted CE provides a biased but consistent approximation to the forward KL gradient, the LV, *when trained on policy*, is an unbiased estimator of the reverse KL gradient, and that objectives based on differentiable simulation can be used to optimise the reverse KL *directly* without bias (as in [Zhang et al., arXiv:2111.15141, ICLR'22] and [Vargas et al., arXiv:2302.13834, ICLR'23], inter alia). Why not consider these prior works and direct optimisation of the reverse KL? Is it prohibitive in this setting? (This may be beyond the scope of a first paper on the multi-particle SB as considered here, but why not also consider the many techniques for combining simulation/MC and learning-based methods for diffusion sampling (e.g., [Chen et al., arXiv:2412.07081, ICLR'25], [Albergo et al., arXiv:2410.02711, ICML'25], and the above [Sendera et al.], among others)?)
  - The metrics in which EntangledSBM improves over baselines are RMSD and THP, which measure fit of the final position to the target. However, EntangledSBM significantly underperforms all baselines in ETS -- which measures transition path quality -- in nearly all cases. This seems a significant drawback of a method intended to model transition dynamics; we have no evidence the proposed algorithm is modelling the pathwise posterior correctly, not just approximating the endpoint distribution well.
3. Partially addressed, as the new analysis is not satisfying without context on cost of baselines. There is still (almost) no analysis of the effect of different modelling choices and hyperparameters.
4. Many of GHiq's concerns were addressed, but I still find some parts of the method exposition difficult to follow. I tried to look at the code to understand better what was done, but the link provided in the rebuttal is broken (only the readme is accessible). On a minor note, there are still some mathematical inaccuracies. (For example, how to understand "$R_B\in\pi_B$" in the statement and proof of Proposition 4.1, where $\pi_{\mathcal{B}}$ is a distribution (or, in places in the paper, its density)?)
5. Addressed in the rebuttal and edits to the paper.

**Reviewer Scores:**

The original scores were 4 (UM1c), 2 (GHiq), and 6 (6fMu). A generous estimate would assume that all reviewers would raise their scores by 2 points. A more conservative estimate assumes that only GHiq would raise their score (due to the improvements in the clarity), while UM1c would keep theirs (because of the concerns about comparisons detailed above) and 6fMu would also maintain their borderline-positive assessment. This places the average final score between 4.67 and 6, which is near borderline.

---

### Decision · Program_Chairs · 2026-01-26

Reject